Natural Hazards and Earth System Sciences
EGU
Discussions

# Analyzing the informative value of alternative hazard indicators for monitoring drought risk for human water supply and river ecosystems at the global scale

Claudia Herbert[1] and Petra Döll[1,2]

[1]Institute of Physical Geography, Goethe University Frankfurt, Frankfurt am Main, 60438, Germany
[2]Senckenberg Leibniz Biodiversity and Climate Research Centre Frankfurt (SBiK-F), Frankfurt am Main, 60325, Germany

*Correspondence to*: Claudia Herbert (c.herbert@em.uni-frankfurt.de)

**Abstract**

Streamflow drought hazard indicators (SDHI) are mostly lacking in large-scale drought early warning systems (DEWS). This paper presents a new systematic approach for selecting and computing SDHI for monitoring drought risk for human water supply from surface water and for river ecosystems that is also relevant for meteorological or soil moisture drought. We recommend considering the habituation of people and ecosystems to the streamflow regime (e.g., a certain interannual variability or relative reduction of streamflow) when selecting indicators. Distinguishing four indicator types, we classify indicators of drought magnitude (water deficit during a pre-defined period) and severity (cumulated magnitude since onset of the drought event). We quantify eight existing and three new SDHI globally using the global hydrological model WaterGAP2.2d. We recommend streamflow hazard indicators that should be included in large-scale DEWS as they are suitable for risk systems that are differently adapted to low water availability and characterized by either perennial or intermittent streamflow regime and the existence or not of large reservoirs. Drought magnitude is best quantified by return period or relative deviation from mean, and severity by return period or water volume below a threshold relative to mean annual streamflow. Both anomaly and deficit indicators should be provided.

## 1 Introduction

Drought occurs when there is a prolonged time period with less water than normal in different components of the hydrological cycle (van Loon et al., 2016) but the term drought also has the connotation that during the drought period there is less water than required (Popat and Döll, 2021). No universal definition of "drought" exists (Lloyd-Hughes, 2014). While drought is a local to regional phenomenon, its impacts can have transnational to global dimensions, in particular related to crop production and trade (Wilhite and Glantz, 1985; van Loon, 2015; UNECE, 2015). Streamflow drought in transboundary basins implies direct international impacts. Hence, global-scale assessment, monitoring and forecasting of drought hazards or risks have the potential to support drought risk management (Pozzi et al., 2013).

Drought poses numerous risks to humans and ecosystems. A specific drought risk is a function of hazard, exposure, and vulnerability, while the term "drought impact" relates to the manifested risk (Field and Barros, 2014). In general, the term "hazard" refers to the physical event, "exposure" to the presence of people or ecosystems that could be negatively affected, and "vulnerability" to the susceptibility of a system to drought impacts and its (short-term) coping and (long-term) adaptation capacity (Field and Barros, 2014).

To assess any risk, it is necessary to first specify the "risk of what for whom". As an example, the risk of "not being able to provide enough water to fulfill the customers' water demand during times of lower water availability than normal" constitutes a drought risk for water supply companies. Drought risk indicators, and thus drought hazard, exposure and vulnerability indicators, should be designed specifically for the targeted risk (Lloyd-Hughes, 2014; Spinoni et al., 2018; Hagenlocher et al., 2019). If, for the example risk, the water supply source is a river, a suitable drought hazard indicator should be based on information on streamflow or on water storage in the upstream reservoir. Previous research has revealed that there is often no common understanding among stakeholders about drought hazard concepts (Steinemann et al., 2015), and a very large number of drought hazard indicators has been proposed and applied by experts without stringent consideration of the targeted risks (Wilhite and Glantz, 1985; Lloyd-Hughes, 2014). To identify a suitable risk-specific hazard indicator, the first step is to decide which water flow or storage should be taken into account, e.g. precipitation, soil moisture or streamflow, as the determined drought hazards depend on the considered physical variable (Satoh et al., 2021). Even after determining the appropriate risk-specific physical variable, e.g. streamflow, there is still a large choice of possible drought hazard indicators to quantify the occurrence or severity of streamflow droughts (Yihdego et al., 2019). A stakeholder survey encompassing 33 regional to global drought early warning systems (DEWS) revealed that streamflow percentiles are most often used to assess streamflow drought hazard, e.g. in the US Drought Monitor (Bachmair et al., 2016). Other indicators include the Palmer Hydrological Drought Severity Index (Palmer, 1965), cumulative streamflow anomalies (Fleig et al., 2006; Lehner et al., 2006; van Loon et al., 2012; Heudorfer and Stahl, 2017), and the standardized streamflow (Modarres, 2007; Nalbantis and Tsakiris, 2009) or runoff index ( Shukla and Wood, 2008; Satoh et al., 2021). Investigations on how to select the optimal streamflow drought hazard indicator or on the sensitivity of indicated drought hazard to the choice of streamflow drought indicator are missing.

In DEWS, streamflow drought hazard indicators are anyway rare, while drought hazard indicators based on meteorological variables, soil moisture, and remotely-sensed vegetation conditions dominate (Bachmair et al., 2016). At the continental scale, only the European Drought Monitor provides a streamflow drought hazard indicator (Cammalleri et al., 2016a), which has also been tested for global implementation in the Global Drought Observatory (Cammalleri et al., 2020). There is currently no global-scale operational streamflow drought hazard monitoring system.

Streamflow drought hazard can be estimated using either observed or modeled streamflow data. If no such data are available, streamflow drought hazard is estimated by applying meteorological indicators such as the standardized precipitation index SPI (McKee et al., 1993) and the standardized precipitation-potential evaporation index SPEI (Vicente-Serrano et al., 2010), where the delayed response of streamflow to below-normal precipitation is considered through longer averaging periods, i.e., by comparing mean precipitation conditions of the preceding n months to the respective n months of the reference

period (SPIn), where n can range between 1 and 24 (Gevaert et al., 2018). However, studies have shown that meteorological indicators have limitations in describing hydrological drought processes and suggest including streamflow drought indicators in drought management (Haslinger et al., 2014; Blauhut et al., 2016; Laaha et al., 2017), as the meteorological drought signal is modified and propagates due to water transport through soil, groundwater and surface water bodies (van Loon, 2015). Where streamflow observations are not available, hydrological models can compute the response of streamflow to precipitation and other climatic variables to determine spatially and temporally continuous drought hazard indicators that take into account the different characteristics of the river basins (Lehner et al., 2006). Still, the meteorological variables precipitation and potential evapotranspiration are known to be major drivers of streamflow and its variability, and hydrological models, in particular large-scale models, suffer from significant uncertainties such that the added value of simulated streamflow drought hazard indicators should be assessed.

Streamflow drought hazard indicators are commonly classified into threshold-based and standardized indicators (van Loon, 2015). The threshold level method (TLM) was first applied by Yevjevich (1967), who defined that a drought event begins when streamflow falls below a certain threshold (e.g. a percentile) and ends as soon as the threshold is exceeded. Then, drought magnitude is the streamflow deficit at the considered time period (e.g. the difference between the threshold streamflow and the actual streamflow in a certain month), while drought severity is equivalent to the cumulative magnitude since the beginning of the drought event (or of the whole drought event). The standardized streamflow indicator (SSI) quantifies the anomaly of streamflow during a certain time period from long-term mean streamflow in units of standard deviation and is computed like the SPI. Negative values quantify the drought magnitude per time step. The standardized streamflow index SSI has been applied using a 1-month averaging period (SSI1) (Zaidman et al., 2002; Modarres, 2007; Nalbantis and Tsakiris, 2009) as well as longer averaging periods (SSI3, SSI6, SSI12) (Svensson et al., 2017; Wan et al., 2021). However, classification in threshold-based and standardized indicators is somewhat misleading, as standardized indicators quantify drought magnitude, while TLM indicators quantify drought severity (Yevjevich, 1967). TLM and standardized drought indicators therefore quantify different drought characteristics. To quantify drought severity by cumulated standardized indicators, setting of a threshold for drought occurrence is necessary as is the case for TLM indicators (McKee et al., 1993; Barker et al., 2019, van Oel et al., 2018, Tijdeman et al., 2020). A standardized indicator value (e.g. SSI or SPI) of -1 or -0.84 is often defined as threshold for drought occurrence (Agnew, 2000). From time series of standardized or non-standardized severity, further drought characteristics can be derived such as the probability of a drought event of a certain severity (e.g. Cammalleri et al., 2016a; Popat and Döll, 2021).

In the existing literature on drought hazard, drought concepts and indicator definitions are rarely made explicit. While there is a plethora of drought hazard indicators (e.g. WMO and GWP, 2016), there is no clear classification and guidance on the selection of optimal indicators. Moreover, the term drought severity is sometimes used to describe drought magnitude and vice versa (Steinemann et al., 2015, Vidal et al., 2009; López-Moreno et al., 2009). What is also not explicitly considered and described when presenting drought hazard indicators is what is assumed to be "normal". In most descriptions of drought indicator calculations, no matter which physical variable is considered, it is not made explicit that "normal" is the long-term mean

value of the physical variable for the calendar month (and not, for example, the long-term annual mean) and that it is therefore assumed that people and ecosystems are habituated to the seasonality of water availability. While some systems may be vulnerable to anomalies of monthly streamflow, e.g., river biota or farmers pumping river water, there are also water supply systems with man-made reservoirs that can store streamflow over three months, which are therefore vulnerable to anomalies of 3-month streamflow averages. Application of bboth TLM (with low streamflow percentile as threshold, e.g. 20$^{th}$ percentile) and standardized indicators of streamflow drought implies that people or the ecosystem are habituated to the interannual variability of streamflow, which means that in regions with a high interannual variability of streamflow (often semi-arid regions) a drought condition is identified with the same frequency as in regions with a low interannual variability (often humid regions). In case of a drought that occurs in 1 out of 5 years, equivalent of SSI = -0.84 (monthly streamflow is 0.84 standard deviation lower than normal), there may be only 20% less streamflow than normal in a humid region but 60% less than normal in a semi-arid region. The question is if drought hazard is actually the same in these two regions, as would be indicated by both SSI and a TLM indicator with a low streamflow percentile as threshold. If alternatively, percent deviations from mean streamflow were used to quantify the anomaly, i.e., how much less water than normal is available, it would be assumed that people or ecosystems are used to the long-term mean water availability and not the interannual variability. This assumption could also be operationalized by choosing the long-term mean (either annual or monthly) as threshold for a TLM indicator, or, similarly, the median. There may be situations where habituation to the mean or median is a more reasonable assumption than the assumption that people or ecosystems are habituated to interannual variability, in particular if interannual variability is large. Only if people or ecosystems are used to the seasonality and the interannual variability of streamflow (or of other water flows and storages), drought hazard can be quantified appropriately by typical TLM and standardized streamflow hazard indicators. Clearly, the conception of hazard indicators needs to take into account the habituation and thus vulnerability of the system at risk.

A further consideration in designing streamflow drought hazard indicators is how to conceptualize drought in intermittent or highly seasonal streamflow regimes. If periods of zero flow are a normal part of the streamflow regime, as it is the case in arid regions, then it is meaningless to assess streamflow deficits during these periods. Hence, arid regions are often excluded from global drought analyses (Corzo Perez et al., 2011; Prudhomme et al., 2014; Spinoni et al., 2019). Some authors tested rather high percentiles as thresholds to characterize drought in intermittent streamflow regimes (e.g. the 80$^{th}$ percentile, i.e. Q20, the streamflow that is exceeded in two out of ten months) (Woo and Tarhule, 1994; Tate and Freeman, 2000; Fleig et al., 2006). This approach, however, has been criticized as it is not consistent with the anomaly concept of drought (van Huijgevoort et al., 2012). To overcome these limitations, van Huijgevoort et al. (2012) introduced a method to identify streamflow drought at the global scale that is also applicable for intermittent rivers. It combines the TLM with the consecutive dry period method (CDPM) for streamflow, in analogy to the consecutive dry days (CDD) indicator for precipitation (Vincent and Mekis, 2006; Griffiths and Bradley, 2007). Using this combined method, a drought in a period with streamflow identified with the TLM is allowed to continue in a subsequent zero-flow period. Very short periods of zero flow are excluded from the assessment. Although more sophisticated compared to the TLM alone, the combined method as described in van Huijgevoort

et al. (2012) may be too complex to be applied in DEWS. Moreover, the final scaling procedure of percentiles in months where both TLM and CDPM apply might result in thresholds that are not intuitive.

This paper analyzes which drought hazard indicators are suitable for assessing and monitoring drought risk for human water supply from surface water and for river ecosystems, taking into account streamflow regime, habituation of people and river ecosystems to streamflow availability as well as human water use (as alternative threshold). We propose a systematic approach to indicator selection, which encompasses the explicit consideration of habituation as well as a new classification scheme for drought hazard indicators that includes four indicator types and two levels of drought characterization, magnitude, and severity, both of which are either quantified directly, e.g., in terms of water volume, or in terms of frequency or probability. Applying the global water resources and use model WaterGAP2.2d, we compare drought hazard globally during 1986-2015 as determined by eight existing and three newly developed hazard indicators. The former include SPI12, SPEI12, SSI1, SSI12, the percentage deviation of monthly streamflow from mean calendar month streamflow (RDQI1), the percentage deviation of mean streamflow over the preceding twelve months from overall monthly mean streamflow (RDQI12) and the TLM cumulative streamflow drought hazard indicators CDQI1-Q50 and CDQI1-Q80. The new indicators encompass an indicator that is suitable for detecting drought in intermittent and highly seasonal flow regimes (CDQI1-Q80-HS) and two indicators that take into account human surface water demand and environmental flow requirements (CDQI1-WUs and CDQI1-WUs-EFR).

The following section describes how streamflow and other variables required for the computation of the streamflow hazard indicators were computed and defines the eleven investigated streamflow drought hazard indicators. In Sect. 3, we present the new systematic approach for selecting and computing streamflow hazard indicators, and we analyze spatial and temporal patterns of the indicators. In Sect.s 4 and 5, we give recommendations on the general suitability of the indicators as well as for large-scale applications and we draw conclusions.

## 2 Methods and data

### 2.1 Global-scale simulation of streamflow, surface water use and PET

Hydrological drought hazard indicators were computed using output from the global water availability and water use model WaterGAP2.2d (Müller Schmied et al., 2021). WaterGAP2.2d has a spatial resolution of 0.5 degrees latitude by 0.5 degrees longitude (55 km × 55 km at the equator) and covers the whole global land area except Antarctica. The model consists of the WaterGAP Global Hydrology Model WGHM and five water use models for the sectors households, manufacturing, and cooling of thermal power plants (Flörke et al., 2013) as well as irrigation and livestock. WGHM computes daily time series of fast surface and subsurface runoff, groundwater recharge, and streamflow as well as water storage variations in canopy, snow, soil, groundwater, lakes, reservoirs, wetlands, and rivers. Model input includes time series of climate data between 1901 and 2016 and physio-geographic information, such as land cover, soil type, relief, and hydrogeology. For this study, WaterGAP was forced by the WFDEI-GPCC climate data set (Weedon et al., 2014), which was developed by applying the forcing data

methodology from the EU project WATCH on ERA-Interim reanalysis data. Potential evapotranspiration (PET), required for the calculation of SPEI12, was computed using the Priestley-Taylor equation. In addition to the standard model run ("ant": anthropogenic), in which the impact of human water use and man-made reservoirs on streamflow is simulated, naturalized ("nat") conditions were computed by turning off these two types of human activities. Daily model outputs of anthropogenic and naturalized streamflow (Qant and Qnat), PET, and surface water abstractions (WUs) were aggregated to monthly time series. WaterGAP total runoff is calibrated against long-term mean annual streamflow at 1319 gauging stations worldwide covering approximately 54% of the Earth's land area (except Greenland and Antarctica). A detailed model description and evaluation can be found in Müller Schmied et al. (2021).

### 2.2 Streamflow drought hazard indicators

Eleven streamflow drought hazard indicators (Table 1) were computed for the whole land area except Greenland and Antarctica with a spatial resolution of 0.5° using monthly time series of WaterGAP model output for the reference period 1986-2015. For computing each indicator, we used the 30 monthly values available for each of the 12 calendar months individually to determine distributions, thresholds, and deficits. Moreover, selected indicators were quantified for WaterGAP calibration stations using monthly streamflow observations provided by the Global Runoff Data Centre (GRDC, 2019) for the period 1986-2015 (Figs. 2a and 2b and Sect. 3.5).

#### 2.2.1 Standardized meteorological indicators SPI and SPEI

**SPI** time series were computed at the global scale following the method described in McKee et al. (1993). Monthly precipitation data are first fitted to a probability distribution (e.g. gamma or Pearson Type III) and then transformed to the standard normal random variable Z (Eq. 1) (also termed z score), which is the SPI, following an approximation method introduced by Abramowitz and Stegun (1965). The standard normal distribution is characterized by a mean of zero and standard deviation of 1. A value of -1, for example, indicates that the precipitation value deviates from the long-term mean by one standard deviation.

$$Z = (X - \mu)/\sigma \quad (1)$$

with X = variable, $\mu$ = mean, and $\sigma$ = standard deviation.

The SPI can be quantified for different averaging periods of typically 1 to 36 months. In the present study, SPI time series were computed using a 12-month averaging period (SPI12), which is recommended for the assessment of hydrological drought impacts (WMO and GWP, 2016). For a limited sensitivity analysis, SPI time series with averaging periods of 3, 6, 9, and 12 months were derived for 218 WaterGAP calibration stations (Sect. 3.5). The indicator was computed using the SCI package for R (Gudmundsson and Stagge, 2016), fitting a gamma distribution to the precipitation time series.

**SPEI12** time series were calculated at the global scale according to the method presented in Vicente-Serrano et al. (2010) using a 12-month averaging period. Similar to SPI, the SCI package for R was utilized, however, applying the Log-logistics distribution as recommended for the SPEI (Vicente-Serrano et al., 2010; Vicente-Serrano and Beguería, 2016).

### 2.2.2 Standardized streamflow anomaly indicators SSI1 and SSI12

**SSI1** was computed for Qant analogously to SPI1 following the method provided in Kumar et al. (2009) using the R package fitdistrplus and the gamma distribution, taking into account mean monthly streamflow for each of the 12 calendar months. Based on the one-sample Kolmogorov–Smirnov test (KS test) at the 0.05 significance level, the fits were rejected in 17% to 21% of all grid cells (excluding Greenland) depending on the calendar month.

**SSI12** is computed like SSI1, but with an averaging period of 12 months. For SSI12, the fits were rejected in around 6% of all grid cells (excluding Greenland) with only slight variations among the calendar months.

### 2.2.3 Cumulative streamflow deficit indicators CDQI1-Q50, CDQI1-Q80 and CDQI1-Q80-HS

**CDQI1-Q50** is the cumulative, volume-based streamflow deficit computed following the threshold level method (TLM) (Sect. 1). It should be noted that the term "deficit", which is generally used for the TLM, refers to the negative anomaly below a selected threshold, and not to an unsatisfied water demand. With CDQI1-Q50, a deficit is defined to occur if modeled monthly streamflow is lower than the 50$^{th}$ percentile (median) of the long-term mean calendar month streamflow. The empirical percentile Q50 was computed in R using the quantile function with the default quantile algorithm. In each month, the streamflow deficit volume is calculated as the difference between the median of all 30 calendar month streamflow values during the reference period and the water volume that was actually transported in the stream in this month. The last deficit month is the last month of the drought event. Monthly deficits (drought magnitude) are accumulated for all drought months to obtain severity S. Any streamflow surplus over the median in a single month between two deficit months does not decrease the cumulative deficit value. The cumulative streamflow deficit (in units of $m^3$) is normalized by mean annual streamflow (in units of $m^3$). A value of 2 [-], for example, indicates that the cumulative streamflow deficit in a certain month is twice the mean annual streamflow. Following Spinoni et al. (2019), a drought event is defined to start with at least two consecutive months with a deficit and it ends (deficit set to zero) if there are two consecutive months without a deficit (two months criterion, 2mc). This approach avoids that short-term streamflow deficits that hardly pose a drought hazard to humans and other biota are defined as drought events. Q50 as a rather high threshold can be viewed as a "conservative upper bound for low flows" (Smakhtin, 2001: 153).

Streamflow intermittency generally poses a problem, as in grid cells where the threshold (in this case Q50) is zero in a particular calendar month, no drought is ever identified with a threshold-based approach. To overcome this problem, CDQI1-Q50 allows an existing drought to continue during months with Q50=0, but only if Q in the respective month is also zero. In months where Q50 is zero, but Q exceeds zero, the drought event ends. This approach implies that a drought can be prolonged, but never begin in calendar months with Q50=0.

Natural Hazards and Earth System Sciences
EGU
Discussions

**CDQI1-Q80** was calculated in the same manner as CDQI1-Q50, however, using the empirical percentile Q80 per calendar month as threshold, which is the monthly streamflow value that is exceeded in 80% of all 30 calendar months. Daily or monthly Q80 is often used as a threshold for defining the onset and termination of a streamflow deficit period (van Huijgevoort et al., 2014; van Loon et al., 2014; Heudorfer and Stahl, 2017; Laaha et al., 2017), but the selected threshold should represent local water requirements (including environmental flow) (Cammalleri et al., 2016a). The choice of threshold is generally subjective, as impact data for validation is rarely available (Heudorfer and Stahl, 2017).

**CDQI1-Q80-HS** is a variant of CDQI1-Q80 suitable in intermittent and highly seasonal (HS) streamflow regimes where people strongly rely on water storage in man-made reservoirs that needs to be replenished by streamflow. It allows an existing drought to continue in any month where Q80 is zero also if the current streamflow Q exceeds zero. However, the cumulative deficit is reduced by any streamflow surplus over the calendar month Q80. The rationale behind this approach is that streamflow during low-flow months (i.e., calendar months where Q80 is zero) is not relevant for people relying on large reservoirs. Below-normal water storages can only marginally be replenished during a low-flow period, and hence drought severity should remain at the level of the preceding high-flow period. Like **CDQI1-Q80,** a drought can be prolonged but never begin, in months with Q80=0.

### 2.2.4 Empirical percentiles EP1 and cumulative empirical percentiles CEP1(20%)

Empirical streamflow percentiles **EP1** were computed per calendar month following Eq. (2) with an averaging period of one month. EP1 expresses the frequency of non-exceedance, while the inverse is the return period, in years.

$$EP1 = rank(Q)/n \tag{2}$$

where rank(Q) is the rank of a streamflow value of a certain calendar month and n is the sample size, i.e., the number of years in the reference period. Rank 1 was assigned to the smallest streamflow value. If a sample contained several months with the same streamflow value, the largest rank among these months was assigned to the tied streamflow values. For a calendar month comprising, for instance, 26 out of 30 months with zero streamflow, a value of EP1=26/30 would be assigned to the respective 26 months corresponding to a return period of 1.2 years. This method slightly adjusts the approach by Tijdeman et al. (2020), who used the average rank among the tied values. In the given example, this would result in EP1=0.45 and a return period of 2.2 years for the first 26 values. In this study, we chose the largest EP1 for tied values to reflect that frequent streamflow values have a high frequency of non-exceedance and a low return period assuming that people and the ecosystem are habituated to more frequent values including zero streamflow.

The cumulative percentile-based anomaly **CEP1(20%)** was computed in a similar way to CDQI1-Q80 using the 20$^{th}$ percentile (the value that is exceeded in 8 out of 10 months) of the respective 30 EP1 values as threshold per calendar month. Moreover, CEP1 allows an existing drought event to continue during months where both Q80 and the current streamflow are zero.

#### 2.2.5 Relative deviation from mean conditions RDQI1, RDQI12 and cumulative CRDQI1(-50%)

**RDQI1** is the relative deviation of monthly streamflow from mean calendar month streamflow (MMQ) in percent. In each month, it is calculated as the difference between monthly streamflow and the respective MMQ, which is then divided by MMQ. **RDQI12** is the relative deviation of mean streamflow during the preceding 12 months (in $km^3$ $month^{-1}$) from mean annual streamflow (in $km^3$ $month^{-1}$) during the reference period. In this study, RDQI12 is only assessed for two gauging stations (Fig. 2 and Sect. 3.2), but not at the global scale.

The cumulative relative deviation **CRDQI1(-50%)** with a threshold of RDQI1=-50% was derived like CDQI-Q80. Months with MMQ=0 (i.e., the relative deviation is not computable in this calendar month) were defined to end a drought event assuming that people are habituated to zero streamflow in this month.

#### 2.2.6 Water deficit indicators CDQI1-WUs and CDQI1-WUs-EFR

The water deficit indicators **CDQI1-WUs** and **CDQI1-WUs-EFR** are computed like CDQI1-Q80 but using as thresholds mean monthly potential surface water abstraction WUs, and WUs plus environmental flow requirement (EFR), respectively. Following Richter et al. (2012), EFR is assumed to be 80% of mean monthly naturalized streamflow Qnat per calendar month such that 12 EFR values are obtained per grid cell. WUs is the simulated water demand (potential water abstractions from surface water bodies) and not the actual water abstractions (Müller Schmied et al., 2021), but both values are similar in most grid cells. The satisfied (or actual) water use is not suitable to identify periods of water deficit because it decreases along with water availability during drought. Cumulative deficits are normalized by mean annual streamflow. The indicators were not computed in grid cells where mean annual surface water demand in the reference period is zero (approx. 9% of all grid cells excluding Greenland).

#### 2.2.7 Probability of drought events of a certain severity

Following the approach of Cammalleri et al. (2016a) to compute the low-flow index LFI, the probability of drought events of a certain severity was computed for six cumulative indicators, CEP1(20%), four CDQI1 variants (thresholds Q50, Q80, WUs, and WUs+EFR) and CRDQI1(-50%). First, the partial duration series of drought events was derived based on the severities of all drought events of the reference period. Grid cells with less than six drought events were excluded. The exponential cumulative distribution function proposed in Cammalleri et al. (2016a) was used to estimate the probability of non-exceedance p of a certain cumulative streamflow deficit:

$$p(S_i; \lambda) = 1 - e^{-\lambda S_i} \qquad \text{(with } S_i > 0\text{)} \qquad (3)$$

where the variable $S_i$ is the severity of drought event i, as quantified by a cumulative indicator, and the parameter λ is the inverse of the mean of the severities of all completed drought events. For instance, a value of p=0.7 in March 2002 de-

notes that, if the drought event ended in March 2002, its severity would be larger than the severity of 70% of the drought events in the reference period. Different from LFI, which is based on daily streamflow data, time series of monthly streamflow were used for all indicators and the two months criterion (see Sect. 2.2.3) was applied. Since p was computed for each month of the reference period, it describes the non-exceedance probability of both completed drought events and continuing droughts.

## 3 Results and discussion

### 3.1 Proposed systematic approach for selecting and computing streamflow drought hazard indicators

#### 3.1.1 Assumptions about habituation inherent in drought hazard indicators

The choice of drought hazard indicators implies assumptions about the habituation of the system at risk. In the case of streamflow, people and ecosystems are assumed to have adapted to certain characteristics of the flow regime. For example, if drought indicators are computed based on the calendar month-specific distribution of streamflow values, it is implicitly assumed that people and ecosystems are adapted to the seasonality of streamflow. In case of SSI1, for example, it is implicitly assumed that people and ecosystems are, in addition, adapted to a certain degree of interannual variability, e.g., to the low streamflow that is only exceeded in 1 out of 5 years. But also temporally constant thresholds, which have traditionally been used to define hydrological droughts (Stahl et al., 2020), are suitable for certain systems, e.g. for computing drought risk for electricity generation by thermal power plants, which require a certain minimum streamflow for operation.

When conceptualizing or selecting a hazard indicator for a specific drought risk, these assumptions on habituation of the system at risk should be made explicit. At the global scale, it is unknown to which streamflow characteristics different risk systems such as drinking water supply, irrigation water supply, hydropower production and the river ecosystem are accustomed. Therefore, the eleven global-scale drought hazard indicators analyzed in this study cover different types of habituation, including the habituation to a certain degree of interannual variability of streamflow, to streamflow seasonality, to a certain reduction from mean calendar month or mean annual streamflow, and to being able to fulfill the demand for surface water abstractions and environmental flow. Table 1 lists the indicators according to this classification together with unique characteristics relevant for streamflow drought risk assessments.

**Table 1:** Characteristics of conventional streamflow drought hazard indicators suitable for global-scale assessments, classified according to inherent assumptions about habituation of people or other biota

| **Assumed habituation and indicator**<br>*People or other biota accustomed to* | | **Characteristics** |
|---|---|---|
| a certain degree of interannual variability | SSI12/EP12[1] | Suitable for quantifying 1) risk for human water supply in regions with large man-made reservoirs or lakes that buffer seasonal streamflow deficits as well as 2) risk for large lake and wetland ecosystems. |
| | SPI12 | If used as proxy for streamflow drought hazard, assumptions about habituation are the same as for SSI1.<br>Processes in altered flow regimes cannot be characterized. |
| seasonality and a certain degree of interannual variability | SPEI12 | Same characteristics as SPI12; better proxy for streamflow drought hazard as it takes into account the impact of increased potential evapotranspiration on drought. |
| | SSI1/EP1/CDQI1-Q80 | Suitable risk for human water supply and for risk for river ecosystems in regions without access to reservoirs. Streamflow drought hazard might be underestimated in regions with high vulnerability and interannual variability. |
| | and median calendar month streamflow<br>CDQI1-Q50 | Using such a high threshold (median of calendar monthly streamflow) can be beneficial in highly vulnerable regions where people cannot even cope with small reductions in median monthly streamflow. |
| | being able to fulfill demand for surface water abstractions<br>CDQI1-WUs | The system at risk is accustomed to the seasonality of human water demand (WUs). People are used to being able to fulfil human water demand.<br>The health of river ecosystems is not taken into account.<br>An indicator of water deficit rather than drought hazard. |
| seasonality | being able to fulfill demand for surface water abstractions and environmental flow<br>CDQI1-WUs-EFR | The system at risk is accustomed to the seasonality of human water demand (WUs) and to the seasonality of environmental flow requirements (EFR).<br>Alternative 1: EFR based on Qant[2]: The river ecosystem has adjusted to the altered flow regime over the last decades, which is considered the "new normal status".<br>Alternative 2: EFR based on Qnat[2]: the natural flow regime is the aspired status. |
| | a certain reduction from mean calendar month streamflow RDQI1 | Suitable in study regions without large surface water storages.<br>Drought hazard might be overestimated in regions with low vulnerability and interannual variability. |
| a certain reduction from mean annual streamflow | RDQI12 | Suitable in study regions with large man-made reservoirs or lakes, which buffer seasonal streamflow deficits.<br>Drought hazard might be overestimated in regions with low vulnerability and interannual variability. |
| temporally constant minimum streamflow | Not included in this study | Identifies drought hazard whenever water availability drops beneath a certain level (e.g., water intake for cooling of thermal power plants has to be reduced). Identifies no drought in wet season. |

[1] EP12: Empirical streamflow percentile with an averaging period of 12 months (not analyzed in this study)

[1] Qant, Qnat: Modeled anthropogenic streamflow altered by human water use and man-made reservoirs (Qant) and naturalized modeled streamflow (Qnat)

In hydrology, flow duration curves showing the fraction of the time that a certain streamflow is exceeded are a widely used method to assess the low-flow regime (Smakhtin, 2001). For example, the Q90 is the low streamflow that is exceeded in 90% of the time and is equivalent to the 10th percentile of the cumulative distribution function. Percentile-based indicators including empirical streamflow percentiles, standardized indicators and TLM indicators with a low streamflow percentile as threshold) are often applied in DEWS (Bachmair et al., 2016; Cammalleri et al., 2016a). They are perceived as statistically consistent across different temporal and spatial scales, indicating the rarity of the event (Steinemann et al., 2015; WMO and GWP, 2016). Indicators of less than normal water availability such as "percent of normal precipitation" appear to be less preferred as time periods with the same indicator value have different probabilities of occurrence in different regions and thus not the same rarity (Steinemann et al., 2015). However, according to Kumar et al. (2009), percent deviations from mean precipitation (or percent of normal = percent deviation + 100%) have been used to assess drought intensity in India, South Africa and Poland. Kumar et al. (2009) compared percent precipitation deviations and SPI in two districts in India with high and low precipitation. Based on a 39-year record of observed monthly precipitation they showed that during the monsoon months where precipitation is decisive for crop production, much higher percent precipitation deviations occurred in the low precipitation district than in the high precipitation district, e.g-70% and -30%, respectively, in case of SPI = -1. They found that due to the need to fit a function to the actual precipitation data to determine SPI and thus probability of occurrence, a year with a lower precipitation might be indicated by the SPI as being less dry than a year with a relatively higher precipitation. Consequently, severe drought may be underestimated with SPI in particular in the low precipitation district due to non-normality of the distribution for extremely low precipitation values. More importantly, considering the risk for rainfed crop production, yield loss is more closely related to percent of normal precipitation than to the rarity of the low precipitation event, as crop yield depends on actual evapotranspiration in percent of PET, which decreases with precipitation (Siebert and Döll, 2010). Yield loss due to 30% less precipitation than normal can be expected to be much smaller than yield loss due to 70% less, such that percent deviation from the mean can be a good hazard indicator for assessing drought risk for rainfed crop production in these two districts. To quantify risk, this type of hazard indicator could be combined with an indicator of exposure such as growing area and an indicator of social vulnerability such as farmer income.

Application of percentile-based indicators (e.g., SSI12, SSI1, and CDQI1-Q80 in Table 1) implies that people in different climate regions and social systems are equally habituated to a certain interannual variability, which is most likely not the case. Comparing a humid region with low interannual streamflow variability to a semi-arid region characterized by high interannual variability, the same streamflow percentile (e.g. Q80) would correspond to a much stronger negative deviation from mean streamflow in the semi-arid area (e.g. -50%) compared to the humid area (e.g. -20%). Hence, although these indicators have the advantage of being spatially comparable in terms of drought frequency, they might underestimate streamflow drought hazard in semi-arid areas where people (and ecosystems, albeit possibly to a lower degree) are often more vulnerable to reductions in water availability and not necessarily adapted to the high interannual variability of water availability. Regions with high interannual variability are depicted in Fig. A1b. Here, drought hazard indicators that quantify anomalies from the

long-term mean or median might be better suited to define drought conditions. These include percent deviations from mean streamflow (RDQI1, RDQI12 in Table 1) or TLM indicators with higher percentiles as threshold (CDQI1-Q50 in Table 1). Contrastingly, river ecosystems are, in the ideal case, perfectly adjusted to interannual variability of streamflow such that percentile-based drought hazard indicators are often suitable for drought risk assessment for river ecosystems. Therefore, percentile-based indicators and relative deviations from the long-term mean or median should be used complementarily in large-scale assessments to adequately support the assessment of different drought risks.

Another important characteristic of drought hazard indicators is the selected averaging period that defines whether people are habituated to the annual or seasonal flow regime. One can assume that river ecosystems are generally accustomed to seasonality. Therefore, indicators with a short averaging period of, for example, one month (SSI1, RDQI1 and CDQI1 variants in Table 1) are appropriate for quantifying drought hazard for river ecosystems. Furthermore, short averaging periods are suitable in regions where farmers and other water users do not have access to large water storages such as reservoirs, lakes, or groundwater (either due to missing infrastructure or due to water use restrictions) and who need to use the water as it flows down the river. Hence, these users are very vulnerable to seasonal (monthly) streamflow deficits. Indicators with longer averaging periods (SSI12 and RDQI12) on the other hand are suitable in regions with large man-made reservoirs, which are usually replenished during the wet season such that streamflow deficits during the low-flow months are irrelevant. People in these regions are therefore only vulnerable to either interannual variability (SSI12) or mean annual conditions (RDQI12), but not to seasonality. Certainly, other averaging periods may be suitable depending on the region-specific storage capacity.

Since volume-based indicators (TLM indicators) are also important components in water resources management (van Loon, 2015), we propose the volume-based indicator CDQI1-Q80-HS as an alternative for SSI12 and RDQI12 to quantify drought hazard when assessing drought risk for human water supply in case of highly seasonal (HS) streamflow regimes and large reservoirs. If water users need streamflow to fill a reservoir, streamflow availability during the dry season would be of (almost) no interest to the risk takers/water users. For them, it would be worse to have less water than normal during two consecutive wet seasons even if there is slightly more water than normal in the dry season (as the amount of this water is very small compared to the water produced in the wet season). With CDQI1-Q80-HS, an existing drought is allowed to continue during a pre-defined low-flow period, i.e., months where the calendar month Q80 is zero, even if streamflow exceeds zero (Sect. 2.2.3). Consequently, in case of two consecutive wet-season droughts the streamflow deficit continues to accumulate during the second wet season, resulting in a higher drought severity for the drought event (after the first wet season) than for the two wet-season droughts. This most likely reflects the perceived hazard of such a situation better. Nevertheless, this definition is only fulfilled in a limited amount of grid cells and months such that CDQI1-Q80 and CDQI-Q80-HS are very similar (Sect. 3.3.2 and Fig. 5). Therefore, the same assumptions about habituation to the streamflow regime apply for both CDQI1 variants, namely interannual variability and seasonality. Regions where the application of CDQI-Q80-HS is meaningful, i.e., regions with highly seasonal streamflow, are depicted in Fig. A1a.

When SPI12 and SPEI12 are used as proxies to identify streamflow drought hazard, they should ideally correspond to the temporal development of SSI1. Hence, assumptions about the habituation inherent in SPI12 and SPEI12 described in

Table 1 refer to the streamflow regime and not to the meteorological variables P and PET. Accordingly, SPI12 and SPEI12 fall into the same category as SSI1 in Table 1 (interannual variability and seasonality). Obviously, if SPI12 and SPEI12 are used to assess meteorological drought hazard, people and ecosystems are assumed to be habituated to the interannual variability, but not the seasonality, of P and P-PET. The suitability of different averaging periods for SPI for describing streamflow drought hazard is discussed in Sect. 3.5.

For water managers, the status of the actual water deficit in terms of unsatisfied water demand might be as informative as the status of streamflow anomaly. Drought hazard is generally defined as a climate-induced anomaly, i.e., a period of below-normal water availability (McKee et al., 1993; van Lanen, 2006; van Loon, 2015). This concept can be broadened by assuming that a drought only occurs if the anomaly coincides with a water deficit for people or ecosystems (Cammalleri et al., 2016b; Popat and Döll, 2021). This concept is not new and several definitions were already summarized in Wilhite and Glantz (1985), e.g., drought is a "period during which streamflows are inadequate to supply established uses under a given water management system" (Linsley et al., 1975 in Wilhite and Glantz, 1985: 115). Nevertheless, only a few studies exist where the combination of anomaly and deficit was translated into drought hazard indicators for soil moisture (Palmer, 1965; Cammalleri et al., 2016b; Popat and Döll, 2021) and streamflow (Popat and Döll, 2021). In the present study, the water deficit aspect of drought is represented by the indicators CDQI1-WUs and CDQI1-WUs-EFR where surface water demand is taken as the threshold (Table 1). Application of these indicators implies that the system at risk is habituated to the satisfaction of seasonal water demand. While CDQI1-WUs neglects water requirements of the ecosystem, CDQI1-WUs-EFR as computed in this study assumes that the river ecosystems is habituated to the seasonality and magnitude of natural streamflow. As EFR might never be fulfilled during the investigation period in case of streamflow regimes that are strongly altered by human water abstractions and man-made reservoirs, Qnat in the EFR computation can be replaced by Qant in case of strongly altered streamflow regimes. This implies the assumption that the river ecosystem has already adapted to the altered streamflow conditions (Table 1). Figure A1c shows regions where human water demand is high compared to available streamflow and where a drought hazard due to unsatisfied human surface water demand is likely.

### 3.1.2 Levels of drought characterization

Wilhite and Glantz (1985) suggested to distinguish between a conceptual and an operational drought definition, with the former referring to the general qualitative concept of drought and the latter allowing for a quantitative drought characterization including onset, severity, termination, and spatial extent. Accordingly, the qualitative characterization of the system at risk, the targeted drought risk and the associated assumptions about habituation, as discussed in the previous section, constitute the conceptual drought definition. Translating this definition into a quantitative drought hazard indicator is not straightforward due to the complexity of the underlying natural processes and the large number of methods and indicators that can be applied.

In the existing literature, there is agreement about which drought characteristics are relevant for operational applications, namely the temporal component (onset, termination, duration) and the spatial extent as well as drought magnitude and severi-

ty, from which other metrics such as intensity, return period, and frequency or probability of occurrence can be derived (van Lanen et al., 2017). We understand drought *magnitude* as an anomaly or deficit occurring within one time step and *severity* as the accumulated anomaly or deficit over all time steps during the duration of the drought event exceeding a selected threshold (van Lanen et al., 2017). However, the terms drought magnitude and severity, which represent different levels of drought characterization, are not applied consistently in the literature. The terms are not made explicit and sometimes interchanged (Steinemann et al., 2015, Vidal et al., 2009; López-Moreno et al., 2009). In particular, the commonly accepted classification of streamflow drought hazard indicators into threshold-based and standardized indicators (van Loon, 2015) can be somewhat misleading, since the former represents time series of severity and the latter time series of magnitude.

To facilitate a better understanding of the informative value of streamflow drought hazard indicators, we suggest a new indicator classification that includes four types of indicators and distinguishes severity from magnitude indicators (Fig. 1). The four indicator types (columns in Fig. 1) quantify the volume-based anomaly, the standardized or percentile-based anomaly, the relative deviation, and the anomaly combined with the deficit. Anomaly indicators quantify how unusual the water availability deficit with respect to a threshold is. Relative deviation indicators do not indicate how unusual a deficit is but directly show the deficit, i.e., how much less water is available than under mean conditions. Deficit-anomaly indicators combine an anomaly indicator with an indicator of the deficit with respect to optimal water availability. For example, Popat and Döll (2021) combined the volume-based magnitude indicator $p_Q$ (Fig. 1) with an indicator of the streamflow deficit with respect to water demand to obtain QDAI. For each indicator type, two levels of drought characterization can be computed: Level 1 indicates the drought magnitude, i.e., the non-cumulative anomaly or deficit at each time step. Time steps in drought analysis are usually months, but daily time steps may be used in drought monitoring systems (Cammalleri et al., 2016a). Time series of drought magnitude can be expressed as absolute (volume-based) or relative anomaly or in terms of frequency or probability of occurrence. If magnitude indicators are cumulated since drought onset, severity indicators are obtained at level 2. The units of the four indicator types differ both at level 1 and 2, but for all four indicator types, severity of the drought event can be expressed in the same units of probability of non-exceedance (Fig. 1).

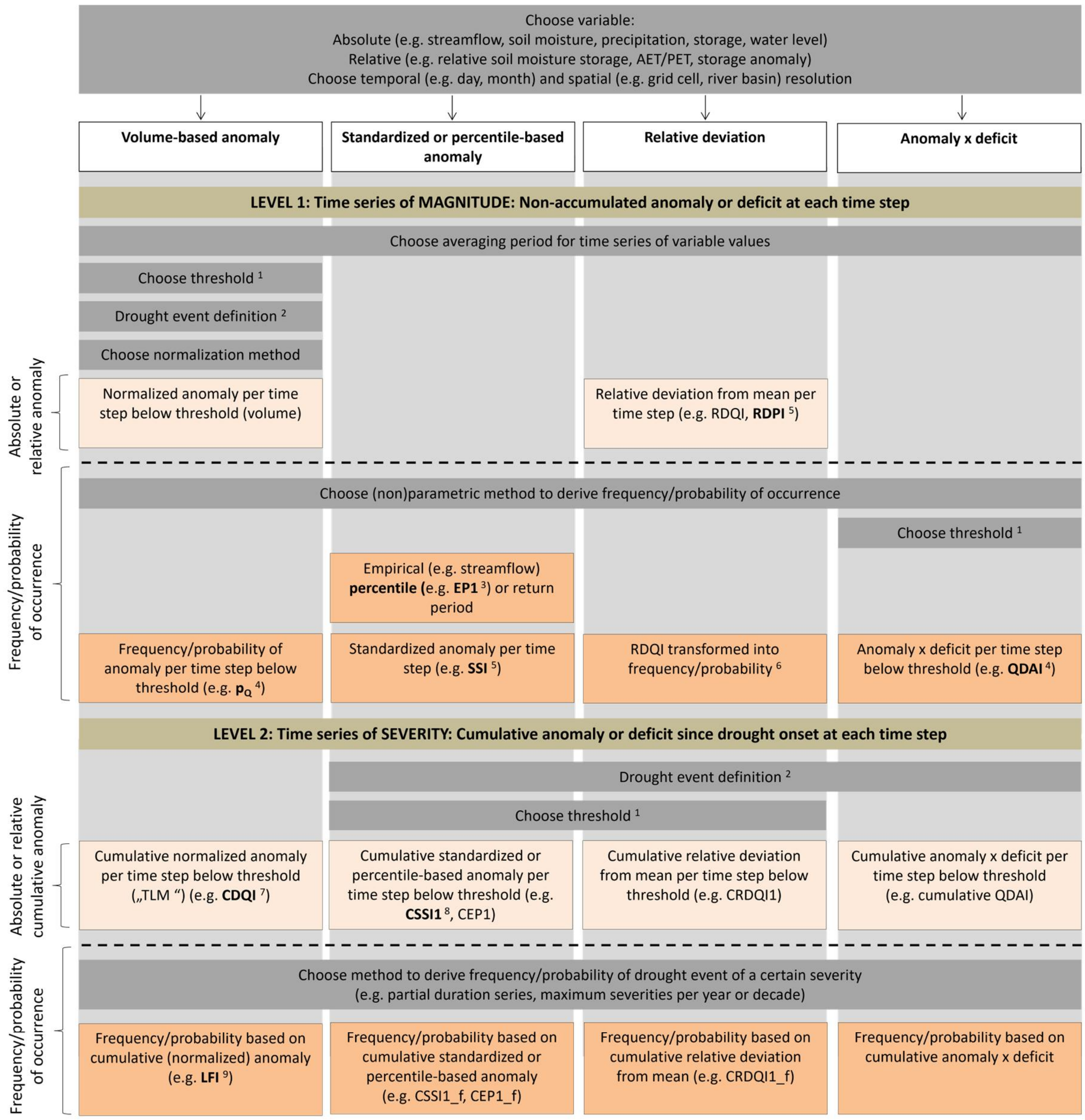

[1] Threshold to define anomaly or deficit based on, e.g. the same variable or a type of (human, plant, ecosystem) water demand and defined by, e.g. constant or seasonal values, percentiles or mean, temporally averaged over the averaging period of the variable or over a different time period (e.g. 31-day running mean of daily values in case of a daily averaging period)
[2] Methods to handle periods of no or low flow or storage; definitions for onset, termination, pooling of drought events
[3] e.g. Tijdeman et al. (2020)
[4] Döll and Popat (2021) (QDAI: streamflow deficit anomaly index; $p_Q$: streamflow drought probability index)
[5] e.g. Modarres (2007)
[6] See Quiring (2009) and Steinemann et al. (2015) and their percentile-based approach for „objective drought definition". Only the percent normal precipitation (relative deviation from mean precipitation, RDPI) was assessed.
[7] e.g. Fleig et al. (2006)
[8] e.g. Barker et al. (2019)
[9] Cammalleri et al. (2016)

**Figure 1:** Schematic for computing four types of drought hazard indicators, indicating 1) magnitude of the drought at a certain time step as deficit and/or anomaly (level 1) or 2) severity of the drought event, i.e. the cumulative magnitude of drought

since drought onset (level 2). Both magnitude and severity can be expressed in terms of frequency/probability to compare the drought of interest to other droughts. The dark grey boxes indicate decisions made when computing the indicators. Indicators in bold have already been applied in the literature. Assumptions about the habituation of people and ecosystems determine the selection of the type of indicator, the averaging period, and the threshold (see Table 1).

In addition to the classification scheme, Figure 1 shows, in the dark grey boxes, the decisions to be made before a drought hazard indicator can be computed, regarding time step length and averaging period, drought threshold and definition of drought events (minimum length of drought event, pooling of drought events). Specific drought hazard indicators are shown in beige and orange boxes. Beige boxes contain indicators that are expressed in absolute or relative values, while orange boxes show indicators that are expressed in terms of frequency/probability of occurrence. Indicators that are currently used in drought monitoring (CDQI1, LFI, percentiles, SSI, RDPI) or that have been applied in the literature ($p_Q$, cumulative SSI, QDAI) are written in bold.

The combined choice of indicator type, averaging period and threshold implies assumptions about the habituation of people and ecosystems to certain streamflow conditions (Table 1). For example, the standardized streamflow drought hazard indicator SSI assumes a habituation to interannual variability due to the division by the standard deviation, while in case of volume-based anomaly indicators, the assumed habituation informs the choice of threshold. Assuming that people or ecosystems are used to interannual variability of water availability is represented by selecting a statistical low flow such as Q80 (that is exceeded in 8 out of 10 averaging periods, CDQI1-Q80 in Table 1), while the assumption that the risk system is habituated to mean/median conditions is expressed by choosing the mean or the median as threshold (CDQI1-Q50 in Table 1). Finally, selecting water demand as threshold implies that people are used to being able to fulfil human and ecosystem water demand (CDQI1-WUs). This threshold selection turns the volume-based anomaly indicator into a deficit indicator. An averaging period of 1 month implies habituation to the seasonal variations of water availability, as water availability in the month of interest is compared to a calendar month-specific threshold (SSI1, RDQI1 and all CDQI1 indicators in Table 1). An averaging period of 12 months implies habituation to mean annual conditions (RDQI12 in Table 1).

The schematic in Figure 1 shows that the specific drought hazard indicators represent different levels of drought characterization (magnitude and severity) and that those pertaining to one of the four indicator types can be transformed between level 1 (magnitude) and level 2 (severity) while still sharing the type-specific conceptual drought definition. Furthermore, the schematic clarifies that each indicator type requires a threshold setting, for defining drought events and for quantifying time series of drought severity, either at level 1 or 2. Hence, the term "threshold-based" applies to any indicator of drought severity and it is therefore not a suitable criterion for distinguishing types of indicators.

Volume-based and standardized or percentile-based anomaly indicators (columns 1 and 2) are based on the same conceptual drought definition if equivalent thresholds are applied. If Q80 is used as threshold for CDQI1 and -0.84 for cumulative SSI (or the 20$^{th}$ percentile for cumulative EP1), both indicators capture the same drought signal, i.e., the 20% lowest streamflow values per calendar month corresponding to a return period of 5 years. Differences between the drought signals

are then attributable to the computational methods for the standardization of streamflow, e.g., which distribution function is selected to compute SSI. Analyzing the sensitivity of SSI1 to different parametric and nonparametric standardization methods in European river basins, Tijdeman et al. (2020) revealed considerable differences in computed SSI1 among seven probability distributions (and two fitting methods) and five non-parametric methods. They argue that there is a conceptual difference between SSI and empirical percentiles (both in the third column in Fig. 1). While SSI indicates the non-exceedance probability enabling extrapolation, empirical percentiles represent the historical non-exceedance frequency within the boundaries of streamflow data. We account for this aspect in Fig. 1 by including both terms (frequency and probability). An advantage of volume-based over standardized indicators is that drought severity can be expressed in volume of "missing" water, i.e. absolute rather than relative values, which is often more informative in water resources management (van Loon, 2015). However, the relative levels of drought severity among the drought events during the reference period differ as volume-based indicators detect absolute and standardized or percentile-based indicators relative drought deficits. For instance, a monthly deficit volume of 1% of mean annual streamflow represents a larger deviation from median streamflow in a low-flow month compared to a high-flow month. Consequently, differences between volume-based and standardized or percentile-based indicators can be large when severities of drought events during the reference period are compared. This difference is illustrated in Sect. 3.3.4 and Fig. 7.

### 3.2 Observation-based streamflow drought hazard indicators

The relation between ten out of the eleven indicators in Table 1 was assessed for time series of monthly streamflow observed at two GRDC gauging stations with different streamflow regimes. CDQI1-Q50, suitable in highly vulnerable regions, was not considered since both selected stations are situated in river basins with an assumed low vulnerability to drought by global comparison. We used observations instead of WaterGAP modelling result to exclude model uncertainties. Only mean monthly WUs used in CDQI1-WUs and CDQI1-WUs-EFR is based on WaterGAP model output. Different from the description in Sect. 2.2.6, EFR in CDQI1-WUs-EFR is computed as 80% of observed mean monthly Qant and not Qnat.

The Little Colorado River near Cameron in the United States (Fig. 2, left) is characterized by comparably low mean annual streamflow (MAQ) (ca. 5 $m^3 s^{-1}$) and high interannual and seasonal variability. Mean monthly streamflow (MMQ) (Fig. 2a) is lowest between May and June as well as November and December (< 2 $m^3 s^{-1}$). Q80 is zero in May, June, and November. Using Q80 as threshold, streamflow deficits are highlighted in orange in Fig. 2a. Fig. 2b depicts time series of drought severity according to different CDQI1-Q80 variants. First, the effect of the two months criterion (2mc) (Sect. 2.2.3) can be deducted by comparing CDQI1-Q80 (w/o 2mc) and CDQI1-Q80. Six one-month droughts at the Little Colorado River are excluded (e.g. 1987, 1989, 2000) and two drought events in 1996 are pooled into one 8-month drought if the 2mc is applied (minimum length of drought is two month, an drought stops if afterwards there are two months above the threshold). This probably better reflects the perceived severity of the event in 1996 where in the single month between deficits, people and the ecosystem cannot recover as streamflow only slightly exceeds Q80. The Danube River at Hofkirchen in Germany (Fig. 2,

right) is characterized by a much higher MAQ of 640 $m^3 s^{-1}$ and lower seasonal and interannual variability. Here, the 2mc leads to the realistic extension of the drought event in 2003 until the end of the year and the non-consideration of several short one-month droughts between 2004 and 2011.

The HS method (Sect. 2.2.3), suitable in highly seasonal flow regimes, allows an existing drought to continue during calendar months with Q80=0 even if streamflow exceeds zero. Comparing CDQI1-Q80-HS and CDQI-Q80, this method can either lead to the mere prolongation of drought events (for example in 1990 and 1991, Fig. 2b, left) (case 1) or to the pooling of two or more wet-season droughts into one drought event (case 2, not identified at the two stations). When computing the frequency distribution of drought severity, there would be no difference between CDQI1-Q80 and CDQI1-Q80-HS in case 1. In contrast, the pooling of two or more wet-season droughts into one drought event (case 2) does change the frequency distribution of drought severity. Among 220 GRDC gauging stations worldwide with continuous monthly streamflow observations between 1986 and 2015, only three stations include calendar months with Q80=0. The selected station in Fig. 2 (left) is the only station with a visible impact of the HS method. Differences between CDQI1-Q80 and CDQI1-Q80-HS are larger for simulated streamflow at the global scale (Sect. 3.3.2 and Fig. 5).

For better comparison with the CDQI1-Q80 variants, only z-scores below -0.84, equivalent to Q80, are shown for the standardized indicators (Fig. 2c). Since fitting of the gamma distribution was rejected for the Little Colorado River station based on the KS test (Sect. 2.2.2), the indicator EP1 (Sect. 2.2.4) was computed instead of SSI1 and transformed into z scores. Both EP1 and SSI1 capture the same drought signal as CDQI1-Q80 (w/o 2mc) at both stations. Nevertheless, the former indicate the drought magnitude during the month of interest only, while CDQI1-Q80 indicates the cumulative magnitude, i.e., severity since the beginning of the drought event. For instance, drought severity of the drought event in 2014 (Fig. 2b, right) exceeds the value in 2011 by a factor of almost 2. The maximum drought magnitude for both events, however, is very similar according to SSI1. Cammalleri et al. (2016a: 356) aptly write that standardized indicators such as SPI and SSI cannot reproduce the "conceptual mechanism behind the evolution of a drought event as a phenomenon that is derived from a continuous hydrological quantity with daily values that are strongly dependent on the antecedent status". We argue though that this is just due to the fact that SSI1, as well as EP1, indicate drought magnitude and could be converted into the severity indicators CSSI1 and CEP1, respectively (see Fig. 1), which take into account the antecedent status of streamflow (Fig. 5f). With SSI12, deviations from normal conditions are smoothed over the preceding 12 months making the indicator suitable in highly seasonal flow regimes with reservoirs. The indicator is not conceptualized to detect streamflow anomalies at a monthly scale and to identify the onset of a streamflow drought. Streamflow anomalies in 2003 and 2014 (Fig. 2c, right), for example, are indicated by SSI12 only shortly before the drought is over. It is rather an indicator of reservoir drought hazard with the ability to detect the onset of a reservoir drought. Furthermore, the drought event in 1996 (Fig. 2c, left) is terminated in 1997 according to SSI12 when streamflow significantly exceeds Q80, which would probably fit better to the drought hazard as perceived by water users depending on reservoir storage.

The correspondence between the meteorological indicators (SPI12 and SPEI12) and the hydrological indicators (SSI1 or EP1 and CDQI1-Q80 variants) is low at both stations (Fig. 2c). For most streamflow drought events, the averaging period of

12 months for the meteorological variables leads to excessive delays in the signal. Many short drought signals are not detected at all. Performance of SPI12 and SPEI12 is equally low at both stations. As a limited sensitivity analysis, SPI time series with averaging periods of 3 to 12 months were correlated with observed SSI1 at 218 gauging stations (Sect. 3.5). At both stations from Fig. 2, the correlation was highest for SPI3.

RDQI (Fig. 2d) indicates the magnitude of streamflow drought hazard under the assumption that the system at risk is habituated to mean monthly streamflow but not to interannual variability. Due to the high interannual variability at the Little Colorado River with a few high-flow years that considerably increase mean monthly streamflow, RDQI1 and RDQI12 are often below -50% (RDQI1 in 60% and RDQI12 in 30% of months during the reference period). At the Danube station, the threshold -50% is only reached twice during the drought years 2003 and 2014. At this station, RDQI1 values roughly correlate with drought signals according to SSI1, while at the Little Colorado River, the strong RDQI1 signal (due to very high interannual variability) is very different from the EP1 (z score) time series with very few anomalies below -0.84. Comparing the indicators between the two stations, extreme drought events with SSI1 < -1.65 correspond to RDQI1 of -50% at the Danube station (lower interannual variability). In contrast, EP1 (z score) values at the Little Colorado River (high interannual variability) exceed -1.65 only once in the depicted period in September 1989. Comparing both indicators at the Little Colorado River, the minimum RDQI1 values below -90% correspond to EP1 values between -1.8 and +0.6. As discussed in Sect. 3.1.1, it is unknown at the global scale to which streamflow characteristics people and other biota are accustomed to, but Fig. 2 visualizes that SSI may underestimate the drought hazard in semi-arid regions. At the same time, RDQI probably overestimates drought hazard in regions where people are well accustomed to the interannual variability of streamflow.

The water demand deficit indicators CDQI1-WUs and CDQI1-WUs-EFR (Fig. 2e) result in very different temporal patterns of drought severity as compared to the CDQI1 variants. While streamflow at the Little Colorado River is below Q80 mainly outside the low-flow period (May-June and November-December), mean monthly WUs are highest in May and June, and consequently CDQI1-WU droughts often occur in these months. The absolute values of CDQI1-WUs (maximum of 0.0002 units of MAQ, Fig. 2e, left) are well below the CDQI1-WUs-EFR range. Drought severity according to CDQI1-WUs-EFR is significantly higher and drought duration is much longer. EFR in Fig. 2e is computed as 80% of mean monthly observed Q. Hence, it is assumed that the river ecosystem is adapted to the seasonality of streamflow, but it is negatively affected in years with very dry streamflow. At the Little Colorado River, water deficits occur in 65% of all months during the depicted period and mainly stem from unsatisfied environmental flow requirements. Application of CDQI1-WUs alone is not suitable to assess the current status of water deficit, as it does not consider the environmental component of water demand, but the indicator can be used complementarily to show the impact of human water demand on the total water deficit. At the Danube station, CDQI1-WUs is always zero, since WUs is only a small fraction of streamflow. Regions where human water demand is high as compared to supply include, e.g., the Mediterranean region, large parts of Turkey, India, and the western United States (Fig. A1c). Here, drought defined as water deficit due to high water demand is likely to occur. In these regions, CDQI-WUs and CDQI-WUs-EFR can indicate those months where human water use would have to decrease to alleviate drought burden on the river ecosystem.

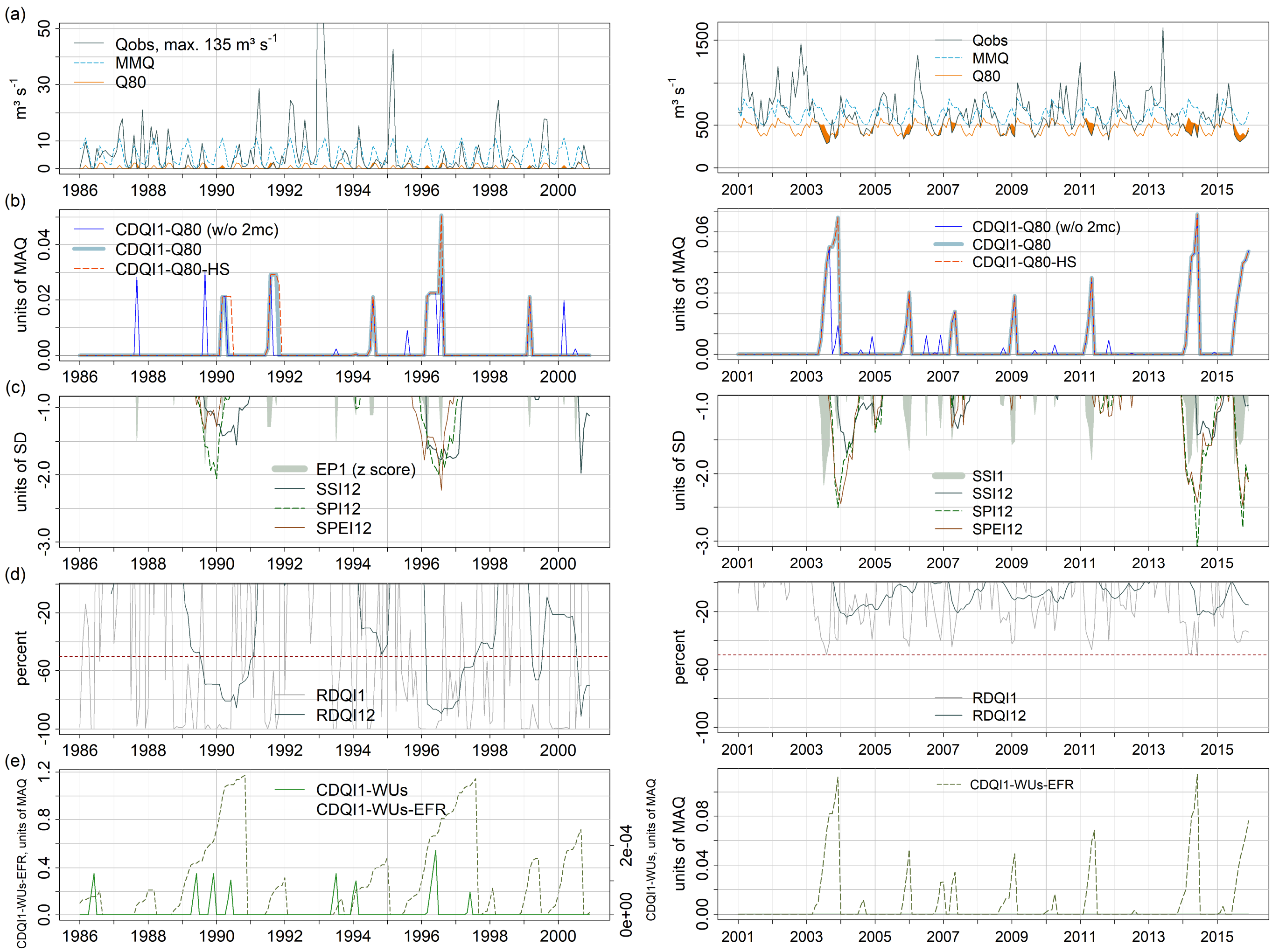

**Figure 2:** Streamflow drought hazard based on observed streamflow during 1986-2015 at two WaterGAP calibration stations in the USA (Little Colorado River near Cameron, 1986-2000) (left) and Germany (Danube River, Hofkirchen, 2001-2015) (right): Monthly observed streamflow Qobs, mean monthly streamflow MMQ and Q80 (a); CDQI1-Q80 variants (b); SPI12, SPEI12, SSI1 or EP1 and SSI12 (c); RDQI1, RDQI12 (d); CDQI1-WUs and CDQI1-WUs-EFR (e). The cumulative indicators in (b) and (e) indicate drought severity; the non-accumulated indicators (c) and (d) indicate drought magnitude (see Fig. 1). In (a), periods where Qobs < Q80 are highlighted in orange. In (c, left), the z score of EP1 is shown instead of SSI1, since gamma fitting was not possible. In (c), only the value range below -0.84 is shown. In (d), only negative relative deviations are depicted. 2mc: 2 months criterion. MAQ: mean annual streamflow. SD: standard deviation.

## 3.3 Simulated streamflow drought hazard indicators

### 3.3.1 Drought magnitude (level 1)

Figure 3 compares indicators of drought magnitude, i.e., the non-cumulative streamflow anomaly during a prescribed averaging period, with averaging periods of one month and twelve months, for March 2002. Comparing SSI1 and RDQI1 (Figs. 3a and b), the patterns are different in many parts of the globe. While RDQI1 identifies most of the drought regions according to SSI1 (e.g., western U.S. and Canada, parts of Brazil, Siberia, India, and China), relative levels of drought magnitude are very different. For instance, streamflow drought hazard in northern Siberia is extreme according to SSI1 (return period of 20 years or higher), but the deviation from mean monthly streamflow according to RDQI1 is comparably moderate (-20-40%). This is due to the low interannual variability of streamflow (Fig. A1b). Moreover, RDQI1 identifies many regions that are not in drought according to SSI1 but suffer from extreme deviations from monthly mean streamflow exceeding -80% in, e.g., southern Africa, Australia, and northeastern China, which are characterized by high interannual streamflow variability. The strong correlation between RDQI1 and the interannual variability can be clearly seen by comparing Fig. 3b and Fig. A1b. Overall, RDQI1 values below -40% are computed for a rather high fraction of grid cells (33% excluding Greenland). Analyzing all March results during the reference period, this fraction varies between 29% and 40%. Figs. 3a and 3b underline that RDQI1 can add value to global-scale assessments by drawing the attention to highly vulnerable regions.

EP1 patterns (Fig. 3c) are very similar to SSI1, since both indicators are based on the same conceptual drought definition. Both indicators generally identify the same drought regions; however, drought classes differ for many grid cells with EP1 indicating both more and less severe droughts within each region. These differences are due to the fitting of the gamma distribution in case of SSI1 and due to the assignment of the maximum rank among tied values within a streamflow sample in case of EP1 (Sect. 2.2.4). Comparing SSI1 with empirical percentiles, Tijdeman et al. (2020) identify several advantages and limitations for both indicators. SSI1 has the disadvantage that for different streamflow regimes, different parametric probability distributions would be required to achieve the best fit. Applying different distributions per grid cell and calendar month, however, reduces consistency among indicator results at the global scale. In this study, the gamma distribution showed the best fit among 23 parametric probability distributions for most grid cells and was applied in each month and grid cell. Of course, using only one distribution for the whole globe results in poorly fitting distributions for some cells and months (Tijdeman et al., 2020) especially at the lower bound. Grid cells where gamma fitting was rejected in March based on the KS test (Sect. 2.2.2) are shown in grey in Figure 3a (18% of all grid cells excluding Greenland).

As an alternative to SSI1, EP1 has the advantage that it does not require fitting of a distribution and can therefore be computed in more grid cells than SSI1. Only if samples, i.e., streamflow values per calendar month, include more zero flows than the selected percentile threshold, it is not possible to identify droughts for this sample. For instance, if the 20$^{th}$ percentile is selected to define drought, a drought event cannot be identified in calendar months with more than six months with zero streamflow (among a sample of 30). For March 2002, these grid cells are highlighted in blue in Figs. 3a and 3c. (In Fig. 3c, these cells all coincide with grid cells where gamma fitting was rejected.) On the other hand, if Q80 is zero and current

streamflow exceeds zero, it is possible to define that the current month is not a drought month (shown in beige in Figs. 3a and 3c). EP1 has the disadvantage that it only allows the quantification of the historical non-exceedance frequency within the reference period, while probabilistic information, for example on extreme events such as a 100-year drought, cannot be derived (Tijdeman et al., 2020).

SPI12 and SPEI12 (Figs. 3c and e) can be used as proxies for identifying streamflow drought hazard if streamflow data is not available. Of course, the correlation with SSI1 strongly depends on the selected averaging period, which varies with different basin characteristics (Sect. 3.5). Here, the selected indicators correlate fairly well with SSI1 by visual inspection; however, the areal extent of extreme drought magnitude below -1.65 is higher according to the proxy indicators (e.g., U.S. east coast, southern Africa, and eastern China). Hence, in a global assessment, different averaging periods for SPI and SPEI should be provided either at the global scale or specific to basins based on a correlation analysis. SSI12 (Fig. 3d) indicates where average streamflow between April 2001 and March 2002 is very low compared to April-to-March periods during 1986-2015. Compared to SSI1, the areal extent of extreme drought magnitude (< -1.65) as identified by SSI12 is larger in, e.g., central Brazil, Morocco, north-eastern China, Siberia, and Greece. If people in these regions need streamflow to fill reservoirs, SSI12 is more suitable than SSI1 to detect the drought hazard. In other parts of the globe, the areal extent of extreme drought magnitude is smaller according to SSI12 (e.g., North America, northern Italy, and southern Africa). Here, SSI1 might detect the onset of a streamflow drought that cannot be captured by SSI12. At the global-scale, it is unknown if people depend directly on streamflow or if they have access to reservoirs. Therefore, a global-scale DEWS should provide hazard indicators for both risk systems. For monitoring drought risk for river ecosystems, short averaging periods are more suitable assuming that the ecosystem is habituated to seasonality.

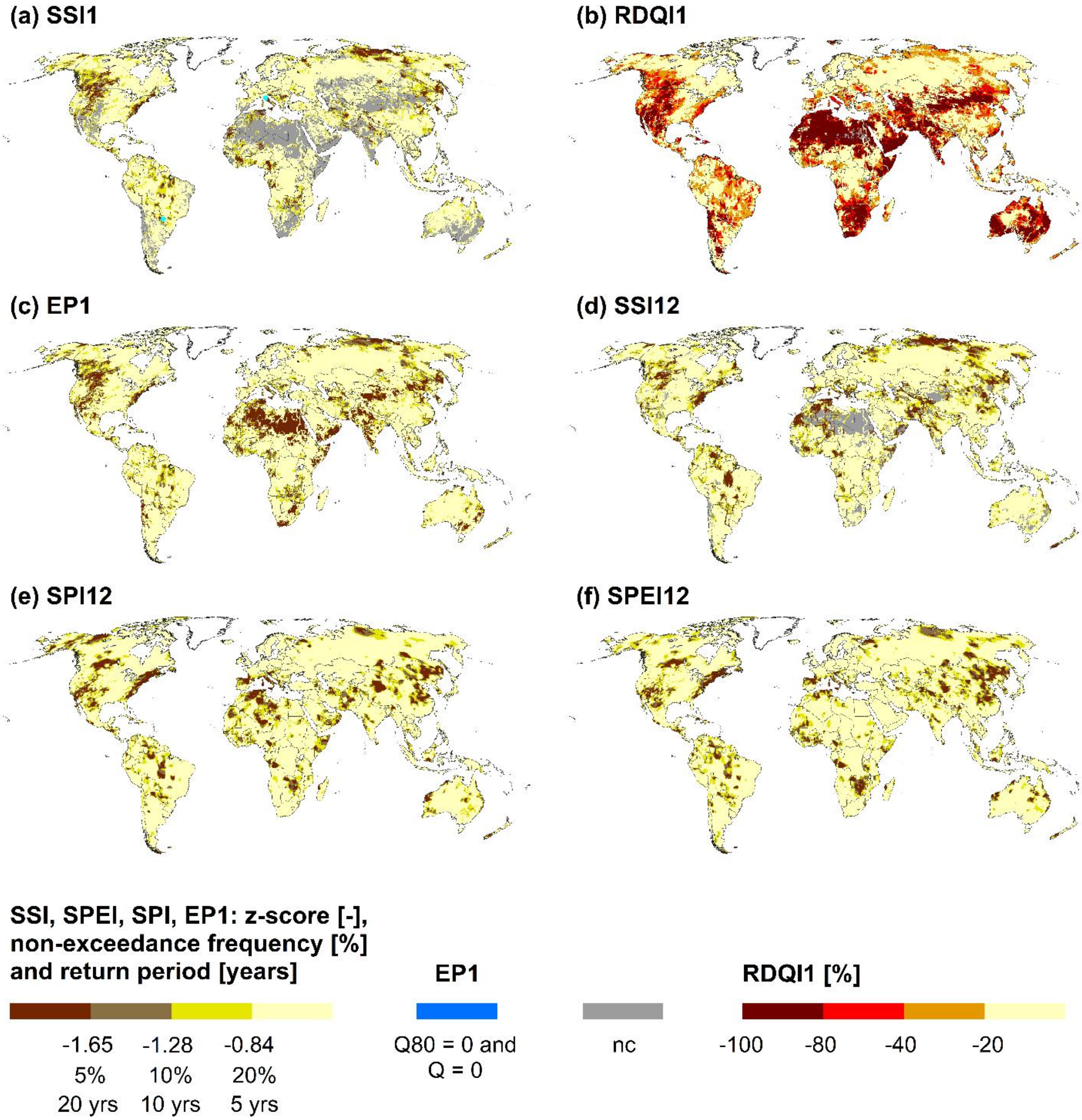

**Figure 3:** Magnitude of drought hazard (level 1 in Fig. 1): Non-cumulative anomaly in March 2002 as indicated by SSI1 (a), RDQI1 (b), EP1 (c), SSI12 (d), SPI12 (e) and SPEI12 (f) for the reference period 1986-2015. For the standardized indicators and EP1, the corresponding return periods are shown. "nc": not computable. The two grid cells from Table 2 are marked in (a) (northern Italy and central Paraguay).

### 3.3.2 Drought severity (level 2)

Figure 4 shows volume-based indicators of drought severity in March 2002, i.e., the cumulative streamflow anomaly (a and c) or deficit (d and e) since the onset of the drought event, expressed in units of mean annual streamflow. Since CDQI1-Q80 (Fig. 4a) is a percentile-based indicator such as SSI1 and EP1, it captures the same drought signal and therefore shows an overall similar spatial pattern as the drought magnitude indicators SSI1 and EP1 (Figs. 3a and 3c). Nevertheless, as the severity indicator includes information on drought development before March 2002 while the magnitude indicators only quantify the drought condition in this month, the severity shows a more differentiated picture of drought conditions in areas with similarly strong drought according to SSI1 and EP1, e.g., Western North America and Northern Siberia. Therefore, drought anomaly indicators and drought severity indicators, either the volume-based version such a CDQU-Q80 or one based on EP1 (CEP1 in Fig. 1), provide different drought hazard information, and can therefore be used complementarily in a DEWS.

A comparison of CDQI1-Q80 and CDQI1-Q80-HS (Fig. 5) reveals that the impact of the HS method is rather small at the global scale but can be relevant at the regional scale. Both indicators allow an existing drought to continue in months where Q80 and the current streamflow are zero. The HS method additionally facilitates drought prolongation in months with Q80=0 if the current streamflow exceeds zero (Sect. 2.2.3). Both indicators have in common that a drought cannot begin in months with Q80=0. Moreover, the two months criterion (2mc), applied for both indicators, allows drought prolongation in case of Q80=0 only if a streamflow deficit was computed in at least two antecedent months with Q80>0. Figure 5a depicts the fraction of drought months as a percentage of all 360 months during the reference period as indicated by CDQI1-Q80. Using Q80 as threshold implies that the time series should be in drought 20% of the time. This is only the case in 6% of all grid cells (excluding Greenland). In 86% of all grid cells, the fraction is reduced to the range of >0% to <20% reflecting that either one-month droughts are ignored (2mc) or that several calendar months with Q80=0 exist where a streamflow deficit can never be identified. The fraction is increased to up to 22% in 5% of all grid cells either due to the pooling of two or more drought events (2mc) or due to drought prolongation in case of Q=0 and Q80=0 (see above). Furthermore, Fig. 5a includes regions where CDQI1-Q80 is always zero (3% of all grid cells). Here, Q80 is zero in nine to twelve calendar months and, in combination with the 2mc, a drought deficit lasting at least two months is never identified. In conclusion, due to the assumed habituation of people and the ecosystem to periods of zero streamflow and to very short streamflow deficits, streamflow drought hazard as quantified by CDQI1-Q80 is less frequent in the grey, green, and beige grid cells (Fig. 5a). Regions where drought occurrence is reduced to less than 14% of the time include Japan, large parts of China, Pakistan, Afghanistan, Iran, North Africa, the western parts of South and North America, and eastern Australia. Application of the HS method leads to an increase in drought months by up to 3 percent points (corresponding to 11 out of 360 months) in 6% of all grid cells (Fig. 5b). Larger increases of up to 12 percent points (43 out of 360 months) are only computed in 0.4% of all grid cells. Higher values for the CDQI1-Q80-HS indicator are computed in parts of India, Pakistan, Afghanistan, Iran, and the western U.S., all of which are regions with highly seasonal streamflow regimes (Fig. A1a) where a drought hazard can be expected to continue even if streamflow in low-flow months (with Q80=0) exceeds zero. Hence, although the HS method has a small effect at the

global scale, differences between CDQI1-Q80 and CDQI1-Q80-HS can be significant in regions with high seasonal streamflow variability.

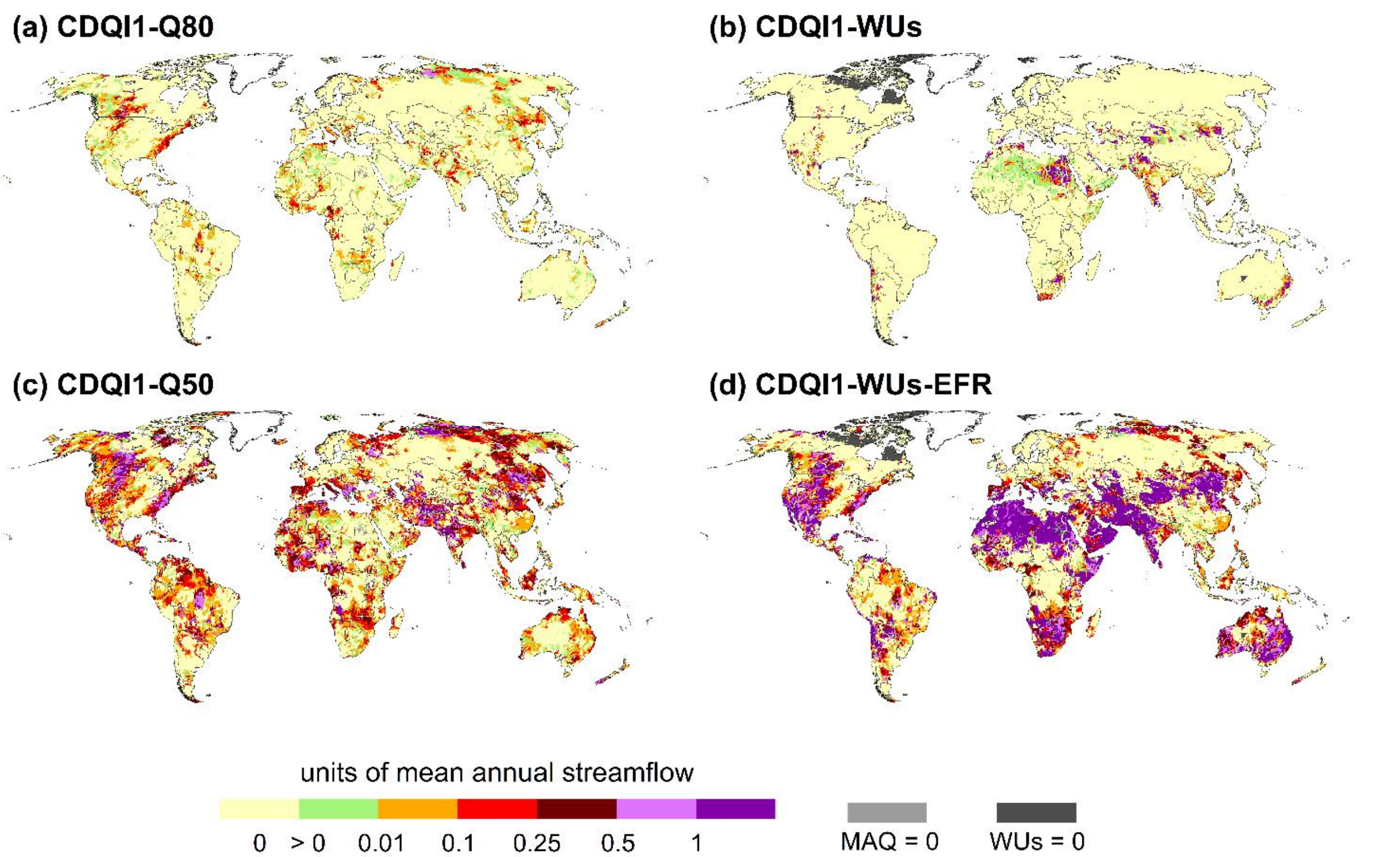

**Figure 4:** Severity of drought hazard (level 2 in Fig. 1): Cumulative deficit in March 2002 since onset of drought event as indicated by CDQI1-Q80 (a), CDQI1-WUs (b), CDQI1-Q50 (c), and CDQI1-WUs-EFR (d) for the reference period 1986-2015. A value of 0.1, for example, denotes that the current cumulative deficit is equivalent to 10% of mean annual streamflow (MAQ). WUs: mean annual surface water withdrawals.

Application of CDQI1-Q50 (Fig. 4c) implies that the system at risk is only habituated to the seasonality of streamflow and that the study region is in drought half of the time. Like RDQI1 (Fig. 3b), the indicator can identify drought in highly vulnerable regions that would otherwise be overseen using lower thresholds such as Q80.

Water deficit in March 2002 according to the water deficit indicators CDQI1-WUs shows a completely different spatial pattern from all the above presented drought hazard indicators (Fig. 4b), since it is driven by both the spatial pattern of water stress (human water demand for surface water as a fraction of mean streamflow, Fig. A1c) and low water availability. While in southern Africa, low water availability leads to high CDQI1-WUs values in March 2002, there seems to be enough water in this month to avoid a water deficit in the equally stressed grid cells in the Central Valley (California, USA) or in Spain.

Natural Hazards and Earth System Sciences
EGU
Discussions

A comparison between the water deficit indicators CDQI1-WUs and CDQI1-WUs-EFR (Fig. 4d) shows that only in a few regions human water demand is the dominant component determining the water deficit in March 2002 (e.g., parts of North America, India, northeastern China, and Australia). In most regions, EFR exceeds human water demand and leads to high cumulative deficits even if seasonal human water demand is small (< 10% of available streamflow, Fig. A1c). Since EFR depends on mean streamflow per calendar month, CDQI1-WUs-EFR shows very similar patterns to RDQI1 (Fig. 3b). CDQI1-WUs-EFR is the only indicator in this study that takes into account ecosystem health, an aspect that should be included in a global-scale DEWS. Alternatively, the cumulative anomaly-deficit indicator QDAI (Popat and Döll, 2021), considering EFR based on a similar approach, can inform decision-makers and water users about the drought hazard for water supply. In strongly altered flow regimes, where simulated anthropogenic monthly streamflow (Qant) is always below 80% of mean monthly naturalized streamflow (Qnat), time series of CDQI1-WUs-EFR are continuously increasing and it is not possible to distinguish drought events. In such cases, it is more meaningful to set EFR to 80% of mean monthly Qant implying that the altered flow regime is the "new normal status" (see also Table 1).

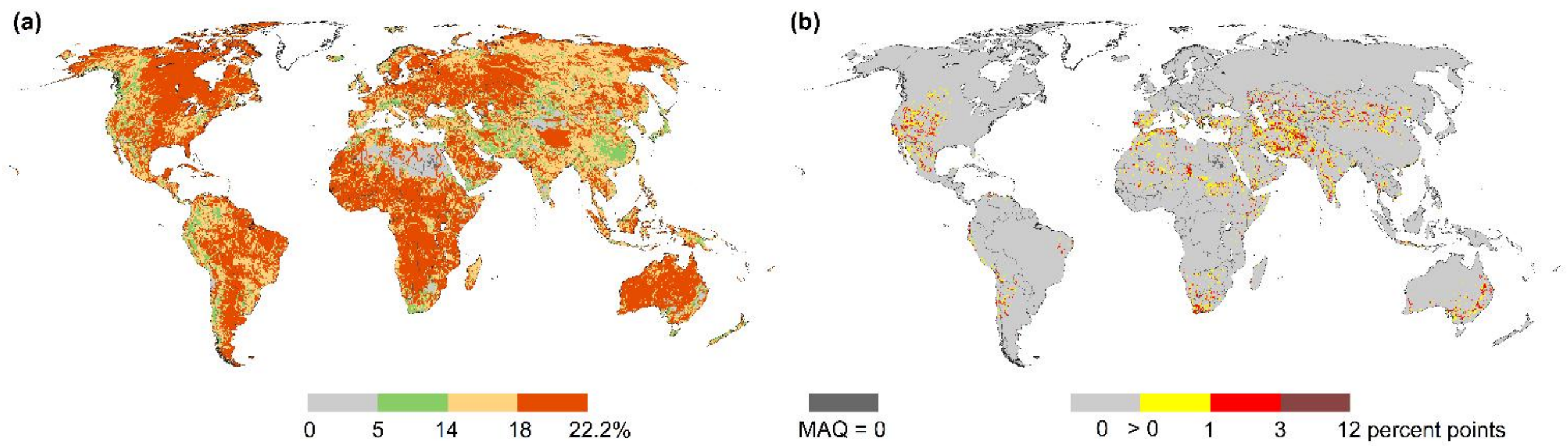

**Figure 5:** Comparison of CDQI1-Q80 and CDQI1-Q80-HS in the reference period 1986-2015: Percent of months in drought based on CDQI1-Q80 (a) and the increase due to the "HS method" in percent points (b). Both indicators allow an existing drought to continue in months where Q80 and the current streamflow are zero. The HS method additionally facilitates drought prolongation in months with Q80=0 if the current streamflow exceeds zero. Neither indicator allows a drought to begin in months with Q80=0. Drought prolongation in case of Q80=0 is only possible if a streamflow deficit was computed in at least two antecedent months with Q80>0 (two months criterion, 2mc, Sect. 2.2.3). In (a), the fraction of drought months is reduced to <20% if one-month droughts are ignored (2mc). In grid cells with 0% in (a), Q80 is either always zero or the few calendar months with Q80>0 result in one-month droughts only. The fraction can be increased to >20% in case of drought pooling (2mc) or in case of drought prolongation if Q80=0. MAQ: mean annual discharge.

**3.3.3 Drought severity expressed as frequency of non-exceedance**

Figure 6 depicts the probability (frequency) of non-exceedance p of drought severity in March 2002 between four CDQI1 variants, the cumulative relative deviation CRDQI1 with a threshold of -50%, and the cumulative empirical percentile CEP1 with a threshold of 20%. A p value of 0.8, for example, indicates a high drought hazard, where the severity up to March 2002 is higher than the severity of 80% of all completed drought events in the reference period. Expressing severity in probability of non-exceedance, as also done in Cammalleri et al. (2016a), facilitates comparison between different indicator types that are quantified with different units; CDQI1 is expressed in volume of mean annual streamflow, CEP1 in percent times month and CRDQI1 in percent times month. Spatial patterns based on CDQI1-Q80_f (Fig. 6a) and CEP1(20%)_f (Fig. 6f) are similar, but differences are visible in several regions. In southeastern Russia, northeastern China, Siberia, and parts of Canada and Alaska, CEP1(20%)_f indicates a more severe drought event than CDQI1-Q80_f, while in the Mediterranean regions and the eastern and southwestern U.S. the severity of the drought events is lower according to CEP1(20%)_f. These differences are due to the fact that CDQI1-Q80 quantifies absolute and CEP1 relative drought deficits resulting in different relative levels of drought severity among the drought events during the reference period (Sect. 3.3.4 and Fig. 7). For CDQI1-WUs_f (Fig. 6b), p values could not be computed in almost half of the grid cells, where less than six drought events were identified such that the map focuses the viewer to grid cells with potential water deficits for human water supply (in particular irrigation). P values based on CDQI1-Q50_f (Fig. 6c) and CDQI1-WUs-EFR_f (Fig. 6d) show high correspondence, as both imply similar assumptions about the habituation of the system at risk to the streamflow regime (see Table 1). Correspondence between these two indicators is higher than between CDQI1-Q50_f and CDQI1-Q80_f. At the global scale, fewer regions with severe drought status are identified in March 2002 based on CRDQI1(-50%)_f (Fig. 6e) compared to CDQI1-Q50_f, but more regions compared to CDQI1-Q80_f.

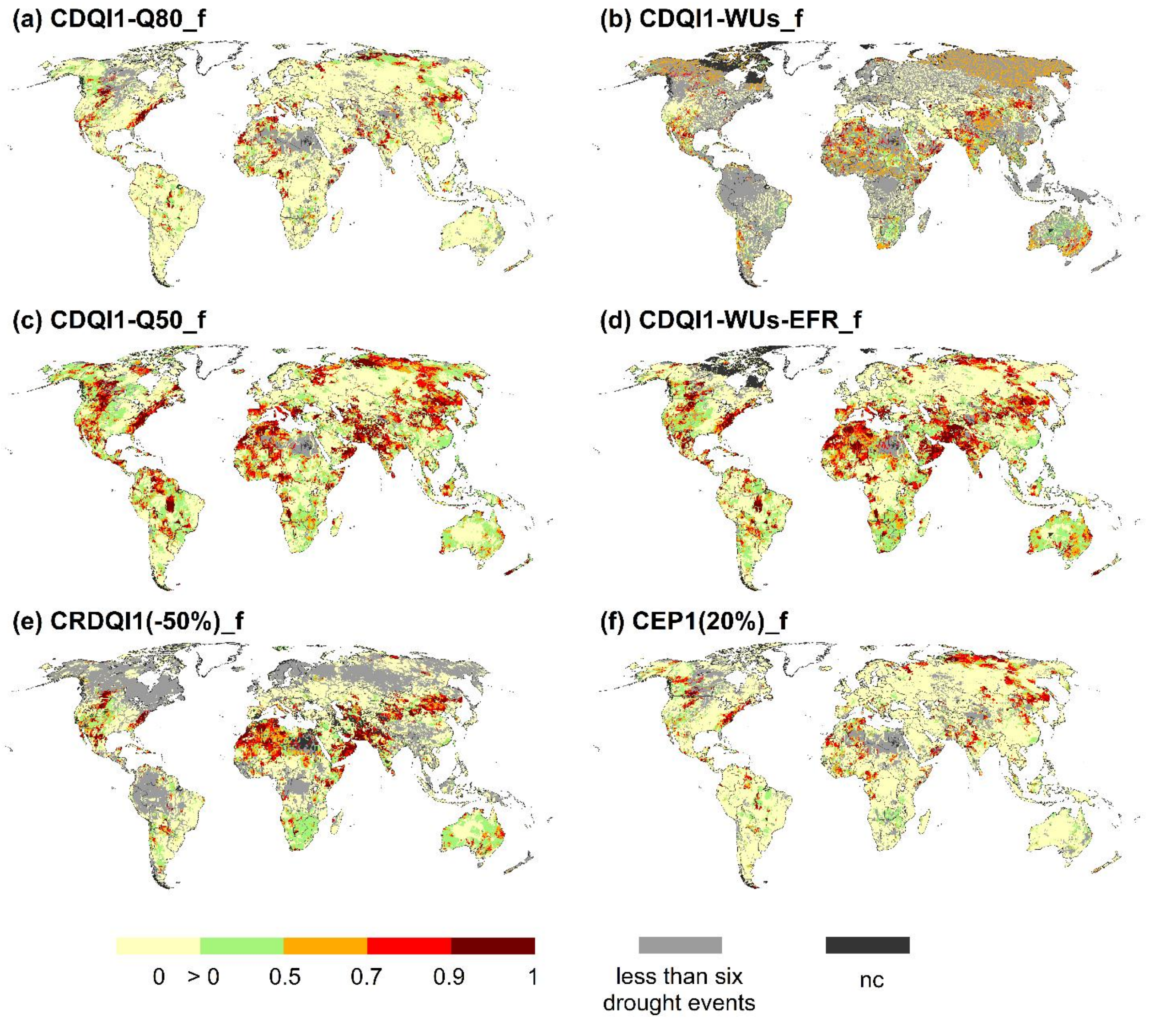

745 **Figure 6:** Probability of non-exceedance of drought events (level 2 in Fig. 1) in March 2002 for the cumulative indicators CDQI1-Q80_f (a), CDQI1-WUs_f (b), CDQI1-Q50_f (c), CDQI1-WUs-EFR_f (d), CRDQI1(-50%)_f (e), and CEP1(20%)_f (f) for the reference period 1986-2015. A value of 0.8, for example, indicates that the cumulative anomaly or deficit, i.e., the severity up to this month, is higher than the severity of 80% of all drought events in the reference period. The probability of non-exceedance was not computed for grid cells shown in light grey, where less than six drought events were computed in the reference period (Sect. 2.2.7). "nc": not computable.
#### 3.3.4 Relation between the various drought hazard indicators

The relation between selected streamflow drought hazard indicators from Figs. 3, 4 and 6 is compared for a grid cell in northern Italy (43.75° N, 11.25° E) and eastern Paraguay (-24.25° N, -56.25° E) in March 2002 (Table 2). Both cells are characterized by low seasonal and high interannual streamflow variability. They suffer from the same drought severity according to CDQI1-Q80, with a water deficit as compared to the threshold Q80 of 4% of mean annual streamflow volume since the start of the drought event. CDQI1-Q80-HS is equal to CDQI1-Q80 in both cells.

Although SSI1 and EP1 are based on the same conceptual drought concept, they indicate different drought anomalies. The streamflow volume in March 2002 is the fourth smallest value in the Italian cell (rank 4, EP1=4/30=0.13) and the second smallest value in the Paraguayan cell (rank 2, EP1=2/30=0.07). However, since the slope at the lower bound of the ranked streamflow values in the Italian cell (not shown) is very small compared to the latter cell, the non-exceedance probability p of the fitted gamma distribution increases equally slowly, and the resulting p (and z score) is smaller than in the Paraguayan cell in March 2002. In fact, the smallest SSI1 in March in the Italian grid cell is -1.76 and in the Paraguayan cell -1.56. Hence, a severe drought, usually defined below SSI1=-1.65, is never identified in March in the latter grid cells. Considering that the interannual variability is slightly higher in the Paraguayan cell, the results are in line with the hypothesis that standardized anomaly indicators may underestimate drought magnitude in such areas. RDQI1 on the other hand is lower in the latter cell reflecting a stronger streamflow deficit in March 2002. Moreover, drought magnitude in March 2002 for water users depending on reservoir storage (SSI12) is higher in the Paraguayan cell, indicating that the previous 12 months were relatively drier in the Paraguayan cell than in the Italian cell.

Transferring the EP1 time series into a severity indicator by selecting the 20$^{th}$ percentile as threshold and expressing drought severity in units of frequency/probability of non-exceedance, CEP1(20%)_f reveals that the rather strong streamflow anomaly in the Italian cell as indicated by EP1 is only a peak within a moderate drought event as compared to the whole reference period. The value of 0.31 indicates that the drought severity up to March 2002 was exceeded by 69% of all drought events between 1986 and 2015. The low EP1 value of the Paraguayan cell is part of a more severe drought event that was exceeded by only 37% of all (completed) drought events. The higher drought magnitude in Paraguay according to RDQI1 corresponds, by chance, to a higher probability of non-exceedance of this drought event (CRDQI1(-50%)_f). This comparison underlines that indicators of drought magnitude are only suitable for assessing the current status of a drought event, but that they do not allow inferences about the status of the whole drought event compared to all other drought events of the reference period.

All severity indicators except CDQI1-Q80 indicate a stronger drought severity for the Paraguayan cell than for the Italian cell. Selection of the median as threshold results in higher cumulative water deficits than selecting Q80, with 36% of mean annual streamflow volume in the Paraguayan cell and 25% in the Italian cell. Selection of the sum of human surface water demand and environmental water demand as threshold indicates, with 56% and 51% of mean annual streamflow volume, respectively, even higher deficits since the onset of the drought event. Considering five different severity indicators that ex-

press drought severity in terms of frequency, the non-exceedance frequency of drought severity in March 2002 ranges between 0.3 and 0.7 in the Italian cell and between 0.6 and 0.8 in the Paraguayan cell (Table 2). In both grid cells, the indicators that do not assume habituation to interannual variability (CDQI1-Q50_f, CDQI1-WUs_EFR_f and CRDQI1(-50%)_f) show the largest severity.

**Table 2:** Comparison of streamflow drought hazard indicators in March 2002 included in Figs. 3, 4 and 6 for a grid cell in northern Italy (43.75° N, 11.25° E) and eastern Paraguay (-24.25° N, -56.25° E).

| Grid cell | Magnitude[1] (Current streamflow anomaly) | | | | Severity[2] (Cumulative streamflow deficit) | | | Severity[3] (Probability (frequency) of non-exceedance) | | | | |
|---|---|---|---|---|---|---|---|---|---|---|---|---|
| | SSI1 | RDQI1 | EP1 | SSI12 | CDQI1-Q80 | CDQI1 -Q50 | CDQI1 -WUs-EFR | CDQI1 -Q80_f | CDQI1 -Q50_f | CDQI1 -WUs-EFR_f | CRDQI1(-50%)_f | CEP1 (20%)_f |
| Italy | -1.47 [4] | -0.71 | 0.13 [4] | -0.83 | 0.04 | 0.25 | 0.51 | 0.46 | 0.73 | 0.69 | 0.44 | 0.31 |
| Paraguay | -1.08 [5] | -0.78 | 0.07 [5] | -1.32 | 0.04 | 0.36 | 0.56 | 0.57 | 0.83 | 0.80 | 0.83 | 0.63 |

[1]: Magnitude expressed as streamflow anomaly in March 2002 in units of standard deviation (SSI1) or empirical percentiles (EP1) with respect to mean March streamflow values during 1986-2015, relative deviation from mean March streamflow values (RDQI1), and streamflow anomaly averaged over April 2001 to March 2002 in units of standard deviation (SSI12) with respect to all April-to-March periods during 1986-2015.

[2]: Severity expressed as water volume deficit with respect to a threshold as a fraction of mean annual streamflow since drought onset until March 2002.

[3]: Severity expressed as probability (frequency) of non-exceedance in March 2002.

[4]: SSI1=-1.47 is equivalent to Q93 and a return period of 14 years; EP1=0.13 is equivalent to a return period of 8 years.

[5]: SSI1=-1.08 is equivalent to Q86 and a return period of 7 years; EP1=0.07 is equivalent to a return period of 15 years.

As CEP1(20%) and CDQI1-Q80 are both anomaly-based drought hazard indicators that assume habituation to interannual variability, they capture exactly the same drought signals for the grid cell in Northern Italy during the reference period (Fig. 7). Nevertheless, the relative levels of drought severity among the drought events during the reference period differ as CDQI1 detects absolute and CEP1 relative drought deficits. Among the twelve drought events in Fig. 7, the two drought events in 1989 and 1990 have the highest severity levels according to both indicators. However, CDQI1 identifies the 1989 drought as the maximum event, while CEP1 detects the 1990 drought as the maximum event. This can be explained by the fact that the mean calendar month streamflow is lowest between June and October. Consequently, higher absolute streamflow deficit volumes (CDQI1) can build up during the 1989 drought ending in June 1989 than in the following drought event span-

ning over June to October 1990. Relative streamflow deficits on the other hand are larger for the latter. This is in line with the higher CDQI1-Q80_f value in March 2002 (0.46) compared to 0.31 for CEP1(20%)_f (Table 2) since the March 2002 event occurs outside the low-flow period. The short drought from June to August 2012 illustrates the difference between both indicators even better. This low-flow drought results in comparably low absolute streamflow deficits but high relative anomalies per calendar month. Among the twelve drought events, this drought event has only the second lowest drought severity according to CDQI1-Q80_f, but the 5$^{th}$ highest severity based on CEP1(20%)_f. Consequently, when monitoring drought hazard, the severity of drought events during low-flow periods with high negative impacts is underestimated relative to other drought events based on frequency values of volume-based severity indicators. Of course, this is only true if monthly deficits are either not normalized (e.g. the low-flow index LFI, Cammalleri et al., 2016a) or normalized against mean annual streamflow volume (e.g. van Loon et al. (2014) and all CDQI1 variants in this paper) instead of mean monthly values.

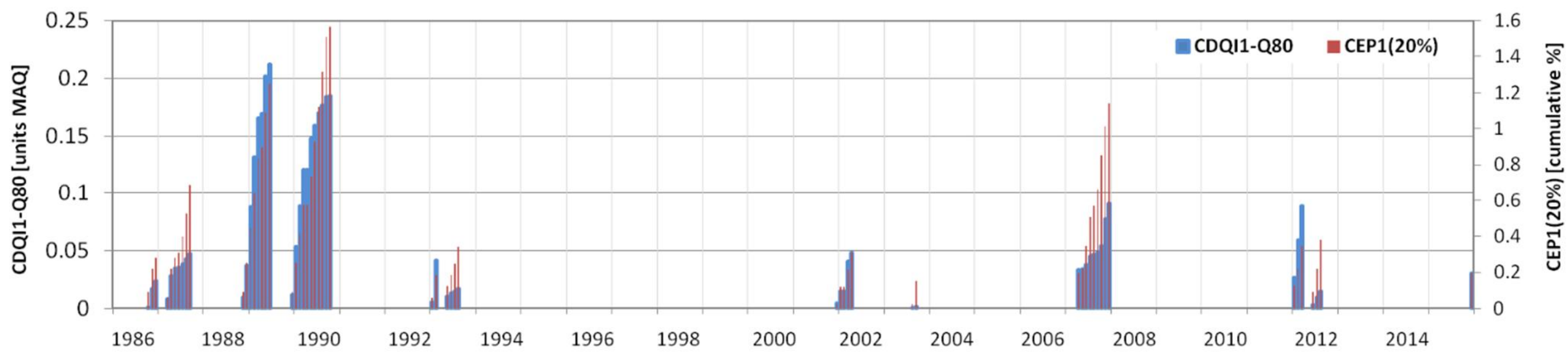

**Figure 7:** Drought severity per month (level 2) during the reference period 1986-2015 for a grid cell in Northern Italy (43.75° N, 11.25° E) as indicated by CDQI1-Q80 (blue) and CEP1(20%) (red). MAQ: mean annual streamflow volume.

### 3.4 Model validation

In an assessment of streamflow drought as quantified based on either observed or simulated streamflow at 293 locations in Europe (Tallaksen and Stahl, 2014), WaterGAP performed well as compared to six other global hydrological models (GHMs). In their Fig. 3, WaterGAP results are better than the multi-model median for all four performance measures. Moreover, WaterGAP performs best regarding the simulation of drought persistence (their Fig. 4). Prudhomme et al. (2011) analyzed the ability of three GHMs to reproduce historical streamflow drought events in European basins using the regional deficiency index (RDI). While all three models are found to broadly capture the spatiotemporal drought development, the authors conclude that WaterGAP "is arguably best suited to reproduce most regional characteristics of large-scale high and low flow events in Europe" (Prudhomme et al., 2011: 1181). However, WaterGAP tends to overestimate the variability in RDI, which is explained by insufficient soil storage capacity. In an intercomparison study among six GHMs (Zaherpour et al., 2018), WaterGAP showed the best results in simulating monthly streamflow in 27 out of 40 river basins worldwide and in each of the eight hydrobelts (their Fig. 2 and Table 3). In five out of eight hydrobelts, the mean weighted absolute error of Q95 was lowest for WaterGAP. Nevertheless, the study revealed that WaterGAP tends to overestimate low flows, and that discrepan-
cies between simulated and observed seasonality and interannual variability can be significant. In a different multi-model validation study based on five global hydrological and land surface models (Veldkamp et al., 2018), WaterGAP was the only model that slightly underestimated variability in monthly streamflow while the others overestimated variability. Correlation with observed monthly streamflow though was highest for WaterGAP in both managed and near-natural basins across the globe (their Fig. 3h). Döll et al. (2016) compared monthly low-flow Q90 as computed by the GHMs WaterGAP and PCR-GLOB-WB to observations at 821 WaterGAP calibration stations across the globe. Overall, low flows could be simulated with reasonable accuracy by both GHMs and were overestimated at most stations. WaterGAP results showed a better fit to observations since it is calibrated against mean annual streamflow at the considered stations (their Fig. 3). Despite calibration, WaterGAP simulations show a lower fit to small observed Q90 values below 1 $km^3$ $month^{-1}$.

As a limited validation exercise in the present study, simulated Q80 values per calendar month were compared to observations from the GRDC database. For 220 out of 1319 WaterGAP calibration stations that have continuous monthly observations between 1986 and 2015, only months with observed Q80 larger than zero were assessed, resulting in a sample size of 2572 months. The analysis revealed that Q80 is overestimated by WaterGAP in 63% of the months and in 53% of the months if only relevant deviations > 10% are considered. Regarding negative deviations, WaterGAP significantly underestimates Q80 by more than 10% in 30% of all months. Hence, the results suggest a slight tendency to overestimate low flows, which is in line with the validation studies above.

### 3.5 Correlation between SPIn and SSI1

To analyze if either simulated SSI1 (SSI1 (sim)) or SPIn is a better estimator of observed streamflow drought hazard, monthly time series of observed SSI1 (SSI (obs)) were correlated with five indicators applying the Pearson correlation: SSI1 (sim), SPI3, SPI6, SPI9, and SPI12. The analysis was limited to the 218 WaterGAP calibration stations with continuous time series of observed monthly streamflow during the reference period 1986-2015 for which all SPI variants were computable. Although longer averaging periods for SPI from 12 to 24 months are recommended for hydrological drought assessments (WMO and GWP, 2016), different studies at the global (Gevaert et al., 2018; Vicente-Serrano et al., 2012) and the regional scale (Yu et al., 2020; Huang et al., 2017; Barker et al., 2016) have demonstrated that shorter averaging periods often perform better in estimating streamflow drought hazard. Using an ensemble of seven global land surface and hydrological models, Gevaert et al. (2018) found that the optimal SPI averaging period strongly varied among the models and with the season and climate regime. Different SPI variants from SPI1 to SPI24 were identified to correlate best with modeled SSI1 time series. In regional studies covering arid to humid climate and basin areas ≤ 10,000 $km^2$, SPI1 to SPI4 had the highest agreement with observed SSI1 values. Only in some humid basins underlain by productive aquifers, longer averaging periods from 6 to 19 months showed the best results. Basin size was found to be positively correlated with higher averaging periods (Yu et al., 2020). Moreover, shorter averaging periods performed better during spring and summer and longer averaging periods in autumn and winter (Huang et al., 2017). The strong influence of basin properties such as area and storage capacity as well as the climate regime was also discussed in van Loon (2015).

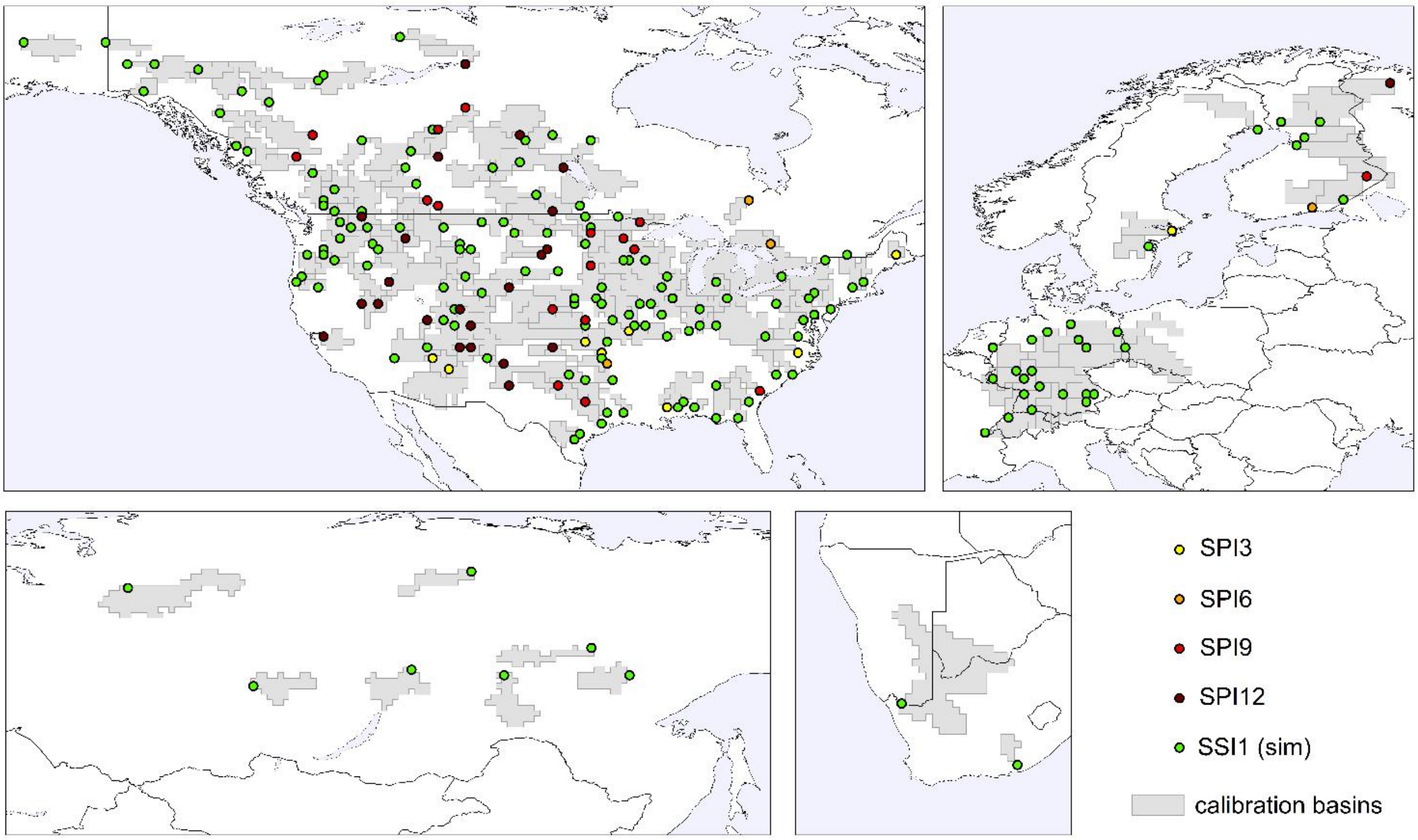

**Figure 8:** Ranking of the Pearson correlation coefficient r at 218 WaterGAP calibration stations between SSI1 (obs: observed streamflow) and each of the five indicators SSI1 (sim: simulated streamflow), SPI3, SPI6, SPI9 and SPI12 for the reference period 1986-2015.

Figure 8 depicts the 218 WaterGAP calibration stations and their basins located in North America, Northern and Western Europe, Russia, and South Africa. At each station, the drought hazard indicator that achieved the highest Pearson correlation coefficient is indicated. Overall, SSI1 (sim) performed best at 165 stations with a median correlation coefficient of 0.7, followed by SPI12 at 23 stations. Among the 165 stations, the next highest correlation was achieved by SPI3 and SPI12 at 50 and 47 stations with median correlation coefficients of 0.65 and 0.46. The performance of WaterGAP is often lower in semi-arid basins, in particular where streamflow is highly altered by irrigation and man-made reservoirs, and in regions dominated by lakes. The SPI indicators outperform SSI1 (sim) only in smaller basins (Fig. 9). Nonetheless, the total number of smaller basins < 80,000 km² where SSI1 (sim) is a better estimator of observed drought hazard still exceeds the number of basins of all SPI variants. Comparing the SPI variants, it would be expected that longer averaging periods should better capture the drought hazard in larger basins where drought propagation through the hydrological compartments is more delayed. In Fig. 9, there is a slight tendency that the 9- and 12-month averaging periods show better results in the larger basins, but this cannot be clearly deducted due to the limited sample size. In conclusion, drought hazard indicators based on modeled streamflow can

Natural Hazards and Earth System Sciences
EGU
Discussions

often better quantify observed streamflow drought hazard than meteorological indicators, but this strongly depends on the goodness-of-fit after streamflow calibration.

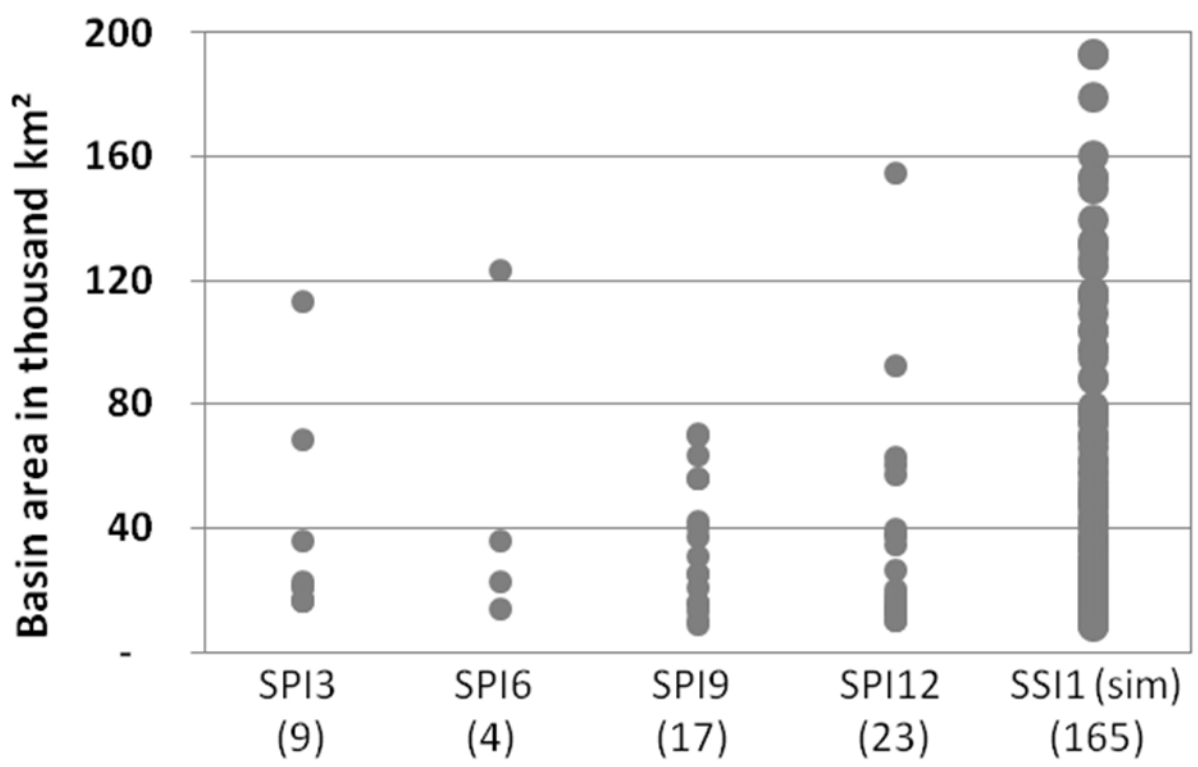

**Figure 9:** Basin area of the 218 WaterGAP calibration stations for each indicator with the highest ranking of the Pearson correlation coefficient r. The total number of stations is indicated in parentheses. The 32 largest basins of SSI(sim) > 200,000 km² are not depicted.

## 4 Recommendations

### 4.1 General recommendations regarding the selection of drought hazard indicators

Drought hazard indicators serve to assess drought risk and to support drought management. Therefore, when deciding on which drought hazard indicator to use, it is first necessary to clearly define "the risk of what for whom" that is to be addressed by the indicator. Drought hazard indicators are risk system specific, and there is not one that fits all. As drought is conceptualized as both "less water than normal" and "less water than needed", the choice of indicator as well as the interpretation of the quantitative indicator values implicitly includes assumptions about what is normal and what is needed, i.e., to what amount of available water people or ecosystems are habituated to without suffering from drought. For example, is the risk bearer well adapted to the seasonal variability as well as the water availability that is exceeded in 8 out of 10 months per calendar month, even though the latter is only a small fraction of the mean or median water availability as is the case in dry regions of the globe? Then, an anomaly-based indicator with the Q80 values of each calendar month as thresholds may be suitable. However, the spatial distribution of the values of such an anomaly indicator is only consistent and informative if it can be assumed that risk bearers everywhere on the map are habituated in the same way. Or is the risk bearer not well adapted to a high interannual variability and would suffer from even small reductions from mean/median streamflow? Then median values could be used as threshold. Alternatively, it could be assumed in such a case that the drought hazard increases with the percent reduction of water availability from average availability in a calendar month. Then, an indicator based on the relative deviation of

the current water condition of the mean calendar month condition is informative. When selecting a drought hazard indicator, we recommend that the assumptions about the habituation of the risk bearer are made explicit first, based on knowledge about the risk system, which is then followed by the selection of a drought hazard indicator that fits to these assumptions.

We also recommend differentiating clearly drought magnitude indicators from drought severity indicators. Magnitude indicators with short averaging periods such as 1 month provide information on current, potentially extreme condition of the water flow or water storage under consideration. They should be used if the well-being of the risk bearer, e.g., river biota, strongly depends on the water condition at the specific time step of the analysis, e.g., the current month. As negative drought impacts are mostly assumed to increase with the length of the drought, however, severity indicators are often more informative than magnitude indicators as they quantify the cumulative magnitude since start of the drought. All drought magnitude indicators can be used to derive drought severity indicators. With exceptions, we recommend that drought severity at a certain point in time is expressed in terms of the probability/frequency of occurrence (return period) of a drought event with such a severity. These recommendations relate to any type of drought variable (precipitation, soil moisture, etc.) and spatial scale.

### 4.2 Recommendations for streamflow hazard indicators in continental and global DEWS

Continental and global DEWS, which encompass near-real time monitoring as well as seasonal forecasts, are to inform about drought hazards for diverse risk systems, which are characterized by different risk bearers (e.g., human water supply, river ecosystems), habituation, streamflow regimes and water storage capacities. Therefore, a large-scale DEWS should provide data for a rather large number of drought hazard indicators together with a clear description of suitability for different risk systems. Then, end-users can select and combine a number of drought hazard indicators that are most informative (as is done e.g., for generating information shown by the US Drought Monitor). Table 3 lists drought hazard indicators that should be provided by large-scale DEWS, together with their suitability for drought risk assessment for 1) human water supply from surface water and 2) river ecosystems, distinguishing intermittent and perennial streamflow regimes as well as low and large water storage capacities.

**Table 3:** Streamflow drought hazard indicators that should be provided by large-scale DEWS, together with their suitability for drought risk assessment for 1) human water supply from surface water and 2) river ecosystems, distinguishing intermittent and perennial streamflow regimes as well as low and large water storage capacities.

| | Intermittent streamflow | | Perennial streamflow | |
|---|---|---|---|---|
| Indicators of | 1) Water users **without** access to large reservoirs<br>2) river ecosystems | 1) Water users **with** access to large reservoirs | 1) Water users **without** access to large reservoirs<br>2) river ecosystems | 1) Water users **with** access to large reservoirs |
| Magnitude | *Return period based on EP1*<br>RDQI1 | *Return period based on EPn*[1]<br>RDQIn[1] | *Return period based on EP1*<br>RDQI1 | *Return period based on EPn*[1]<br>RDQIn[1] |
| | In regions with (suspected) poor quality of hydrological model output, analyze SPEIn, in addition to streamflow indicators. | | | |
| Severity | *CDQI1-Q80*<br>*CDQI1-Q80_f* | *CDQIn-Q80*[1]<br>*CDQIn- Q80_f*<br>*CDQI1-Q80-HS*<br>*CDQI1-Q80-HS_f* | *CDQI1-Q80*[2]<br>*CDQI1-Q80_f*[2] | *CDQIn-Q80*[1]<br>*CDQIn-Q80_f* |
| | | *CEPn(20%)_f*[1]<br>*CEP1(20%)-HS_f* | *CEP1(20%)_f*[2] | |
| | CRDQI1(-50%)_f | CRDQIn(-50%)_f[1]<br>CRDQI1(-50%)-HS_f | CRDQI1(-50%)_f | CRDQIn(-50%)_f[1] |
| | **CDQI1-WUs-EFR**<br>**CDQI1-WUs-EFR_f** | **CDQIn-WUs-EFR**[1]<br>**CDQIn-WUs-EFR_f** | **CDQI1-WUs-EFR**<br>**CDQI1-WUs-EFR_f** | **CDQIn-WUs-EFR**[1]<br>**CDQIn-WUs-EFR_f** |

[1] n: For global-scale DEWS, an averaging period n of 6 or 12 months is suggested.
[2] CEP1(20%)_f preferred over CDQI-Q80 indicators.
*italics*: Indicator assumes habituation to a certain degree of interannual variability (see Fig. A1b).
**bold**: Indicator assumes the ability to fulfill seasonally varying demand for surface water abstractions and environmental flow.
normal font: Indicator assumes habituation to a certain reduction from mean monthly streamflow.

To assess drought *magnitude*, we recommend using empirical percentiles and relative deviations to cover risk systems that are either habituated to a certain degree of interannual variability or to a certain reduction to mean calendar month streamflow. An averaging period of 1 month is suitable for river ecosystems and water users without access to large reservoirs, who depend on the water as it flows down the river. Longer averaging periods of 6 or 12 months are suitable in regions where people have access to reservoir storage that is replenished during high-flow periods and that can alleviate short periods of below-normal streamflow. We favor empirical percentiles EP over SSI as the former are more transparent to end-users of a DEWS and do not entail uncertainties due to the fitting of a probability distribution. Moreover, application of one selected

probability distribution function at large scales will always exclude many grid cells where the fitting is not possible. Here, other methods such as empirical percentiles would be required in any case. Expressing percentiles as return period (in years) may further increase the transparency of EP as end-users are accustomed to quantifying flood hazards by return periods.

For all four risk systems in Table 3 (intermittent or perennial streamflow, both with and without access to large reservoirs), drought *severity* should be assessed with indicators that imply habituation to a certain degree of interannual variability (CEP variants and/or CDQI-Q80 variants), to a certain reduction from mean monthly streamflow (CRDQI variants), and to the ability to fulfill seasonally varying water demand from surface water abstractions and environmental flow (CDQI-WUs-EFR variants). The severity indicators expressed in cumulative percent (CEP and RDQI variants) should be provided as frequency of non-exceedance (denoted with suffix “f”) as the informative value of a cumulative percentage is low. Application of longer averaging periods of 6 or 12 months is recommended for all severity indicators in regions with large reservoirs where the impact of short-term droughts below 6 months is probably low. In addition, the HS method (Sect. 2.2.3) is recommended in intermittent flow regimes with reservoirs for CDQI1, CRDQI1 and CEP1. The method allows existing high-flow droughts to continue during low-flow periods (defined as calendar months with Q80=0 or MMQ=0). The HS method follows the assumption that risk bearers in these regions cannot recover from high-flow droughts during low-flow periods such that drought severity should be kept at the initial level. CEP1(20%)-HS_f (not assessed in this study) is computed like CDQI1-Q80-HS_f (Sect. 2.2.3) with potential drought prolongation in calendar months with Q80=0. CRDQI1(-50%)-HS_f (not assessed in this study) allows an existing high-flow drought to continue in calendar months where mean monthly streamflow MMQ is zero (and thus all 30 streamflow values in this calendar month). The CDQI variants are suitable for all risk systems as they inform end-users of a DEWS about drought severity in units of absolute streamflow volume. CEP1 was found to be more sensitive to low-flow droughts than CDQI1 (Fig. 7 and Sect. 3.3.4). In intermittent streamflow regimes without reservoirs, however, droughts are mostly detected during the high-flow periods, and CEP1(20%)_f would not add value to a drought hazard assessment. In intermittent streams with reservoirs on the other hand, application of CEPn(20%)_f or CEP1(20%)-HS_f is valuable, since these indicators quantify relative streamflow deficits, and the ranking of drought events by their severity is different from the ranking according to CDQI variants (Fig. 7). Regarding the risk for ecosystems or water supply in perennial rivers without large reservoirs, CEP1 is preferred over CDQI1-Q80 due to the sensitivity of the former to low-flow droughts. In perennial rivers with large reservoirs upstream, CDQIn-Q80 is preferable.

According to Stahl et al. (2020), practitioners often use particular streamflow values rather than anomalies as trigger for management actions. These practitioners could use forecasted RDQI1 as provided by the global-scale DEWS to determine whether this trigger will be reached by computing streamflow from RDQI1 and observed mean monthly streamflow.○

## 4 Conclusions

This paper presents a new systematic approach for selecting global-scale hazard indicators for monitoring drought risk for human water supply and river ecosystems. The methodology replaces the conventional and imprecise classification into threshold-based and standardized indicators by a new taxonomy that distinguishes indicators pertaining to four indicator types by a) their inherent assumptions about the habituation of people and the ecosystem to the streamflow regime and b) their level of drought characterization, namely drought magnitude and drought severity, with the latter either in units of cumulative drought magnitude or in terms of frequency of occurrence. We applied the new classification scheme to a set of eight existing and three newly developed drought hazard indicators that can be meaningfully quantified at the global scale. Several types of habituation to the streamflow regime were covered including the habituation to a certain degree of interannual variability of streamflow, seasonality, a certain reduction from mean calendar month or mean annual streamflow and being able to fulfill the demand for surface water abstractions and environmental flow. We analyzed the sensitivity of indicated drought hazard to the choice of indicator and discussed which indicator is best suited for quantifying a specific drought risk for a specific system at risk. The new classification scheme facilitates a better understanding of the information value of drought hazard indicators. It can support the development of a (large-scale) DEWS as well as water managers who rely on the output of drought hazard indicators.

The eleven drought hazard indicators were quantified for the reference period 1986-2015 using the latest version of the global water resources and water use model WaterGAP. Indicators of drought magnitude included the standardized anomaly indicators SPI12, SPEI12, SSI1, SSI12, empirical percentiles EP1, and relative streamflow deviations from mean conditions, RDQI1 and RDQI12. Indicators of drought severity comprised the cumulative volume-based drought severity indicators CDQI1-Q50 and CDQI1-Q80 (with median streamflow and Q80 as threshold), cumulative empirical percentiles CEP1(20%) ($20^{th}$ percentile as threshold), and cumulative relative deviations from mean conditions CRDQI1(-50%) (-50% as threshold). We developed an approach for handling intermittent streamflow conditions in the computation of severity indicators as well as a new severity indicator for highly seasonal (HS) streamflow regimes with access to large reservoirs, CDQI1-Q80-HS, that allows existing droughts to continue during calendar months with Q80=0. The rationale behind this approach is that streamflow during low-flow months is not relevant for people relying on large reservoirs. Below-normal water storages can only marginally be replenished during a low-flow period, and hence drought severity should remain at the level of the preceding high-flow period. Moreover, two new water deficit indicators were developed, CDQI1-WUs and CDQI1-WUs-EFR, both considering mean monthly surface water use, and in case of the latter also mean monthly environmental flow requirements, EFR, assumed to be 80% of mean monthly naturalized streamflow. CDQI1-WUs-EFR is the only indicator in this study that takes into account ecosystem health when assessing drought risk for human water supply, an aspect that should be included in a global-scale DEWS.

Natural Hazards and Earth System Sciences
EGU
Discussions

The comparison of indicators shows, for the first time explicitly for the two levels of drought characterization (drought magnitude and severity), how conceptualization and selection of indicators can lead to very different spatial and temporal patterns of drought hazard. Indicators of drought magnitude are only suitable for assessing the current status of a drought event, but they do not allow inferences about the status of the whole drought event compared to all other drought events of the reference period. Using two example grid cells, the set of indicators resulted in a high range of non-exceedance frequencies (0.3-0.7 and 0.6-0.8) of drought severity of the same drought event. Even two similar indicators like CDQI1-Q80 and the cumulative empirical percentile CEP1(20%) with a threshold equivalent to Q80 can lead to different frequencies of occurrence for the same drought event.

A limited validation exercise revealed that SSI1 based on modeled streamflow often outperformed SPI with different averaging periods, but this strongly depends on the goodness-of-fit after streamflow calibration. In uncalibrated basins, meteorological drought indicators should be used complementarily as proxies for hydrological drought hazard due to the uncertainty of modeled streamflow. Human interactions are well represented in WaterGAP, and drought indicators including human water use can be meaningfully quantified at the global scale. Drought propagation in river basins where the streamflow regime is strongly altered by reservoir management or water transfers, however, cannot be adequately represented by WaterGAP and global hydrological models in general.

When providing drought hazard information in a global- or continental-scale DEWS, it is unknown which streamflow characteristics people and river ecosystems are locally accustomed to, and it is uncertain to what degree people have access to water stored in reservoirs. The suitability of hazard indicators is region- and risk-specific (Blauhut et al., 2021) and can only be evaluated with regional knowledge about the vulnerability of the system at risk. Therefore, a large-scale DEWS should provide data for a rather large number of drought hazard indicators that characterize the condition of various water flows (streamflow, actual evapotranspiration as a fraction of potential evapotranspiration) and water storage compartments (snow, soil, groundwater, lakes). A major component of the DEWS are clear explanations for the end-users about the suitability of drought hazard indicators for specific risk systems. When selecting hazard indicators, we recommend that the end-user makes the assumptions about the habituation of the risk bearer explicit, based on knowledge about the risk system, before selecting a drought hazard indicator that fits to these assumptions.

To assess drought magnitude for human water supply from surface water as well as for river ecosystems, we recommend using return periods based on empirical percentiles and relative deviations to cover risk systems that are either habituated to a certain degree of interannual variability or to a certain reduction to mean calendar month streamflow. We favor empirical percentiles over SSI as the former are more transparent to end-users of a DEWS and do not entail uncertainties due to the fitting of a probability distribution. Drought severity should be assessed with indicators that imply habituation to a certain degree of interannual variability (different variants of cumulative empirical percentiles CEP and/or CDQI-Q80 variants), to a certain reduction from mean monthly streamflow (CRDQI variants), and to seasonality and the ability to fulfill water demand from surface water abstractions and environmental flow (CDQI-WUs-EFR variants). The severity indicators expressed in cumulative percent (CEP and RDQI variants) should be provided as frequency of non-exceedance as the informative value of

a cumulative percentage is low. Application of longer averaging periods of 6 or 12 months is recommended for all severity indicators in regions with large reservoirs where the impact of short-term droughts below 6 months is probably low. In intermittent flow regimes with reservoirs, the indicators CDQI1, CRDQI1 and CEP1 should be combined with a method for highly seasonal flow regimes to allow high-flow droughts to continue during low-flow periods.

## Appendix

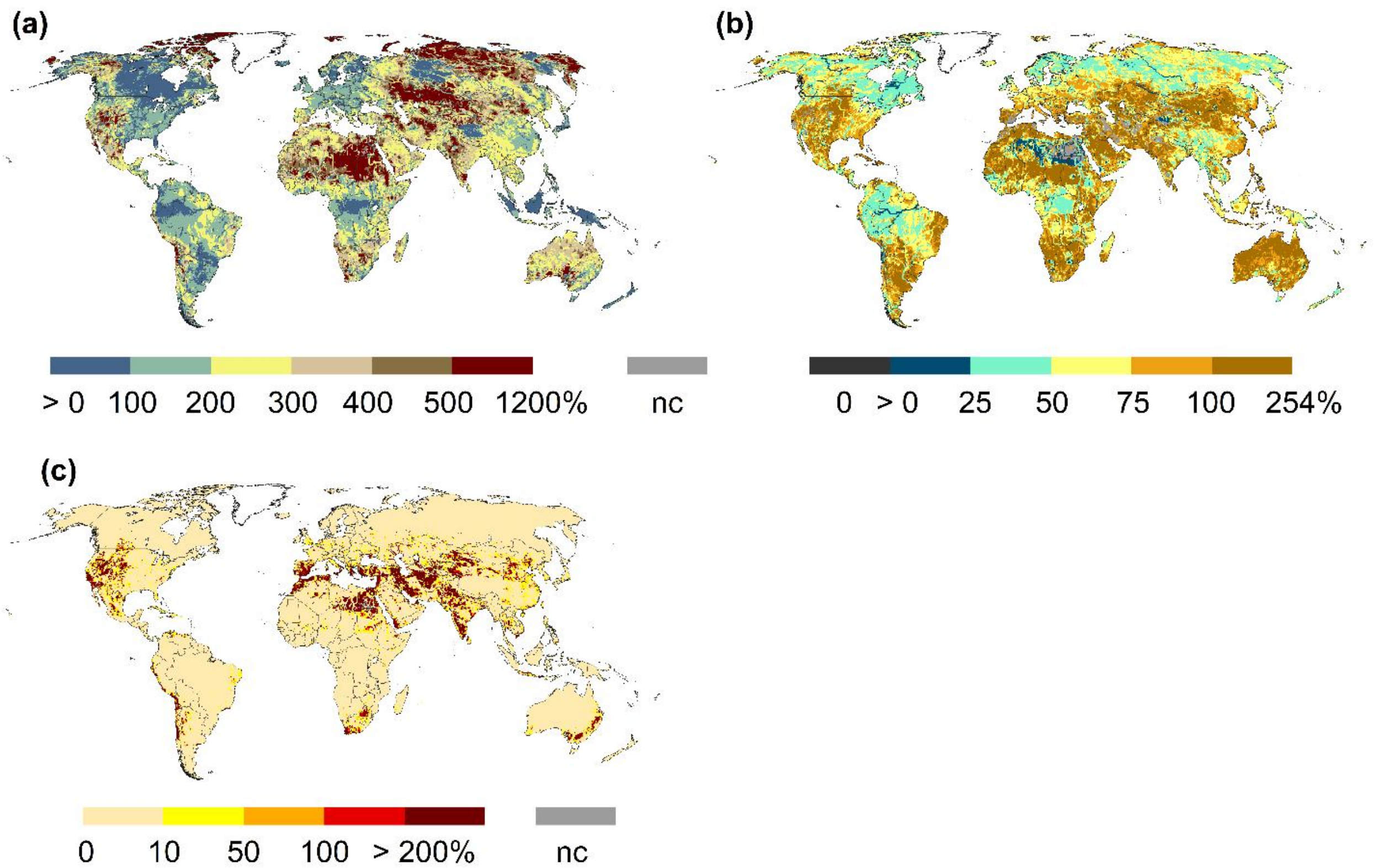

**Figure A1:** Seasonal streamflow variability indicated by the seasonal amplitude, i.e. Q in calendar month with highest mean monthly Q minus Q in calendar month with lowest mean monthly Q divided by MMQ (mean monthly Q over all calendar months) (a), interannual streamflow variability indicated by the average of the 12 calendar month values of (Q20-Q80)/Qmean (b), and average of the 12 calendar month values of WUsmean/Qmean (c). All values in percent.

*Data availability.* WaterGAP 2.2d model output data used in this study are available at https://doi.org/10.1594/PANGAEA.918447 (Müller Schmied et al., 2021). The outputs from this study are available at

https://zenodo.org/record/6647609 (Herbert and Döll, 2022). GRDC monthly streamflow data are available at: http://grdc.bafg.de (GRDC, 2019).

*Author contributions.* This manuscript was conceptualized by PD and CH. CH conducted the data analysis, visualization, and interpretation. The original draft was written by CH and revised by PD.

*Competing interests.* The authors declare that they have no conflict of interest.

*Acknowledgments.* We thank Eklavyya Popat for computing the time series of SSI1 and SSI12.

*Financial support.* This research was part of the project GlobeDrought and has been supported by the German Federal Ministry of Education and Research (BMBF; grant no. 02WGR1457B) through its Global Resource Water (GRoW) funding initiative.

This open-access publication was funded by the Goethe University Frankfurt.

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
