# Peer review of "Analyzing the informative value of alternative hazard indicators for monitoring drought risk for human water supply and river ecosystems at the global scale"

_Natural Hazards and Earth System Sciences, 2022_

## Referee Comment (RC3)

[referee-annotated manuscript omitted]

---

## Author Comment (AC1)

https://doi.org/10.5194/nhess-2022-174-RC1, 2022
Nat. Hazards Earth Syst. Sci. Discuss.

**"Analyzing the informative value of alternative hazard indicators for monitoring drought risk for human water supply and river ecosystems at the global scale"** by Claudia Herbert and Petra Döll

**Response to Anonymous Referee #1**

We thank you very much for your helpful comments and constructive suggestions for improving the manuscript. Below, each comment (in italics, indicated by "**RC**") is followed by our answer (normal font, indicated by "**AC**"). Changes in the manuscript are written in bold.

*RC: In this paper the authors present an approach for the selection and calculation of streamflow drought hazard indicators for monitoring human surface water supply and for river ecosystems. In doing so, the authors discuss and propose to consider the habituation of people and ecosystems to the streamflow regime. For this purpose, eight existing drought indicators are compared and quantified and three new ones are proposed based on the global hydrological model WaterGAP2.2d.*

*This article has positive aspects, in particular the effort in modelling that the authors have made is remarkable, however it needs to be reviewed thoroughly as there are several points that need to be clarified and addressed by the authors. In short, the article has potential, but the authors need to make an effort to focus the analysis and make a readable manuscript. I hope that the comments below will help in this direction.*

      **AC**: Thank you for the positive feedback.

**Structure of the article**

*RC: First, the structure of the paper could be improved. It is very long, repetitive in some aspects, lacking clarity on the objectives and results, that should be better highlighted both in the description and conclusions sections.*

*Similarly, the order of the sections does not follow a story, some sections could be shortened, removed, or moved to increase the readability. This structure surely makes it very difficult to read. For example, the introduction should be significantly shortened to focus on the description relevant to the objectives of this work. Section 3.4 should be moved into the methodologies section. In addition, there is an annex that includes only one figure that is quite relevant. I suggest including it in the main text, if the number of figures is an editorial requirement I suggest removing some other figures (e.g. Figure 9).*

      **AC**: We deleted and shortened repetitive or too detailed paragraphs throughout the text and used more concise language. We restructured and renamed several sections to improve the storyline of the manuscript. "Results and Discussion" (old Sect. 3) are now divided into Sect. 3 and 4, with Sect. 3 focused on the proposed systematic approach for selecting streamflow drought hazard indicators (SDHIs) and Sect. 4 comprising the global-scale assessment.

    **3 Proposed systematic approach for selecting and computing SDHIs** (old Sect. 3.1)

        **3.1 Assumptions about habituation inherent in drought hazard indicators** (old Sect. 3.1.1)

        **3.2 Levels of drought characterization** (old Sect. 3.1.2)

        **3.3 Illustration of habituation-based classification approach** (old Sect. 3.2, new title)

    **4 Quantification of global streamflow drought hazard by a global hydrological model** (old Sect. 3.3, new title)

        **4.1 Model validation** (old Sect. 3.4)

        **4.2 Discrepancies in drought hazard as quantified by different SDHIs** (old Sect. 3.3, new title)

**4.2.1 Drought magnitude (level 1)** (old Sect. 3.3.1)

**4.2.2 Drought severity (level 2)** (old Sect. 3.3.2)

**4.2.3 Drought severity expressed as frequency of non-exceedance (level 2)** (old Sect. 3.3.3)

**4.2.4 Relation between the various SDHIs** (old Sect. 3.3.4)

**4.2.5 Suitability of SPIn to quantifiy streamflow drought hazard** (old Sect. 3.5, new title)

**AC**: We added the following paragraph at the beginning of the new Sect. 3 to clarify the objectives and the structure:

**Wilhite und Glantz (1985) suggested distinguishing between a conceptual and an operational drought definition, with the former referring to the general qualitative concept of drought and the latter allowing for a quantitative drought characterization including onset, severity, termination, and spatial extent. In the following Sect. 3.1, aspects that relate to the conceptual drought definition are discussed comprising the description of the targeted drought risk and the system at risk. In particular, assumptions about the habituation of the system at risk to the streamflow regime are discussed, an aspect that is currently not taken into account or not made explicit in drought hazard studies. In order to translate these conceptual definitions into operational drought hazard indicators, a new classification system for hazard indicators is proposed in Sect. 3.2. The new systematic approach is illustrated in Sect. 3.3 for selected SDHIs using streamflow observations at two gauging stations with different streamflow regimes.**

**AC**: We added the following paragraph at the beginning of the new Sect. 4 to clarify the objectives and the structure:

**The objective of this chapter is to identify which of the SDHIs presented in Table 1 can be meaningfully quantified at the global scale using WaterGAP 2.2d and which SDHIs are appropriate for monitoring different drought risks in large-scale DEWS. After a limited validation of modeled streamflow (Sect. 4.1), SDHIs of drought magnitude and severity are compared separately (Sect. 4.2.1-4.2.3) following the classification system presented in Fig. 1. The SDHIs are shown in global maps for a selected month (March 2002), as it is important to understand the relation between indicators at a certain point in time, especially for the application in DEWS, which are focussed on the current situation or the near future. As patterns of indicators depend on characteristic of the streamflow regime and water use that are temporally constant over the reference period, the reasons for similarities and differences between indicators can be deducted in any month of the reference period. March 2002 was selected as it was among the months with the highest difference between CQDI-Q80 and CQDI-Q80-HS. In addition to the analysis for the selected time step, the latter two indicators are compared at the global scale with respect to drought occurrence during the whole reference period. Discrepancies and similarities of the indicators are discussed in more detail for two illustrative grid cells with the same CQDI-Q80 value in March 2002 (Sect. 4.2.4). Finally, the suitability of SPI with different averaging periods to estimate streamflow drought hazard is assessed using streamflow observations from 218 GRDC gauging stations (Sect. 4.2.5). Based on this global-scale analysis and the proposed habituation-based classification approach, selected SDHIs are recommended for implementation in large-scale DEWS (Sect. 5).**

**AC**: In the introduction, we deleted paragraphs and sentences that anticipated aspects of the proposed new classification system (e.g., lines 92-118) as well as repetitive and too detailed sentences (e.g., lines 141-147). Overall, we shortened the introduction by one page.

**AC**: In the introduction, we added the following sentence (in bold) to highlight the research gap with respect to drought hazard concepts:

[Clearly, the conception or selection of hazard indicators needs to take into account the habituation and thus vulnerability of the system at risk.] **However, investigations and guidance on how to select the optimal SDHI, considering both the targeted risk and the habituation of the system at risk to the streamflow regime, are missing.**

**AC**: In the introduction, we added the following sentence to highlight the research gap with respect to a hazard indicator classification:

**Certainly, an improved classification of drought hazard indicators would facilitate a better understanding of drought characteristics and provide guidance in selecting appropriate drought hazard indicators.**

**AC:** The first paragraph of the model validation (lines 823-843) describing the performance of WaterGAP in model intercomparison studies was moved to the end of the model description (Sect. 2.1). Here, we added an introductory sentence and a recently published study:

**In several model intercomparison studies, WaterGAP was often among the best performing global hydrological models (GHMs). Kumar et al. (2022) assessed the ability of nine catchment-scale models and eight GHMs to simulate hydrological droughts in eight large catchments around the world. Comparing simulated and observed streamflow deficits and SSI1 (SRI) (their Tables 2 and 3), WaterGAP is among the two to three best performing GHMs with performance indicators ($R^2$ and Nash-Sutcliffe efficiency) comparable to those of the catchment-scale models.**

**AC:** The second paragraph of the model validation (old Sect. 3.4, lines 844-850) was moved to the beginning of the new Sect. 4 (see new structure above) as suggested by referee #3. We added the new Fig. 3 below for a more detailed analysis of Q80. Furthermore, we added the paragraph below describing the results of another recent WaterGAP model validation. Hence, the model validation is now focused on Q80 and a SDHI (SSI3), which are both relevant for the subsequent global-scale analysis of SDHIs.

[Figure]

**Figure 3: Percent deviations of simulated Q80 per calendar month from Q80 based on GRDC observations using the reference period 1986-2015.**

**In a recent study, WaterGAP 2.2d model output was validated against GRDC data by comparing SSI3 based on simulated and observed monthly streamflow (SSI3(sim) and SSI3(obs)) during 1971-2000 at 183 globally distributed GRDC stations (Wan et al. 2021). Applying drought hazard classes for SSI according to Agnew (2000), the agreement between simulated and observed hazard classes in each month was analyzed. Among all stations, the agreement ranged between 29 to 88% of all 360 months (their Fig. S4 and Table S3). At 68% of all stations (covering 83% of the assessed basin area), SSI3(sim) and SSI3(obs) resulted in the**

**same drought hazard class in 70 to 88% of the time. Moreover, the goodness-of-fit was evaluated based on the Nash-Sutcliffe efficiency (NSE) for monthly streamflow and SSI3 (their Fig. S3). With a median NSE of 0.5 and an interquartile range of 0.2-0.7 for SSI3 and 0.14-0.7 for streamflow, WaterGAP 2.2d model output showed a moderate agreement with the observations. Both NSEs exceeded 0.7 at 25 out of the 183 stations, which are located in Central and Eastern Europe (twelve stations), the United States (ten stations), and South Africa (one station).**

**AC**: We think that Figure A1 should remain in the appendix, since it does not show indicators, and since we refer to this figure in almost all chapters.

**AC**: We deleted the old Fig. 9 as suggested.

**AC:** We shortened the conclusion section by almost one page. For instance, we deleted the last paragraph (lines 1033-1045) and summarized the recommendations as follows:

**Out of the twelve analyzed SDHIs, we recommend a set of magnitude and severity indicators for large-scale DEWS specific to the risk systems 1) human water supply from surface water and 2) river ecosystems, distinguishing intermittent and perennial streamflow regimes as well as low and large water storage capacities.**

*RC: The methodology can also be reduced, focusing on the description of the proposed new indicators and model validation. In addition, some aspects should be clarified, e.g., eq(1) for SPI, X is noted as a generic variable, if this variable is precipitation this representation could be misleading.*

**AC**: Given the inconsistent indicator descriptions identified in the literature (or the absence of any description), we think that the detailed description of the conventional indicators is as important as the description of the new indicators. We would like to keep equation 1 in this generic form as it can be used for any variable. However, we adjusted the description below the equation as follows:

with X = variable **(e.g. precipitation)**, $\mu$ = mean, and $\sigma$ = standard deviation.

*Focus of the manuscript and sectoral risk representation*

*RC: The authors indicate that the focus of the article is on analysing indicators for monitoring drought risk in very specific sectors (human water supply and river ecosystems). I agree that the selection of the hazard indicator is key to determine this dimension of drought risk and the discussion on that direction is more than welcome and needed. However, apart from referring to these sectors as the focus of the article and several speculative and unsupported assertions, there is no information in this article on how drought specific indicators affect these sectors. The dynamics of how these sectors or systems are affected is surely complex, depending on various factors that determine their exposure and vulnerability beyond whether they rely on upstream reservoirs or the systems are seasonally dependent. However, this is not enough to characterize the vulnerability of these sectors. Further discussion and analysis in this regard is needed.*

*On the one hand, considering the way the article is structured, orienting the analysis in the description and comparison between indicators, derived metrics, etc. in a concise and targeted manner can improve the focus and structure of the article. On the other hand, to strictly evaluate whether the proposed indicators are valid to represent risk, a more detailed analysis of the proposed sectors is needed, with a description of their vulnerabilities and how their impacts are produced as a consequence of the combination of the different dimensions of risk.*

**AC:** WaterGAP 2.2d output was used in two drought risk studies for South Africa and at the global scale (Meza et al. 2021; Meza et al. 2020). We think that such and impact assessment is beyond the scope of this

study. Especially the South Africa study showed that vulnerability and exposure indicators can only be meaningfully estimated at the regional to local scale. Such information is not available at the global scale. We do not make "speculative and unsupported assertions" about sectors. On the contrary, we write (e.g., in the conclusions, lines 1023-1029) that this kind of information is not available at the global scale and we recommend providing different hazard indicators covering different habituations to the streamflow regime. Then, people with local knowledge on this type of information can decide, which hazard indicator fits best to the targeted risk. To clarify that hazard indicators are only one out of three components to estimate drought risk, we added the following sentence in the new Sect. 3.1 (Assumption about habituation inherent in drought hazard indicators):

**In conclusion, percentile-based hazard indicators and relative deviations from the long-term mean or median should be used complementarily in large-scale DEWS in combination with adequate vulnerability and exposure indicators to cover different drought risks.**

*Grid cells – case study selection and description*

*RC: The selection of the two grid cells seems to be motivated by characteristics derived from some of the modelled variables. However, no description of these sites exists in the manuscript. A quick search turns up that one gridcell is in central eastern Paraguay (perhaps including a portion of the Paraguay river) and the second near Firenze (Arno). Both points with very different realities regarding how they might be exposed and how they are vulnerable to droughts and surely each will have a very specific risk profile. Here I see a missed opportunity, as one of the objectives of this paper is to find out how the exposed systems can be used. Surely, a discussion along these lines would greatly improve the discussion on the usefulness of the various indicators. Indeed, the comparison between indicators is merely informative, which does not enable identification or validation between them (Please refer to my final comment) Similarly, March 2002 has been used to describe Figures 3 and 4. But it is not clear why this period was chosen or how this comparison can be extrapolated to the whole period.*

AC: With respect to the two grid cells, we rephrased the first sentence in the new Sect. 4.2.4 (see below). Discussion of the risk profile in the two grid cells is, from our point of view, beyond the scope of this study as described in our previous answer.

**The relation between selected SDHIs from Figs. 4, 5 and 7 is compared for two illustrative grid cells that share the same CQDI-Q80 value of 0.04 in March 2002 (Table 2).**

AC: With respect to the selected month March 2002, we added the following sentences at the beginning of the new Sect. 4:

**The SDHIs are shown in global maps for a selected month (March 2002), as it is important to understand the relation between indicators at a certain point in time, especially for the application in DEWS, which are focused on the current situation or the near future. As patterns of indicators depend on characteristic of the streamflow regime and water use that are temporally constant over the reference period, the reasons for similarities and differences between indicators can be deducted in any month of the reference period. March 2002 was selected as it was among the months with the highest difference between CQDI-Q80 and CQDI-Q80-HS.**

*Comparison not validation*

*RC: As proposed in this article, the fundamental purpose of any drought indicator is to represent the sectoral impacts in the best possible way. Indeed, the validation of the best indicator should be consistent in how it represent sectoral impacts. In this sense, it is perfectly legitimate to compare indicators, but it is not possible to validate or rank one over another without looking at independent variables that represent potential impacts. Please elaborate further on this issue.*

AC: We think that an impact assessment is beyond the scope of this paper (see above). However, we added this aspect as future research need as last sentence in the conclusions:

**Since an impact assessment was beyond the scope of this study, future studies could analyze how well these hazard indicators, in combination with suitable vulnerability and exposure indicators, can estimate drought impacts in the targeted risk systems at regional or national scales.**

**References**

Kumar, Amit; Gosling, Simon N.; Johnson, Matthew F.; Jones, Matthew D.; Zaherpour, Jamal; Kumar, Rohini et al. (2022): Multi-model evaluation of catchment- and global-scale hydrological model simulations of drought characteristics across eight large river catchments. In: *Advances in Water Resources* 165, S. 104212. DOI: 10.1016/j.advwatres.2022.104212.

Meza, Isabel; Eyshi Rezaei, Ehsan; Siebert, Stefan; Ghazaryan, Gohar; Nouri, Hamideh; Dubovyk, Olena et al. (2021): Drought risk for agricultural systems in South Africa: Drivers, spatial patterns, and implications for drought risk management. In: *The Science of the total environment* 799, S. 149505. DOI: 10.1016/j.scitotenv.2021.149505.

Meza, Isabel; Siebert, Stefan; Döll, Petra; Kusche, Jürgen; Herbert, Claudia; Eyshi Rezaei, Ehsan et al. (2020): Global-scale drought risk assessment for agricultural systems. In: *Nat. Hazards Earth Syst. Sci.* 20 (2), S. 695–712. DOI: 10.5194/nhess-20-695-2020.

Wan, Wenhua; Zhao, Jianshi; Popat, Eklavyya; Herbert, Claudia; Döll, Petra (2021): Analyzing the Impact of Streamflow Drought on Hydroelectricity Production: A Global-Scale Study. In: *Water Res* 57 (4). DOI: 10.1029/2020WR028087.

Wilhite, D.; Glantz, M. (1985): Understanding the drought phenomenon: the role of definitions. In: *Water International* 10 (3), S. 111–120. DOI: 10.1080/02508068508686328.

---

## Author Comment (AC2)

https://doi.org/10.5194/nhess-2022-174-RC2, 2022
Nat. Hazards Earth Syst. Sci. Discuss.

**"Analyzing the informative value of alternative hazard indicators for monitoring drought risk for human water supply and river ecosystems at the global scale"** by Claudia Herbert and Petra Döll

**Response to Anonymous Referee #2**

We thank you very much for your helpful comments and constructive suggestions for improving the manuscript. Below, each comment (in italics, indicated by "**RC**") is followed by our answer (normal font, indicated by "**AC**"). Changes in the manuscript are written in bold.

*RC: This paper analyses which drought hazard indicators are suitable for assessing and monitoring streamflow drought risks for human surface water supply and for river ecosystems. The authors recommend considering the habituation of people and ecosystems to the streamflow regime when selecting indicators. Eight existing indicators and three new indicators are proposed and evaluated using the model results of WaterGap.*

*The paper has potential but needs a proper revision before publication. The paper is very long and thereby is not always that clear/focussed and repetitive in some aspects. Overall sections can be shortened and may be ordered in a more logical way. For example, the introduction is very lengthy and lacks focus relevant to the objective of the work. Also, the methodology is at some aspects lengthy (too lengthy) whereas other aspects are not discussed at all. Altogether, from the current version of the manuscript, it is hard to judge the full potential of the research and research paper.*

> **AC**: Thank you for the constructive feedback. We deleted and shortened repetitive or too detailed paragraphs throughout the text and used more concise language. We shortened the introduction by one page and the conclusion section by almost one page. In the introduction, we deleted paragraphs and sentences that anticipated aspects of the proposed new classification system (e.g., lines 92-118) as well as repetitive and too detailed sentences (e.g., lines 141-147).

*Methods and data:*

*RC: Section 2.1: First, a minor comment, the model names WaterGAP2.2d, WGHM, WaterGAP are used for the same model I assume. However, this is a bit confusing.*

> **AC:** WaterGAP consists of the hydrological model WGHM and five water use models. WGHM is only used twice in the model description. We changed the sentence in line 157 from "The model consists of the WaterGAP Hydrological Model WGHM and five water use models […]" to "**WaterGAP** consists of the WaterGAP Hydrological Model **(**WGHM**)** and five water use models […]". Throughout the text, we now use "WaterGAP" for descriptions that apply to WaterGAP in general and the term "WaterGAP 2.2d" for version-specific descriptions.

*RC: Furthermore, previous model results are used (namely from Müller Schmied et al 2021) and all details on model description and evaluation are not discussed in this paper. However, to better understand the results it would be useful to at least read a summarized description of the for this study, most important parameters, assumptions made, uncertainties, and sensitivities and how this has, or has not, an impact on the evaluation of your drought indicators.*

> **AC**: We used model output from our (at the time of writing) latest model version 2.2d. This model version is documented in Müller Schmied et al. (2021). Since modeled streamflow in WaterGAP is the result of many processes, it is difficult to select only a few parameters that have an impact on low flows. However, we moved, as suggested by another reviewer, the section describing WaterGAP model performance in model intercomparison studies (previously, Sect. 3.4) at the end of the model description. Here, we focus on studies

that analyze streamflow, low flows, and drought hazard indicators, all of which are relevant for the present study.

In the model validation (previously, Sect. 3.4), we added the new Fig. 3 below for a more detailed analysis of Q80. Furthermore, we added the paragraph below Fig. 3 describing the results of another recent WaterGAP 2.2d model validation performed by us comprising 183 gauging stations. Hence, the model validation is now focused on Q80 and a SDHI (SSI3), which are both relevant for the global-scale analysis of streamflow drought hazard indicators. We think that this is more valuable to assess the indicators than a detailed model description.

[Figure]

**Figure 3: Percent deviations of simulated Q80 per calendar month from Q80 based on GRDC observations using the reference period 1986-2015.**

**In a recent study, WaterGAP 2.2d model output was validated against GRDC data by comparing SSI3 based on simulated and observed monthly streamflow (SSI3(sim) and SSI3(obs)) during 1971-2000 at 183 globally distributed GRDC stations (Wan et al. 2021). Applying drought hazard classes for SSI according to Agnew (2000), the agreement between simulated and observed hazard classes in each month was analyzed. Among all stations, the agreement ranged between 29 to 88% of all 360 months (their Fig. S4 and Table S3). At 68% of all stations (covering 83% of the assessed basin area), SSI3(sim) and SSI3(obs) resulted in the same drought hazard class in 70 to 88% of the time. Moreover, the goodness-of-fit was evaluated based on the Nash-Sutcliffe efficiency (NSE) for monthly streamflow and SSI3 (their Fig. S3). With a median NSE of 0.5 and an interquartile range of 0.2-0.7 for SSI3 and 0.14-0.7 for streamflow, WaterGAP 2.2d model output showed a moderate agreement with the observations. Both NSEs exceeded 0.7 at 25 out of the 183 stations, which are located in Central and Eastern Europe (twelve stations), the United States (ten stations), and South Africa (one station).**

*RC: Section 2.2: this section is very long and could be focused more on the new indicators and evaluation of the results.*

**AC:** Given the inconsistent indicator descriptions identified in the literature (or the absence of any description), we think that the detailed description of the conventional indicators is as important as the description of the new indicators.

*RC: In the method section, there is no description of how the results will be presented or evaluated/compared.*

**AC**: After dividing the "Results and Discussion" section (old Sect. 3) into two sections (see new structure below), we added an introductory paragraph at the beginning of each section (see below) explaining the objectives and the structure. Section 3 is now focused on the proposed systematic approach for selecting streamflow drought hazard indicators (SDHIs) and Sect. 4 comprises the global-scale assessment.

**3 Proposed systematic approach for selecting and computing SDHIs** (old Sect. 3.1)

        **3.1 Assumptions about habituation inherent in drought hazard indicators** (old Sect. 3.1.1)

        **3.2 Levels of drought characterization** (old Sect. 3.1.2)

        **3.3 Illustration of habituation-based classification approach** (old Sect. 3.2, new title)

**4 Quantification of global streamflow drought hazard by a global hydrological model** (old Sect. 3.3, new title)

        **4.1 Model validation** (old Sect. 3.4)

        **4.2 Discrepancies in drought hazard as quantified by different SDHIs** (old Sect. 3.3, new title)

                **4.2.1 Drought magnitude (level 1)** (old Sect. 3.3.1)

                **4.2.2 Drought severity (level 2)** (old Sect. 3.3.2)

                **4.2.3 Drought severity expressed as frequency of non-exceedance (level 2)** (old Sect. 3.3.3)

                **4.2.4 Relation between the various SDHIs** (old Sect. 3.3.4)

                **4.2.5 Suitability of SPIn to quantifiy streamflow drought hazard** (old Sect. 3.5, new title)

**AC**: We added the following paragraph at the beginning of the new Sect. 3 to clarify the objectives and the structure:

**Wilhite und Glantz (1985) suggested distinguishing between a conceptual and an operational drought definition, with the former referring to the general qualitative concept of drought and the latter allowing for a quantitative drought characterization including onset, severity, termination, and spatial extent. In the following Sect. 3.1, aspects that relate to the conceptual drought definition are discussed comprising the description of the targeted drought risk and the system at risk. In particular, assumptions about the habituation of the system at risk to the streamflow regime are discussed, an aspect that is currently not taken into account or not made explicit in drought hazard studies. In order to translate these conceptual definitions into operational drought hazard indicators, a new classification system for hazard indicators is proposed in Sect. 3.2. The new systematic approach is illustrated in Sect. 3.3 for selected SDHIs using streamflow observations at two gauging stations with different streamflow regimes.**

**AC**: We added the following paragraph at the beginning of the new Sect. 4 to clarify the objectives and the structure:

**The objective of this chapter is to identify which of the SDHIs presented in Table 1 can be meaningfully quantified at the global scale using WaterGAP 2.2d and which SDHIs are appropriate for monitoring different drought risks in large-scale DEWS. After a limited validation of modeled streamflow (Sect. 4.1), SDHIs of drought magnitude and severity are compared separately (Sect. 4.2.1-4.2.3) following the classification system presented in Fig. 1. The SDHIs are shown in global maps for a selected month (March 2002), as it is important to understand the relation between indicators at a certain point in time, especially for the application in DEWS, which are focussed on the current situation or the near future. As patterns of indicators depend on characteristic of the streamflow regime and water use that are temporally constant over the reference period, the reasons for similarities and differences between indicators can be deducted in any month of the reference period. March 2002 was selected as it was among the months with the highest difference between CQDI-Q80 and CQDI-Q80-HS. In addition to the analysis for the selected time step, the latter two indicators are compared at the global scale with respect to drought occurrence during the whole reference period. Discrepancies and similarities of the indicators are discussed in more detail for two illustrative grid cells with the same CQDI-Q80 value in March 2002 (Sect. 4.2.4). Finally, the suitability of SPI with different averaging periods to estimate streamflow drought hazard is assessed using**

streamflow observations from 218 GRDC gauging stations (Sect. 4.2.5). Based on this global-scale analysis and the proposed habituation-based classification approach, selected SDHIs are recommended for implementation in large-scale DEWS (Sect. 5).

*Figures:*

*RC: It is not clear why the two grid cells were chosen. Also, it is not clear why March 2002 was chosen.*

AC: With respect to the two grid cells, we rephrased the first sentence in the new Sect. 4.2.4:

**The relation between selected SDHIs from Figs. 4, 5 and 7 is compared for two illustrative grid cells that share the same CQDI-Q80 value of 0.04 in March 2002 (Table 2).**

AC: With respect to the selected month March 2002, we added the following sentences at the beginning of the new Sect. 4:

**The SDHIs are shown in global maps for a selected month (March 2002), as it is important to understand the relation between indicators at a certain point in time, especially for the application in DEWS, which are focused on the current situation or the near future. As patterns of indicators depend on characteristic of the streamflow regime and water use that are temporally constant over the reference period, the reasons for similarities and differences between indicators can be deducted in any month of the reference period. March 2002 was selected as it was among the months with the highest difference between CQDI-Q80 and CQDI-Q80-HS.**

*RC: In the current manuscript legends and labels are off for e.g. Fig 3 (what are the lower and upper labels in the left legend, and the upper label in the right); Fig 4, placement of '0' and what is meant with '0 >0'; what comes after '1'; Fig 5, placement of '0'; etc.*

AC: The legends are correct. To make the figures better comprehensible, we extended the figure captions, e.g. by explaining for one or two example the meaning of different colors. We did this for the new Figs. 4 and 5 (old Figs. 3 and 4).

Figure caption of the new figure 5 (old Fig. 4) (new text is underlined):

**Figure 5: Severity of drought hazard (level 2 in Fig. 1): Cumulative deficit in March 2002 since onset of drought event as indicated by CQDI1-Q80 (a), CQDI1-WUs (b), CQDI1-Q50 (c), and CQDI1-WUs-EFR (d) for the reference period 1986-2015. A deficit of zero is shown in beige. Values larger than zero and below 0.1 are shown in green. A value of 0.1, for example, denotes that the current cumulative deficit is equivalent to 10% of mean annual streamflow (MAQ). WUs: mean annual surface water withdrawals.**

However, in Fig. 3c (now Fig. 4c), we erroneously did not show the blue grid cells. We corrected the figure (see below together with the extended figure caption where the new text is underlined).

[Figure]

**Figure 4: Magnitude of drought hazard (level 1 in Fig. 1): Non-cumulative anomaly in March 2002 as indicated by SSI1 (a), RQDI1 (b), EP1 (c), SSI12 (d), SPI12 (e) and SPEI12 (f) for the reference period 1986-2015. For the standardized indicators and EP1, the z scores and the corresponding frequencies of non-exceedance and return periods are shown. In the blue grid cells in (c), drought identification is not possible with EP1, since Q80 and Q are zero. "nc": not computable. The two grid cells from Table 2 are marked in (a) (northern Italy and central Paraguay).**

**References**

Agnew, C. T. (2000): Using the SPI to Identify Drought.

Wan, Wenhua; Zhao, Jianshi; Popat, Eklavyya; Herbert, Claudia; Döll, Petra (2021): Analyzing the Impact of Streamflow Drought on Hydroelectricity Production: A Global-Scale Study. In: *Water Res* 57 (4). DOI: 10.1029/2020WR028087.

Wilhite, D.; Glantz, M. (1985): Understanding the drought phenomenon: the role of definitions. In: *Water International* 10 (3), S. 111–120. DOI: 10.1080/02508068508686328.

---

## Author Comment (AC3)

https://doi.org/10.5194/nhess-2022-174-RC3, 2022
Nat. Hazards Earth Syst. Sci. Discuss.

**"Analyzing the informative value of alternative hazard indicators for monitoring drought risk for human water supply and river ecosystems at the global scale"** by Claudia Herbert and Petra Döll

**Response to Anonymous Referee #3**

We thank you very much for your thorough review and constructive suggestions for improving the manuscript. Below, each comment (in italics, indicated by "**RC**") is followed by our answer (normal font, indicated by "**AC**"). Changes in the manuscript are written in bold. We first respond to the comments in the main document and second to the comments in the supplement.

**1) Response to comments in the main document**

**RC:** *This study proposes a new classification of streamflow drought hazard indicators (SDHI) to guide researchers or practitioners in selecting an appropriate indicator(s) for their objective(s). In order to demonstrate the importance of the proper selection of SDHIs in a drought assessment, the authors compared eleven SDHIs, including three new indicators, and quantified their similarities and discrepancies. Because human society and ecosystems adapt to a local streamflow regime, this study underscores their habituation implicitly assumed in each SDHI. This includes how one can define a normal condition and a condition under a lack of water. In this context, the results show that drought severity, i.e., accumulated drought magnitude, rather than mere magnitude at a time step, may need to be evaluated to investigate prolonged drought events. The dry season in the arid region, which is not suitable for a conventional threshold method, and the water deficit in comparison to water demand are also paid particular attention in this study. Classifying SDHIs and organizing their features, as a result, the authors stress that one needs to select an appropriate SDHI(s) in view of their assumption(s) and operational drought early warning systems need to cover a wide range of DHIs to support a widespread users'
demand.*

*The paper provides several important insights regarding SDHIs. Because a better understanding of the application and interpretation of DHIs is a crucial challenge, the results are valuable for the drought research community and practitioners. However, I have some major concerns that the authors need to address during this discussion phase.*

> **AC**: Thank you very much for the positive feedback.

**Structure of the manuscript**

**RC**: *I assume the authors have a certain intention, but I would say that the current structure of the manuscript is not so reasonable to me. Due to the structure, it was difficult for me to read the manuscript and grasp this study's point and novelty. Therefore, the author needs to reconsider the design of the storyline. The followings are my suggestions:*

*I recommend the authors move Section 3.1 to Method (or partly, Introduction) because this section describes (i) debatable points, difficulties, or caveats in selecting/applying SDHIs, (ii) characteristics of each SDHI that serve as premises for subsequent analysis/discussion, and (iii) the definitions of key terms (e.g., magnitude/severity, conceptual/operational, etc.). Moreover, this section includes relatively general contents, some of which were mentioned in the Introduction, and refers to many preceding studies. Probably, the authors aim to present the classification as a new result, but I would say that it should be reasonable to present the classification as a new classification "method" in Chapter2 and demonstrate the needs and validity of the classification in the Result Chapter.*

> **AC:** We think that the proposed systematic approach for selecting SDHIs is a novel methodological approach and can stand alone as a result. While it is build on existing knowledge (indicator definitions, advantages, disadvantages), it incorporates new perspectives on hazard indicators with respect to habituation and allows for a consistent definition and classification of indicators that it currently missing in the literature. We agree, however, that the structure can be improved and changed the structure of the results and discussion section (see answer to the next referee comment). In order to clarify the study's point and novelty, we added several

sentences and paragraphs in different chapters (see below) highlighting research gaps as well as the study's objectives.

**AC**: In the introduction, we deleted paragraphs and sentences that anticipated aspects of the proposed new classification system (e.g., lines 92-118) as well as repetitive and too detailed sentences (e.g., lines 141-147). Overall, we shortened the introduction by one page.

**AC**: In the introduction, we added to following sentence (in bold) to highlight the research gap with respect to drought hazard concepts:

[Clearly, the conception or selection of hazard indicators needs to take into account the habituation and thus vulnerability of the system at risk.] **However, investigations and guidance on how to select the optimal SDHI, considering both the targeted risk and the habituation of the system at risk to the streamflow regime, are missing.**

**AC**: In the introduction, we added to following sentence to highlight the research gap with respect to a hazard indicator classification:

**Certainly, an improved classification of drought hazard indicators would facilitate a better understanding of drought characteristics and provide guidance in selecting appropriate drought hazard indicators.**

*RC: In conjunction with the previous comment, I would recommend the authors modify the structure of Chapter4 and rename Chapters 3 and 4 as Results and Discussion. This is because the paragraphs in P. 37 and 38 in Chapter4 give a good summary of the results. Although the authors describe the essence of a comparison at the end of each comparison in Chapter 3, these key lines are fragmented in the Chapter. Thus, the summary and interpretation of the results fit the best right after Chapter3 (= at the beginning of Chapter4) with Table 3. I would title the summary section "systematic approach for selecting streamflow drought hazard indicators", and the Recommendation could be section 4.2.*

**AC**: The "Results and Discussion" (old Sect. 3) are now divided into Sect. 3 and 4, with Sect. 3 focused on the proposed systematic approach for selecting streamflow drought hazard indicators (SDHIs) and Sect. 4 comprising the global-scale assessment (see structure below). Some of the titles were renamed to better reflect the objectives of the sections. We think that it is not possible to strictly separate results and discussion in this manuscript as it is primarily a methodological study where the discussion of methods is part of the result. However, with the new structure and the new introductory paragraphs in each section (see below), we think that the storyline is improved and the objectives are clarified. We would like to leave the recommendations (new Sect. 5) as a summary of the systematic approach and the global-scale assessment after the new Sect. 4, followed by the conclusions in the new Sect. 6.

**3 Proposed systematic approach for selecting and computing SDHIs** (old Sect. 3.1)

    **3.1 Assumptions about habituation inherent in drought hazard indicators** (old Sect. 3.1.1)

    **3.2 Levels of drought characterization** (old Sect. 3.1.2)

    **3.3 Illustration of habituation-based classification approach** (old Sect. 3.2, new title)

**4 Quantification of global streamflow drought hazard by a global hydrological model** (old Sect. 3.3, new title)

    **4.1 Model validation** (old Sect. 3.4)

    **4.2 Discrepancies in drought hazard as quantified by different SDHIs** (old Sect. 3.3, new title)

        **4.2.1 Drought magnitude (level 1)** (old Sect. 3.3.1)

AC: We added the following paragraph at the beginning of the new Sect. 3 to clarify the objectives and the structure:

**Wilhite und Glantz (1985) suggested distinguishing between a conceptual and an operational drought definition, with the former referring to the general qualitative concept of drought and the latter allowing for a quantitative drought characterization including onset, severity, termination, and spatial extent. In the following Sect. 3.1, aspects that relate to the conceptual drought definition are discussed comprising the description of the targeted drought risk and the system at risk. In particular, assumptions about the habituation of the system at risk to the streamflow regime are discussed, an aspect that is currently not taken into account or not made explicit in drought hazard studies. In order to translate these conceptual definitions into operational drought hazard indicators, a new classification system for hazard indicators is proposed in Sect. 3.2. The new systematic approach is illustrated in Sect. 3.3 for selected SDHIs using streamflow observations at two gauging stations with different streamflow regimes.**

AC: We added the following paragraph at the beginning of the new Sect. 4 to clarify the objectives and the structure:

**The objective of this chapter is to identify which of the SDHIs presented in Table 1 can be meaningfully quantified at the global scale using WaterGAP 2.2d and which SDHIs are appropriate for monitoring different drought risks in large-scale DEWS. After a limited validation of modeled streamflow (Sect. 4.1), SDHIs of drought magnitude and severity are compared separately (Sect. 4.2.1-4.2.3) following the classification system presented in Fig. 1. The SDHIs are shown in global maps for a selected month (March 2002), as it is important to understand the relation between indicators at a certain point in time, especially for the application in DEWS, which are focussed on the current situation or the near future. As patterns of indicators depend on characteristic of the streamflow regime and water use that are temporally constant over the reference period, the reasons for similarities and differences between indicators can be deducted in any month of the reference period. March 2002 was selected as it was among the months with the highest difference between CQDI-Q80 and CQDI-Q80-HS. In addition to the analysis for the selected time step, the latter two indicators are compared at the global scale with respect to drought occurrence during the whole reference period. Discrepancies and similarities of the indicators are discussed in more detail for two illustrative grid cells with the same CQDI-Q80 value in March 2002 (Sect. 4.2.4). Finally, the suitability of SPI with different averaging periods to estimate streamflow drought hazard is assessed using streamflow observations from 218 GRDC gauging stations (Sect. 4.2.5). Based on this global-scale analysis and the proposed habituation-based classification approach, selected SDHIs are recommended for implementation in large-scale DEWS (Sect. 5).**

*RC: The position of the description in Model validation is not reasonable (section 3.4) #1; the 1st paragraph of the section has to be in section 2.1 because these are not results of this study and are justifications for the use of WaterGAP.*

AC: We moved the first paragraph to the end of section 2.1. Furthermore, we inserted an introductory sentence to this paragraph followed by an additional, recently published study:

**In several model intercomparison studies, WaterGAP was often among the best performing global hydrological models (GHMs). Kumar et al. (2022) assessed the ability of nine catchment-scale models and eight**

GHMs to simulate hydrological droughts in eight large catchments around the world. Comparing simulated and observed streamflow deficits and SSI1 (SRI) (their Tables 2 and 3), WaterGAP is among the two to three best performing GHMs with performance indicators (R² and Nash-Sutcliffe efficiency) comparable to those of the catchment-scale models.

*RC: The position of the description in Model validation is not reasonable (section 3.4) #2; the validation results (the 2nd paragraph of section 3.4) have to be presented before the model-based analyses (i.e., section 3.3).*

**AC**: We moved this paragraph before the model-based analyses (new Sect. 4.1, see above).

**Model validation**

*RC: I require the authors to present a more description on the model validation concerning stream drought reproducibility. Currently, no figure has been presented for section 3.4. Also, I expect an additional analysis to evaluate how well the simulated stream flow data reproduce hydrological drought events detected by the observation data. I would say that, at least, the comparison of observation- and simulation-based drought detections at the selected two gauge stations should be presented similarly to Figure 2.*

**AC**: In the model validation (new Sect. 4.1), we added the new Fig. 3 below for a more detailed analysis of Q80. Furthermore, we added the paragraph below Fig. 3 describing the results of another recent WaterGAP 2.2d model validation performed by us comprising 183 gauging stations. Hence, the model validation is now focused on Q80 and a SDHI (SSI3), which are both relevant for the subsequent global-scale analysis of SDHIs.

[Figure]

**Figure 3: Percent deviations of simulated Q80 per calendar month from Q80 based on GRDC observations using the reference period 1986-2015.**

In a recent study, WaterGAP 2.2d model output was validated against GRDC data by comparing SSI3 based on simulated and observed monthly streamflow (SSI3(sim) and SSI3(obs)) during 1971-2000 at 183 globally distributed GRDC stations (Wan et al. 2021). Applying drought hazard classes for SSI according to Agnew (2000), the agreement between simulated and observed hazard classes in each month was analyzed. Among all stations, the agreement ranged between 29 to 88% of all 360 months (their Fig. S4 and Table S3). At 68% of all stations (covering 83% of the assessed basin area), SSI3(sim) and SSI3(obs) resulted in the same drought hazard class in 70 to 88% of the time. Moreover, the goodness-of-fit was evaluated based on the Nash-Sutcliffe efficiency (NSE) for monthly streamflow and SSI3 (their Fig. S3). With a median NSE of 0.5 and an interquartile range of 0.2-0.7 for SSI3 and 0.14-0.7 for streamflow, WaterGAP 2.2d model output showed a moderate agreement with the observations. Both NSEs exceeded 0.7 at 25 out of the 183 stations, which are located in Central and Eastern Europe (twelve stations), the United States (ten stations), and South Africa (one station).

*The selected two gauge stations*

*RC: The basin names and their characteristics (lines 507-510, lines 516-517) should be described right after the first sentence of section 3.2 in order to explain the phrase "at two GRDC gauge stations with different streamflow re-gimes".*

    **AC**: We implemented the suggestion.

*March 2002*

*RC: I am curious about why March 2002 was selected as an example in this manuscript. An additional short line is expected in this regard. The authors need to discuss the generality of the results in Section3.3 for March 2002, alt-hough I assume that similarities and discrepancies among SDHIs are similar for other months and years too.*

    **AC**: With respect to the selected month March 2002, we added the following sentences at the beginning of the new Sect. 4:

    **The SDHIs are shown in global maps for a selected month (March 2002), as it is important to understand the relation between indicators at a certain point in time, especially for the application in DEWS, which are focused on the current situation or the near future. As patterns of indicators depend on characteristic of the streamflow regime and water use that are temporally constant over the reference period, the reasons for similarities and differences between indicators can be deducted in any month of the reference period. March 2002 was selected as it was among the months with the highest difference between CQDI-Q80 and CQDI-Q80-HS.**

*Other*

*RC: The writing often lacks sharpness. It includes many restated phrases (for example, 21 "i.e." in total) and repetition (e.g., the authors repeatedly stress that severity is accumulated value.). This manuscript can be more concise to deliver the authors' messages to readers.*

    **AC:** We deleted and shortened repetitive or too detailed paragraphs throughout the text and used more concise language. We shortened the introduction by one page and the conclusion section by almost one page. Severity is now explained in the beginning and in figure captions only. "i.e." is left six times in the text.

**2) Response to comments in the supplement**

    **AC:** All minor comments and suggestions in the supplement were implemented to the letter. These include the comments in lines 17, 78, 141, 158, 280, 368, 370, 434, 440, 441, 460, 461, 463, 494, 501, and 654. All other comments and suggestions are addressed in the following.

**RC (line 15)***: The authors intentionally distinguish deficit and anomaly throughout the text. Is anomaly not included here? There seems to be some similar inconsistency regarding deficit/anomaly in the manuscript.*

    **AC:** We admit that it is difficult to strictly distinguish anomaly and deficit. Like in other studies, we also use the term "deficit" for the volume-based anomaly below a certain threshold (CQDI indicators). Setting a threshold implies that the volume below this threshold is perceived as a deficit by the risk bearer, although this event can occur in a month without any human water demand. Regarding the comment in the abstract, we replaced "water deficit" by "**e.g., water anomaly**" [during a pre-defined period].

**RC (line 128)***: If van Huijgevoort et al. 2012 is important, CDPM needs to be explained more.*

    **AC:** The CDPM method in van Huijgevoort et al. (2012) is certainly valuable and sophisticated. However, for large-scale DEWS requiring transparent indicators, we concluded that the method is too complex and partly

not intuitive (see lines 131-133). As we do not use the concept and given the length of the manuscript, we feel that our description of the method is sufficient.

**RC (line 202)**: *It should be good to briefly explain why this test is necessary in this fitting.*

**AC:** We changed the text as follows:

**The goodness-of-fit between simulated streamflow values and the probability distribution was assessed based on the one-sample Kolmogorov–Smirnov test (KS test) at the 0.05 significance level.**

**RC (line 206)**: *CQDI seems to be correct if its full name is cumulative streamflow deficit indicator.*

**AC:** We implemented the suggestion. To be consistent, we also change RDQI to RQDI.

**RC (line 221)**: *A reference is required.*

**AC:** Spinoni et al. (2019) who also used the 2 months criterion that we applied wrote "With the two months criterion we excluded the very short droughts (one month only), which are rarely impacting (differently than flash floods) and could outstandingly increase the number of events of low importance in the database.". Therefore, we added as reference to the sentence:

**This approach avoids that short-term streamflow deficits that hardly pose a drought hazard to humans and other biota are defined as drought events (Spinoni et al., 2019).**

**RC (line 232)**: *It this the right position for this sentence?*

**AC:** We deleted the sentence.

**RC (line 341-347)**: *It is unclear why the authors mention the first and the third sentences here, while the second sentence seems to be relevant to Q50 indicators.*

**AC:** We agree with the reviewer that the third sentence (starting with "To quantify risk, …") is superfluous and rather distracting as the whole section is about drought hazard indicators that take into account habituation of people and ecosystems, and not about drought risk. Therefore, we deleted the third sentence. Regarding the discussion of indicating drought hazard with relative deviations from mean conditions as compared to the rarity of a drought condition (as people in particular in the dry regions of the globe might not be adapted to the high interannual variability there but rather to mean conditions), we decided to shorten and rewrite the text starting in line 335 to make it more easily understood, more general and more concise:

Old: Kumar et al. (2009) compared percent precipitation deviations and SPI in two districts in India with high and low precipitation. Based on a 39-year record of observed monthly precipitation they showed that during the monsoon months where precipitation is decisive for crop production, much higher percent precipitation deviations occurred in the low precipitation district than in the high precipitation district, e.g-70% and -30%, respectively, in case of SPI = -1. They found that due to the need to fit a function to the actual precipitation data to determine SPI and thus probability of occurrence, a year with a lower precipitation might be indicated by the SPI as being less dry than a year with a relatively higher precipitation. Consequently, severe drought may be underestimated with SPI in particular in the low precipitation district due to non-normality of the distribution for extremely low precipitation values. More importantly, considering the risk for rainfed crop production, yield loss is more closely related to percent of normal precipitation than to the rarity of the low precipitation event, as crop yield depends on actual evapotranspiration in percent of PET, which decreases with precipitation (Siebert and Döll, 2010). Yield loss due to 30% less precipitation than normal can be expected to be much smaller than yield loss due to 70% less, such that percent deviation from the mean can be a good hazard indicator for assessing drought risk for rainfed crop production in these two districts.

New: **Kumar et al. (2009) compared percent precipitation deviations and SPI in two districts in India, one in a humid region with high mean precipitation and low interannual variability and the other in a semi-arid region with low mean precipitation and higher interannual variability. Based on a 39-year record of observed monthly precipitation they showed that much higher percent deviations occurred, in the case of SPI = -1, in the low precipitation district than in the high precipitation district, e.g., -70% and -30%, respectively. Consequently, drought hazard may be underestimated with SPI in the low precipitation district. For example, yield loss is more closely related to percent of normal precipitation than to the rarity of the low precipitation event, as crop yield depends on actual evapotranspiration in percent of PET, which decreases with precipitation (Siebert and Döll, 2010).**

**RC (line 351)**: *longer*

    **AC:** The whole sentence was rephrased:

    **Similar to the example above from Kumar et al. (2009), the 20th streamflow percentile (or SSI1 = -0.84) would correspond to a low relative streamflow deviation (e.g. -20%) in a humid region (low interannual variability) compared to a higher deviation (e.g. -50%) in a semi-arid region (high interannual variability).**

**RC (line 359)**: *A reference is required.*

    **AC:** We think that the statement "Contrastingly, river ecosystems are, in the ideal case, perfectly adjusted to interannual variability of streamflow" is of general nature and does not require a reference.

**RC (line 390)**: *I am afraid I could not understand this sentence. Would you explain more with any references?*

    **AC:** We changed the sequence of the sentences and added one sentence (underlined below). A reference does not exist, since this statement is a result from our methodological analysis. The paragraph now reads:

    **Obviously, if SPI12 and SPEI12 are used to assess meteorological drought hazard, people and ecosystems are assumed to be habituated to the interannual variability, but not the seasonality, of P and P-PET. However, when they are used as proxies to identify streamflow drought hazard, they should ideally correspond to the temporal development of SSI1. Their performance would be assessed by comparing the goodness-of-fit between time series of SPI12 or SPEI12 with SSI1. In this case, assumptions about the habituation inherent in SPI12 and SPEI12 refer to the streamflow regime and not to time series of P and P-PET. Accordingly, SPI12 and SPEI12 fall into the same category as SSI1 in Table 1 (interannual variability and seasonality). The suitability of different averaging periods for SPI for describing streamflow drought hazard is discussed in Sect. 4.2.5.**

**RC (line 440)**: *What is QDAI?*

    **AC:** We added "**the streamflow deficit anomaly indicator**" as explanation.

**RC (line 448)**: *Because of the section title, a schematic that includes systematic selection processes may be expected by readers in this section.*

    **AC:** We replaced "schematic" by "**classification system**".

**RC (line 468)**: *The similar is already explained in section 2.2.3 so that this seems to be redundant*

    **AC:** Most of the paragraph was deleted (lines 463-473).

**RC (line 478)**: *(end of the sentence is marked by the referee)*

    **AC:** The sentence was shortened and now reads: "**Furthermore, the classification system clarifies that each indicator type requires a threshold setting either at level 1 or 2.**"

**RC (line 512)**: *what is w/o?*

    **AC:** "w/o" was replaced by "without" in the text and in Fig. 2.

**RC (line 514)**: *Is this sentence necessary? If so, this should be explained in the previous sentence.*

    **AC:** The marked sentence was deleted.

**RC (line 551)**: *Would you elaborate the process behind?*

    **AC:** To keep the description concise, we only inserted one sentence regarding drought propagation and re-phrased the end of the paragraph. The text section now reads:

    The correspondence between the meteorological indicators SPI12 and SPEI12 (Fig. 2c) and the hydrological indicators (SSI1 or EP1 and CQDI1-Q80 variants) is low at both stations. For most streamflow drought events, the averaging period of 12 months for the meteorological variables leads to excessive delays in the signal. Many short drought signals are not detected at all. Performance of SPI12 and SPEI12 is equally low at both stations. **Hence, drought propagation through the hydrological cycle is faster than estimated by SPI12 and SPEI12. This is also supported by the sensitivity analysis of SPI averaging periods in Sect. 4.2.5. At both stations in Fig. 2, an averaging period of 3 months resulted in the highest correlation between SPI and observed SSI1.**

**RC (line 580-584)**: *This section is about the case study about the two gauge stations. I would say that this sentence does not fit this section.*

    **AC:** With the new title of this section (Illustration of habituation-based classification approach), we clarify that the two examples are also used to gain general knowledge about the suitability of indicators. Therefore, we feel that this section can remain in the text in order to put the results in a wider context.

**RC (line 597)**: *This restatement seems redundant.*

    **AC:** The statement was deleted.

**RC (line 632)**: *This sentence seems inconsistent with L.551.*

    **AC:** The sentence was changed to: "SPI12 and SPEI12 (Figs. 3c and e) **are sometimes** used as proxies […]".

**RC (line 707)**: *The authors mention to QDAI several times within this manuscript. Why this indicator is not included in this analysis?*

> **AC:** QDAI was computed with the same WaterGAP model version 2.2d in a recent study from 2021, and it has been analyzed at the global scale. In contrast, the drought hazard indicators in the present study have not been computed with WaterGAP at the global scale. Hence, QDAI was not included to avoid redundancy with the study of Popat and Döll (2021).

**RC (line 731)**: *This is not explained in the text.*

> **AC:** After the first sentence of this section, we added **"The indicators are denoted with the suffix "f" for frequency."**

**RC (line 1019-1022)**: *These two sentences look inconsistent.*

> **AC:** We deleted the whole paragraph and added a summary of the model validation:

> **A limited validation exercise revealed a tendency of WaterGAP 2.2d to overestimate observed Q80 between October and April, while Q80 during the low-flow period in the Northern Hemisphere (May to September) is better captured. In a recent study comparing simulated and observed monthly streamflow and SSI3 at 183 stations worldwide, WaterGAP 2.2d model output showed a moderate agreement with observations. However, high agreement was identified at 25 stations in Central and Eastern Europe, the United States, and South Africa. Moreover, the current study showed that SSI1 based on modeled streamflow often outperformed SPI with different averaging periods. However, this strongly depends on the goodness-of-fit after streamflow calibration. In uncalibrated basins, meteorological drought indicators should be used complementarily as proxies for hydrological drought hazard due to the uncertainty of modeled streamflow.**

**References**

Agnew, C. T. (2000): Using the SPI to Identify Drought.

Kumar, Amit; Gosling, Simon N.; Johnson, Matthew F.; Jones, Matthew D.; Zaherpour, Jamal; Kumar, Rohini et al. (2022): Multi-model evaluation of catchment- and global-scale hydrological model simulations of drought characteristics across eight large river catchments. In: *Advances in Water Resources* 165, S. 104212. DOI: 10.1016/j.advwatres.2022.104212.

Kumar, N. M.; Murthy, C. S.; Sesha Sai, M. V. R.; Roy, P. S. (2009): On the use of Standardized Precipitation Index (SPI) for drought intensity assessment. In: *Met. Apps* 16 (3), S. 381–389. DOI: 10.1002/met.136.

Siebert, Stefan; Döll, Petra (2010): Quantifying blue and green virtual water contents in global crop production as well as potential production losses without irrigation. In: *Journal of Hydrology* 384 (3-4), S. 198–217. DOI: 10.1016/j.jhydrol.2009.07.031.

Wan, Wenhua; Zhao, Jianshi; Popat, Eklavyya; Herbert, Claudia; Döll, Petra (2021): Analyzing the Impact of Streamflow Drought on Hydroelectricity Production: A Global-Scale Study. In: *Water Res* 57 (4). DOI: 10.1029/2020WR028087.

Wilhite, D.; Glantz, M. (1985): Understanding the drought phenomenon: the role of definitions. In: *Water International* 10 (3), S. 111–120. DOI: 10.1080/02508068508686328.

---

## Author Response (AR2)

https://doi.org/10.5194/nhess-2022-174
Nat. Hazards Earth Syst. Sci. Discuss.

**"Analyzing the informative value of alternative hazard indicators for monitoring drought risk for human water supply and river ecosystems at the global scale"** by Claudia Herbert and Petra Döll

**Response to Editor**

Dear Ms. Van Loon,

We are very grateful for the many helpful, informative comments and the constructive and very detailed suggestions that you have provided. We thank you very much for your time and effort to help us improve the manuscript.
Below, each of your comments (in italics, indicated by "**EC**") is followed by our answer (normal font, indicated by "**AC**"). Changes in the manuscript are written in bold.

*EC: Thanks for the revised version of the paper. I appreciate the effort and also the reviewer who looked at the paper again sees improvement. However, some issues indicated by the reviewers were not addressed sufficiently and the new reviewer suggests to reject the paper because they find it unreadable. I still want to give you another opportunity to improve the paper, because I see value in the classification of indicators that you are proposing and I think there are some interesting findings in the paper. But I do need you to more thoroughly re-work the manuscript. Otherwise I will not be able to accept the paper for publication in NHESS.*

> **AC**: Thank you for the positive feedback. We shortened the manuscript by ten pages, rephrased many sections, and restructured the results section following your first suggestion (see below).

*EC: One option is to move a lot of material to supplementary material, for example Section 3.3, 4.1 and 4.2.5, and to focus the paper much more on the core message and its illustration. A suggestion then could be to restructure the paper so that the illustrations better fit the message, in a way that you make a few subsection that each have a different message based on a comparison of different indicators and in each you use a different visualization (a global map at a specific point in time or a time series in a selected grid cell of a selection of indicators) to show the point you are making in that subsection.*

> **AC**: We moved part of Sect. 2.1 (suitability of WaterGAP to model drought hazard indicators) and Section 4.1 (model validation) to the supplement. We deleted large parts of Sect. 3.3 and we provide results for the (new) gauging stations in the supplement. We excluded the indicators SPI and SPEI from our analysis and deleted Sect. 4.2.5 (Suitability of SPIn to quantifiy streamflow drought hazard).

> Sect. 3 now only describes the theoretical basis for the suggested classification system, which is then exemplified in Sect. 4 using a) modeled output from WaterGAP and b) observed monthly streamflow at four selected gauging stations. As suggested, Sect. 4 is now divided into subsections with different core messages illustrated using different figures.

*EC: Another option is to divide the paper in two separate ones. One opinion paper on habituation and introducing the classification (Section 3.1 and 3.2). And one research paper on a global comparison of hazard indicators (referring to the suggested classification). In the latter paper it would then be good to show more maps of other droughts years, for example selecting months that had drought hotspots in each of the continents (so one selected month that was a drought hotspot in South America, one in North America, one in Africa, etc.), because processes and therefore indicators can be very different.*

> **AC**: We now illustrate the core messages using four gauging stations as well as modeling results of two selected months of the reference period. We describe the approach at the beginning of Sect. 4 as follows:

> **The objective of this chapter is to identify which of the SDHIs presented in Table 1 can be meaningfully quantified at the global scale using WaterGAP 2.2d and which SDHIs are appropriate for monitoring**

different drought risks in large-scale DEWS. We emphasize that the objective is not a drought impact assessment, which is beyond the scope of this study. We want to show how the conceptual discrepancies and similarities between SDHIs (Sect. 3), which are of a general nature and apply to any month of the reference period, are translated into global-scale hazard indicators and how these indicators should be interpreted by end-users of a large-scale DEWS. The indicators are illustrated in global maps for two example months, July 2003 and July 2015, with known drought events in Europe. Following the classification of Table 1, SDHIs indicate drought magnitude (Figs. 2 and S3) or drought severity, the latter either expressed as volume-based anomaly or deficit (Figs. 3 and S4) or as frequency of non-exceedance (denoted with the suffix "_f") (Figs. 5 and S5). In addition, CQDI1(Q80) and CQDI1(Q80-HS) are compared at the global scale with respect to drought occurrence during the whole reference period (Fig. 4). SDHIs are further illustrated for four selected gauging stations with different streamflow regimes and assumed vulnerabilities of the risk system to streamflow anomalies (Figs. 6, S2 and S6). These include two stations with low interannual streamflow variability (Danube River at Hofkirchen, Germany (probably low vulnerability), and Angara River at Boguchany, Russia (possibly higher vulnerability)) and two stations with high interannual variability (White River near Oacoma, U.S. (probably low vulnerability), and Orange River at Vioolsdrif, South Africa (possibly higher vulnerability)).

*EC: Please consider a thorough restructuring of the paper(s) and also address the reviewers comments. And also clarify these additional point from my side:*
*- There still is some confusing about the methodology (see reviewer #1).*

> **AC**: The description of SPI and SPEI was deleted, since the indicators are not analyzed any more.

*EC: - There are some explanations of methods or methodological choices in the Results sections, for example, what data was used to calculate the indicators (l.119-120: "Hydrological drought hazard indicators were computed using output from the global water availability and water use model WaterGAP2.2d" and l.470: "We used observations instead of WaterGAP modelling result to exclude model uncertainties."). Clarify this in the Methods section and add "in this subsection" to l.470.*

> **AC**: We added a new chapter at the beginning of the methods section, which describes the streamflow data used in this study (streamflow observations and modeled streamflow). The section with line 470 was deleted.

*EC: - l. 503: "Performance of SPI12 and SPEI12 is equally low" > Avoid phrasing like high or low performance, realistic, good, etc. throughout the manuscript since you are not comparing to objective drought observations or impacts that can tell you which indicator is best. Your point is that they are different and should be picked carefully because they represent different assumptions about habituation, so just discuss the implications of choosing a different indicator.*

> **AC**: We replaced these phrases throughout the manuscript. The comparison of SDHIs with SPI and SPEI was deleted due to the length of the manuscript.

*EC: - Reservoirs and SSI12 / CQDI1-Q80-HS: WaterGAP has reservoirs included, so why accumulate streamflow and why not take into account actual flow in dry areas? If the model would simulate natural flow, then it would make sense to use SSI12 in regions where water is stored in reservoirs, and to calculate CQDI1-Q80-HS ("drought to continue in any month where Q80 is zero also if the current streamflow Q exceeds zero"), but since WaterGAP already has reservoirs, aren't you double counting?*

> **AC**: Thank you for pointing it out. We addressed this issue in the recommendations for reservoir managers in Sect. 5 and the description for Table 2 (previously Table 3):
>
> Line 613: [**] **Reservoir managers should be informed to consider SDHIs of the grid cell that represent inflow into the reservoir.**

Line 667: **Importantly, reservoir managers should only consider SDHIs of the grid cell that represent inflow into the reservoir.**

*EC: - The selection of grid cells and month: you now say: "March 2002 was selected as it was among the months with the highest difference between CQDI-Q80 and CQDI-Q80-HS" & "two illustrative grid cells with the same CQDI-Q80 value in March 2002" >> These are not valid justifications. To avoid this problem, I would suggest to add more months and locations (either in the paper if you split the paper in two separate ones, or in Supplementary material), for example the months with the maximum drought in each of the continents.*

**AC**: We now compare July 2003 and July 2015 and explain the approach at the beginning of Sect. 4 (**The objective of this chapter is to identify […]**) (see paragraph in bold on page 1 and 2 of this document). The results for July 2015 are provided in the supplement.

The comparison of the "two illustrative grid cells" was deleted due to the length of the manuscript.

*EC: - l.784-799: also here there is methods justification in the Results section. Please move.*

**AC**: The whole section was deleted.

*EC: - Section 4.2.5: You calculated correlation between simulated SSI and different SPIn with observed SSI for the basins in which the model was also calibrated. > You should have used other observed stations for validation. Please change and/or move to Supplementary material.*

**AC**: The whole section was deleted.

https://doi.org/10.5194/nhess-2022-174
Nat. Hazards Earth Syst. Sci. Discuss.

**"Analyzing the informative value of alternative hazard indicators for monitoring drought risk for human water supply and river ecosystems at the global scale"** by Claudia Herbert and Petra Döll

**Response to Anonymous Referee #1, report 2**

We thank you very much for your helpful comments and constructive suggestions for improving the manuscript. Below, each comment (in italics, indicated by "**RC**") is followed by our answer (normal font, indicated by "**AC**"). Changes in the manuscript are written in bold.

*RC: I thank the authors for their efforts made to improve the manuscript. However, I think that for the sake of clarity and to increase the impact of the article, it could still be reduced and simplified a little. Below some general points that need to be further clarified.*

> **AC**: We shortened the manuscript by ten pages and restructured and rephrased many sections to improve the readability and to clearly communicate the core messages and recommendations. Many sections were moved to a new supplementary material. To exemplify the suggested indicator classification system, we now use four gauging stations as well as two months of the reference period (July 2003 and July 2015) with known drought events in Europe. Many figures were moved to the supplement to keep the manuscript as short as possible.

*RC: Section 2.2.1 still needs to be reviewed. Please, clarify whether the standardised variables have been calculated by adjusting the distribution or by using the z-score as a simplification.*
*The approximation of precipitation anomalies as a deviation from the mean is not the best approximation given that precipitation in general is non-normal, for shorter aggregation periods as well as in the context of semi-arid regions or regions with marked dry seasons where the authors focus part of the analysis.*

> **AC**: Due to the length of the manuscript, the two indicators SPI and SPEI were removed from the analysis.

*RC: The rationale given for the selection of sites and month of analysis shows that the analysis focuses on a drought indicator modelling study disconnected from a risk analysis. This approach is correct, but what I had stressed in my comment is that in order to make recommendations on how to properly introduce this information into risk analysis, more information and analysis is needed.*

*For instance, the authors mention in their reply "…that this kind of information is not available at the global scale and we recommend providing different hazard indicators covering different habituations to the streamflow regime. Then, people with local knowledge on this type of information can decide, which hazard indicator fits best to the targeted risk".*
*Here I understand that local knowledge includes the introduction of other variables that are defining drought risk, more analyses, etc. This position seems to be a simplification which historically has not helped the evolution of the concept of drought risk.*
*I encourage the authors to leave all necessary caveats open and to quantify appropriately here the nature of their results.*

> **AC**: In different paragraphs throughout the manuscript, we try to highlight that the manuscript is focused on drought hazard, and that an impact assessment is beyond the scope of this study. For example, to underline that the figures are not used for an impact assessment, we added the following sentence in the introduction (line 83 in the new version):

**This new methodology is exemplified at the global scale for eight existing and three newly developed SDHIs using a) modeled output from the global water resources and use model WaterGAP2.2d and b) observed monthly streamflow at four selected gauging stations.**

**AC**: At the beginning of Sect. 4 (Similarities and discrepancies in SDHIs as quantified by a global hydrological model), we address this issue as follows:

**The objective of this chapter is to identify which of the SDHIs presented in Table 1 can be meaningfully quantified at the global scale using WaterGAP 2.2d and which SDHIs are appropriate for monitoring different drought risks in large-scale DEWS. We emphasize that the objective is not a drought impact assessment, which is beyond the scope of this study. We want to show how the conceptual discrepancies and similarities between SDHIs (Sect. 3), which are of a general nature and apply to any month of the reference period, are translated into global-scale hazard indicators and how these indicators should be interpreted by end-users of a large-scale DEWS.**

**AC**: We conclude the manuscript with the following suggestion for future research:

**We suggest that future studies analyze how well these hazard indicators, in combination with suitable vulnerability and exposure indicators, can estimate drought impacts in the targeted risk systems at regional or national scales.**

**AC**: In this study, we give recommendations on how to select and interpret drought hazard indicators from a global model, a topic that has not been covered so far in such detail. From our point of view, this focus on the hazard aspect is not a simplification but a necessary delimitation of the topic.

*RC: The explanation for the choice of March 2002 is not entirely clear. It is also unclear why it is relevant that the two chosen points present the same value for the CQDI-Q80 in a given month and year. Why not using the points already presented in Figure 2. Please further elaborate for the sake of clarity.*

**AC**: To exemplify the suggested indicator classification system, we now use four gauging stations as well as two months of the reference period (July 2003 and July 2015) with known drought events in Europe. We emphasize in Sect. 4, however, the generality of the results:

**We want to show how the conceptual discrepancies and similarities between SDHIs (Sect. 3), which are of a general nature and apply to any month of the reference period, are translated into global-scale hazard indicators and how these indicators should be interpreted by end-users of a large-scale DEWS.**

**AC**: We deleted the analysis of the grid cells in Italy and Paraguay (old Sect. 4.2.4 and old Table 2) with the same CQDI1-Q80 value.

https://doi.org/10.5194/nhess-2022-174
Nat. Hazards Earth Syst. Sci. Discuss.

**"Analyzing the informative value of alternative hazard indicators for monitoring drought risk for human water supply and river ecosystems at the global scale"** by Claudia Herbert and Petra Döll

**Response to Anonymous Referee #4, report 1**

We thank you very much for your helpful comments and constructive suggestions for improving the manuscript. Below, each comment (in italics, indicated by "**RC**") is followed by our answer (normal font, indicated by "**AC**"). Changes in the manuscript are written in bold.

**General comments**

*RC: The authors collected 12 drought indicators and discussed which of them are useful to display in large-scale drought early warning systems. The authors concluded "drought magnitude is best quantified by return period or relative deviation from mean, and severity by return period or water volume below a threshold relative to mean annual streamflow (from abstract)".*

*Witnessing the recent frequent occurrence of severe drought events in many parts of the world, large-scale drought early warning is undoubtedly important. Reviewing and comparing drought indicators are also important, because it is widely recognized that drought is difficult to define or quantify. Although I fully understand the importance of the topic, I am unable to recommend publication of this work in the current from.*

*In short, the manuscript is unreadable. I have tried to go through this manuscript twice, but I couldn't complete it. Below I raise concrete examples why I am saying it is unreadable. Also I have to say that this paper is too long. In a nutshell, the authors' conclusions seem "drought magnitude is best quantified by return period or relative deviation from mean, and severity by return period or water volume below a threshold relative to mean annual streamflow (from abstract)". In my view, these conclusions are already well-perceived by hydrologists. I don't see any valid reasons why this long paper is needed to convey these unsurprising conclusions. Actually, I observe many paragraphs can be omitted. Again, I am not deprecating the authors' work. I just want to say that the manuscript is not ready to communicate with potential readers.*

> **AC**: We shortened the manuscript by ten pages and restructured and rephrased many sections to improve the readability and to clearly communicate the core messages and recommendations. Many sections were moved to a new supplementary material.

**Specific comments**

*RC: Line 29-33 "Drought poses…": Likely this paragraph can be omitted. The concepts of "hazard", "exposure", and "vulnerability" seem to appear only in the next paragraph and play marginal role in this work.*

> **AC**: We deleted this paragraph.

*RC: Line 34-52 "Drought risk indicators…": This paragraph contains a lot of information, but in my view, it is undirected. It really puzzled me what is the point of the authors. Finally, I noticed that I can better understand just ignore this paragraph and proceed the next paragraph.*

> **AC**: We deleted lines 34-44. We think that the second part of this paragraph focusing on current regional to global-scale DEWS is relevant for the manuscript.

*RC: Line 62-74 "Streamflow drought hazard can be estimated…": Again, the paragraph includes a lot of information,*

*but it is too specific. Readers wants to understand the background and objective of the study in Introduction, not the details. I have to say, this paragraph should be also omitted.*

> **AC**: We deleted this paragraph.

*Line 75-90 "SDHIs are commonly classified into ": the former part (general classification of drought indicators) is informative, but the latter part (explanation of the standardized streamflow indicator SSI) looks too specific.*

> **AC**: We deleted the sentence specifying averaging periods of SSI (lines 82-83). However, we think that the latter part of this paragraph is relevant as it describes why an improved classification system for drought hazard indicators is required from our point of view.

*RC: Line 91-105 "A further consideration in designing SDHI is how to conceptualize drouth in intermittent or highly seasonal streamflow regimes…": Again, the first sentence is okay, but the following part looks simply too specific and unorganized.*

> **AC**: We deleted lines 94-97 and shortened the remaining paragraph. We now describe only one important paper addressing drought in intermittent streamflow regimes in two sentences.

*RC: Line 104-116 "This paper analyzes": I think these are the only fully understandable paragraphs of this section. I just want to see a literature review which is directly relevant to these paragraphs.*

> **AC**: We considerably shortened the introduction to the most relevant aspects.

*RC: Line 136 "In several model intercomparison studies…": Too long. What the authors need to convey here is that WaterGAP is well validated, intensively compared with other models, and the results are usable for this analysis. I think this paragraph can be condensed into a few lines.*

> **AC**: The whole paragraph was moved to the new supplementary material and the core messages are now summarized in this section in a few sentences.

*RC: Line 168-267: Here the authors provide lengthy explanation for 12 indexes. I believe readers first want to know the definition of each index here, but I found it quite difficult, because each part is structured differently and including too many trivial information. I think here definition of index (hopefully with equations), threshold, and the parameters (in particular the time window the authors chose) would far enough to proceed reading. The remaining information can be put into supplemental material or appendix of this paper.*

> **AC**: The sections about SPI and SPEI were deleted, since the indicators are not assessed any more (due to the length of the manuscript). For the severity indicators, equations were added that define the computation of the respective deficit. The time period is included in a new section 2.1 describing the observed and modeled streamflow data used in this study.
>
> We are aware that the detailed indicator description may seem trivial to experts in this field. However, after studying the literature on drought (hazard) indicators, definitions are sometimes imprecise, missing or wrong, and we feel that such a detailed description is valuable especially for people new to the field.

*RC: Line 295 "The choice of drought hazard indicators implies assumptions about the habituation of the system at risk": I was totally puzzled by this part and following discussion. First of all, what is "choice"? Who chooses for what? What does "imply" mean? Whose "assumptions"? The following lines do not answer any of these questions. Actually, hereafter, I felt myself reading a unpolished first draft.*

> **AC:** We rephrased especially the first paragraph of this section to address the issues raised by the reviewer (see paragraph below in bold). The first line now reads **"The selection of drought hazard indicators for a**

**DEWS requires a clear definition of […]".** We think that it is now clear that it is the end-user and/or developer of a DEWS who chooses, or selects, hazard indicators.

We further write "**Consequently, the selection of an indicator requires a definition, often based on assumptions, about "what is normal or needed", i.e., to what the risk system is habituated to.**" to introduce the terms "assumption" and "habituation".

The term "imply" generally means that an idea is involved without saying it directly. So in this case, an indicator implies a certain idea (in this case, an assumption) about what the risk system is habituated to.

Rephrased paragraph:

**The selection of drought hazard indicators for a DEWS requires a clear definition of "the risk of what for whom". Drought hazard indicators are risk system-specific (Blauhut et al. 2021), and there is not one that fits all. Drought is usually conceptualized as anomaly ("less water than normal") and/or deficit ("less water than needed"). Consequently, the selection of an indicator requires a definition, often based on assumptions, about "what is normal or needed", i.e., to what the risk system is habituated to. In the case of streamflow, people and ecosystems are assumed to have adapted to certain characteristics of the flow regime. For example, if drought indicators are computed based on the calendar month-specific distribution of streamflow values, it is implicitly assumed that the risk system has adapted to the seasonality of streamflow. But also temporally constant thresholds, which have traditionally been used to define hydrological droughts (Stahl et al. 2020), are suitable for certain systems, e.g., for computing drought risk for electricity generation by thermal power plants, which require a certain minimum streamflow for operation.**

*RC: Table 1: Tables in a paper is typically a summary of some specific paragraphs or sections. However, I couldn't figure out how Table 1 and text are related. Furthermore, the table includes a lot of subjective judgement (e.g. "a certain degree of" "suitable for" "better proxy for" without showing rationale. It doesn't help my understanding of discussion.*

**AC:** Table 1 summarizes for each indicator the assumed habituation, which is the topic of this section. We think that a summary table is valuable here to give a quick overview of the numerous indicators. In the table caption, we added the following sentence to explain the terms "a certain degree", etc.:

**The general terms "a certain degree" or "a certain reduction" in the first column are specified in a drought assessment by selected thresholds for drought definition.**

**AC:** The term "a better proxy for" is not used any more, since the indicators SPI and SPEI were deleted.

*RC: Line 319 "In hydrology, flow duration curves showing the fraction of the time …": I couldn't contextualize the sentence here. I couldn't see any logical flow. Many readers expect the first sentence of paragraph is a summary of the paragraph, but this is not case of this manuscript.*

**AC:** We deleted this first sentence and moved or deleted most of this paragraph. The first paragraph is now focused on a comparison between percentile-based indicators and relative deviations. Lines 326-334 describing the findings of Kumar et al. (2009) were deleted.

*RC: Line 335 "Similar to the example above from Kumar et al. (2009), the 20th streamflow percentile (or SSI1=-0.84) would correspond to a low relative streamflow deviation (e.g. -20%) in a humid region (low interannual variability) compared to a higher deviation (e.g. -50%) in a semi-arid region (high interannual variability)": Impossible to understand. Is this a specific result of Kumar et al. (2009)? Must the readers go through Kumar et al. (2009)? Does SSI1=-0.84, 20%, 50% have any specific meaning or just some example numbers? Actually, I got a similar impression for numerous lines hereafter. First I blamed myself about my carelessness, but soon I stopped it: simply the text is too unorganized to read.*

**AC:** We rephrased the paragraph as follows:

**Utilization of percentile-based indicators (e.g., SSI12, SSI1, and CQDI1(Q80) in Table 1) implies that people in different climate regions and social systems are equally habituated to a certain interannual variability, which is most likely not the case. The 20th streamflow percentile (or SSI1 = -0.84) would correspond to a low relative streamflow deviation (e.g. -20%) in a humid region (low interannual variability) compared to a higher deviation (e.g. -50%) in a semi-arid region (high interannual variability). Hence, percentile-based indicators might underestimate streamflow drought hazard in semi-arid areas where people (and ecosystems, albeit possibly to a lower degree) are often more vulnerable to reductions in water availability.**

*RC: Line 346 "In conclusion, percentile-based hazard indicators and relative deviation from the long-term mean or median should be used complementarily in large scale DEWS in combination with adequate vulnerability and exposure indicators to cover different drought risks": I am happy to see conclusion was given here. It would be even better if this sentence comes at the beginning of this paragraph.*

**AC:** We shortened and restructured the paragraph comparing percentile-based indicators and relative deviations to improve readability. We think that the sentence in line 346 is a conclusion drawn from the indicator comparison and it should therefore remain at the end.

*RC: Line 382 "This concept is not new…, Nevertheless, only a few …": Hard to read. What is the point?*

**AC:** We deleted the sentence in line 382.

*RC: Line 409 "The indicator types (columns in Fig1) include the volume-based anomaly, the standardized or percentile-based anomaly, and the relative deviation, all of which are described in the previous section": I couldn't find where exactly these are described. If these categories are important, better to define explicitly in prior.*

**AC:** We replaced "**all of which are described in the previous section**" by "**(Sect. 2.3)**". Sect. 2.3 (previously Sect. 2.2) comprises the indicator description.

*RC: Figure 1 Caption "Classification system" and diagram: What does "system" mean? Many times the verb "choose" appear in the diagram, but who chooses for what? What the arrows indicate? Again, the Figure 1 doesn't enhance my understanding at all.*

**AC:** We think that the term "classification system" (i.e., putting items into categories or groups based on their characteristics) is suitable here, since indicators are grouped based on their characteristics.

**AC:** We replaced the term "choose" by "select" in Figure 1. In the figure caption, we rephrased the following sentence: "**The dark grey boxes indicate decisions that have to be made when computing the indicators, e.g. which averaging period is selected**".

Since Sect. 3.1 now starts with "**The selection of drought hazard indicators for a DEWS**", we think that it is clear at this point that the end-user or developer of a DEWS "chooses" (or selects) indicators for the DEWS. Moreover, the second to last paragraph of the introduction also clarifies this aspect: "**This paper analyzes which SDHIs are suitable for assessing and monitoring drought risk for human water supply from surface water and for river ecosystems in large-scale DEWS. We propose a systematic approach to indicator selection […]**".

---

## Author Response (AR3)

https://doi.org/10.5194/nhess-2022-174
Nat. Hazards Earth Syst. Sci. Discuss.
**"Analyzing the informative value of alternative hazard indicators for monitoring drought risk for human water supply and river ecosystems at the global scale"** by Claudia Herbert and Petra Döll

**Response to Editor**

Dear Ms. Van Loon,

We are very grateful for your thorough review of the manuscript and the many helpful suggestions that helped improving the text. Below, each of your comments (in italics, indicated by "**EC**") is followed by our answer (normal font, indicated by "**AC**"). Changes in the manuscript are written in bold. Lines indicated in the authors' comments refer to the revised manuscript without markups.

**EC:** *Dear authors,*
*Thanks for doing the effort to restructure the paper. I think it has removed massively now. I like Section 3.1 that explains the implicit assumptions and the figures are clearer. A few changes are still needed to further improve the paper. Please address the suggestions below.*

> **AC:** Thank you very much for the positive feedback.

**EC:** *I suggest the authors to remove the word "risk" from the paper when they are talking about monitoring drought risk, assessing drought risk, etc. because (as the reviewer indicated) the paper only includes hazard, while exposure and vulnerability are not included. A few examples:*
*- L.11: "monitoring drought risk" >> "monitoring drought"*
*- L.19: "assessing the drought risk of" >> "assessing the drought hazard of"*

> **AC:** We implemented your suggestion and replaced the word "risk" by "hazard" (L. 2, 21, 84, 299, Table 1, rows 1 and 2, L. 332, and 709) or removed it (L. 11 and 334). We now use the word "risk" only when we refer to the risk system or the targeted risk of a drought assessment as well as in one citation (L. 41).

**EC:** *- L.21-22: "habituation of the risk system" I still find 'habituation' a very unclear term. After reading the paper a few times I understand what you mean, but for first time readers it is not very helpful. Also 'risk system' is not clear. I suggest to either find different words / terms (for example 'adaptation', like in l.601) or define them at the start of the abstract and of the introduction and then refer back to this.*

> **AC:** In the abstract (L. 12), we added the following definitions for the terms "habituation" and "risk system":
>
> We recommend considering the habituation of the system at risk **(e.g., a drinking water supplier or small-scale farmers in a specific region)** to the streamflow regime when selecting indicators**, i.e., users of the DEWS should determine to which type of deviation from normal (e.g., a certain interannual variability or a certain relative reduction of streamflow) the risk system of interest has become used and adapted.**
>
> **AC:** In the introduction (L. 68), we added:
>
> Also, in most descriptions of drought indicator calculations, it is not made explicit what is assumed to be "normal", **i.e., to what people and ecosystems are used and adapted, hereafter referred to as "*habituation*".**
>
> **AC:** After the introduction, the words "habituation" and "risk system" are again used in Sect. 3. Here, we added a reference to the definitions section 1 in L. 284 and 285:

In the following Sect. 3.1, aspects that relate to the conceptual drought definition are discussed comprising the description of the targeted drought risk and the system at risk **(see Sect. 1).** In particular, assumptions about the habituation **(see Sect. 1)** of the system at risk to the streamflow regime are discussed, […].

*EC: - L.417: "monitoring different drought risks" >> "monitoring different drought hazards"*

**AC:** We implemented your suggestion.

*EC: The selection of gauging stations and time periods is much more logical and justifiable now. For the time periods you select two drought events in Europe for testing global-scale drought hazard indicators. I strongly suggest to replace one of them with a drought event in a different continent where processes (and habituation) may be different. l.421-422: "The indicators are illustrated in global maps for two example months, July 2003 and July 2015, with known drought events in Europe."*

**AC:** We replaced July 2015 with September 1993, where a drought occurred in South Africa. We adjusted the following text sections:

**L. 424: The indicators are illustrated in global maps for two example months capturing known drought events in Europe (July 2003) and South Africa (September 1993), two regions that are characterized by different streamflow regimes and assumed habituation.**

**L. 440: In September 1993, SSI1 indicates a higher drought hazard than EP1 for the Orange River along the Namibia-South Africa border, but a lower hazard in a few grid cells in central South Africa and Lesotho.**

**L. 449:** Grid cells where gamma fitting was rejected in the calendar months July **and September** based on the KS test (Sect. 2.3.1) are shown in grey in Figs. 2a and S3a […].

**L. 491: For instance, the drought event in 2003 in central and eastern Europe (Fig. 3) identified by CQDI1(Q80) is not indicated by CQDI1(WUs), while the latter shows an additional drought hazard in the northern part of South Africa (Fig. S4). This is because CQDI1(WUs) strongly depends on surface water stress, which is generally low in Europe and high in South Africa (Fig. A1c).**

**L. 436 and 559:** "July 2015" replaced by "September 1993".

**AC:** In the supplement, we replaced Figs. S3 to S5 with the results for September 1993.

*EC: You need to increase the resolution of the maps and also make them bigger, because if you want the reader to look at the drought in Europe, then Europe should be distinguishable in the figures.*

**AC:** We increased the resolution of the global maps from 600 dpi to 1200 dpi. Figures 2, 3, S3, and S4 are now full-page figures. We increased the size of Figs. 5 and S5 as much as possible.

***Minor comments:***

*EC: What do you mean with "that is also relevant for meteorological or soil moisture drought" in l.12? Please remove or make into a separate sentence.*

**AC:** We removed this part of the sentence.

*EC: l.16-17: remove "and other drought hazard indicators"*

**AC:** Done.

*EC: l.19-20: "suitable for assessing the drought risk of 1) river ecosystems, 2) water users without access to large res-*

*ervoirs, 3) water users with access to large reservoirs, and 4) that are suitable for informing reservoir managers"* point *4 is not part of the list, so change to >> "suitable for assessing the drought hazard of 1) river ecosystems, 2) water users without access to large reservoirs, and 3) water users with access to large reservoirs, and that are suitable for informing reservoir managers"*

**AC:** We implemented your suggestion.

*EC: l.40-43: "A stakeholder survey encompassing 33 regional to global drought early warning systems (DEWS) revealed that streamflow drought hazard indicators (SDHIs) are rarely applied in DEWS" >> this needs a reference. Or, if you did this yourself, methods and results should be included (in supplementary material?).*

**AC:** We added the reference Bachmair et al. (2016).

*EC: l.46: "European Drought Monitor" >> do you mean "European Drought Observatory"?*

**AC:** Yes, thank you. We changed the name accordingly.

*EC. l.137-138: "First, the 30 monthly streamflow values per calendar month were fitted to the gamma distribution using the R package fitdistrplus." >> the other way around, a gamma distribution was fitted to the streamflow data.*

**AC:** We corrected the sentence.

*EC: l.597: "are to inform" >> "aim to inform"*

**AC:** We replaced "are" by "aim".